# Statistical, Robustness, and Computational Guarantees for Sliced Wasserstein Distances

**Sloan Nietert**
Cornell University
nietert@cs.cornell.edu

**Ritwik Sadhu**
Cornell University
rs2526@cornell.edu

**Ziv Goldfeld**
Cornell University
goldfeld@cornell.edu

**Kengo Kato**
Cornell University
kk976@cornell.edu

## Abstract

Sliced Wasserstein distances preserve properties of classic Wasserstein distances while being more scalable for computation and estimation in high dimensions. The goal of this work is to quantify this scalability from three key aspects: (i) empirical convergence rates; (ii) robustness to data contamination; and (iii) efficient computational methods. For empirical convergence, we derive fast rates with explicit dependence of constants on dimension, subject to log-concavity of the population distributions. For robustness, we characterize minimax optimal, dimension-free robust estimation risks, and show an equivalence between robust sliced 1-Wasserstein estimation and robust mean estimation. This enables lifting statistical and algorithmic guarantees available for the latter to the sliced 1-Wasserstein setting. Moving on to computational aspects, we analyze the Monte Carlo estimator for the average-sliced distance, demonstrating that larger dimension can result in faster convergence of the numerical integration error. For the max-sliced distance, we focus on a subgradient-based local optimization algorithm that is frequently used in practice, albeit without formal guarantees, and establish an $O(\epsilon^{-4})$ computational complexity bound for it. Our theory is validated by numerical experiments, which altogether provide a comprehensive quantitative account of the scalability question.

## 1 Introduction

Sliced Wasserstein distances consider the average or maximum of Wasserstein distances between one-dimensional projections of the two distributions. Formally, for $1 \leq p < \infty$, they are defined as

$$\underline{\mathsf{W}}_p(\mu, \nu) := \left[ \int_{\mathbb{S}^{d-1}} \mathsf{W}_p^p(\mathfrak{p}_\sharp^\theta \mu, \mathfrak{p}_\sharp^\theta \nu) d\sigma(\theta) \right]^{1/p} \quad \text{and} \quad \overline{\mathsf{W}}_p(\mu, \nu) := \max_{\theta \in \mathbb{S}^{d-1}} \mathsf{W}_p(\mathfrak{p}_\sharp^\theta \mu, \mathfrak{p}_\sharp^\theta \nu), \quad (1)$$

where $\mathfrak{p}_\sharp^\theta \mu$ is the pushforward of $\mu$ under the projection $\mathfrak{p}^\theta : x \mapsto \theta^\intercal x$ from $\mathbb{R}^d$ to $\mathbb{R}$ and $\sigma$ is the uniform distribution on the unit sphere $\mathbb{S}^{d-1}$ in $\mathbb{R}^d$. Sliced Wasserstein distances were introduced in [49] as a means to mitigate the computational burden of evaluating classic $\mathsf{W}_p$, which rapidly becomes excessive as $d$ grows. Indeed, sliced distances are readily computable using the closed-form expression for $\mathsf{W}_p$ between distributions on $\mathbb{R}$ (as the $L^p$ norm between quantile functions). Further, $\underline{\mathsf{W}}_p$ and $\overline{\mathsf{W}}_p$ are metrics on $\mathcal{P}_p(\mathbb{R}^d)$ and generate the same topology as classic $\mathsf{W}_p$ [10, 44, 6, 42]. As such, the sliced distances have been applied to various statistical inference and machine learning tasks, including barycenter computation [49, 9], generative modeling [18, 17, 44, 58], autoencoders [29], differential privacy [50], Bayesian computation [41] and topological data analysis [12].

36th Conference on Neural Information Processing Systems (NeurIPS 2022).

## 1.1 Statistical, Robustness, and Computational Aspects of Sliced Distances

In practice, the sliced Wasserstein distances in (1) must be approximated from two aspects: (i) empirically estimate the population measures $\mu$ and $\nu$, and (ii) employ numerical integration or optimization methods to compute the average- or max-sliced distances, respectively. While these approximations are implemented in all but every application of sliced distances, formal guarantees concerning their accuracy are partial or even missing. For the estimation error, the question boils down to quantifying the rate at which $\underline{W}_p(\hat{\mu}_n, \mu)$ and $\overline{W}_p(\hat{\mu}_n, \mu)$ decay to 0, where $\hat{\mu}_n$ is the empirical distribution of $n$ independent observations from $\mu$.[1] These rates are known to adapt to the low-dimensionality of the projected distribution, but previously derived rates do not seem to be sharp [35], rely on high-level assumptions that may be hard to verify in practice [47], or hide dimension-dependent constants whose characterization is crucial for understanding the scalability of sliced distances [42]. More recently, [37] showed that near-parametric rates (i.e., up to polylogarithmic factors) are achievable for the average-sliced $p$-Wasserstein distance in the two-sample case, under the alternative ($\mu \neq \nu$). Limit distributions for sliced Wasserstein distances were studied in [37, 21, 60, 59], but these results inherently neglect constants and dependence on dimension.

Concerning robust estimation, while these aspects were studied for classic Wasserstein distances [4, 45, 40, 30, 53, 46], they were not considered under sliced $W_p$. Improvement in robustness to outliers due to projection-averaging was demonstrated for the Cramér-von Mises statistic in the context of multivariate two-sample testing [27]. It therefore stands to reason that similar gains would emerge for Wasserstein distances, which is especially appealing since robust estimation of classic $W_p$ in high dimensions is hard. Indeed, [46] showed that when an $\epsilon$-fraction of data is contaminated, $W_p$ admits worst-case estimation risk $\sqrt{d}\epsilon^{1/p-1/2}$ over distributions with bounded covariance. Consequently, obtaining accurate estimates of $W_p$ from contaminated data is infeasible in high dimensions when $\epsilon = \Omega(1)$, which further motivates exploring robustness under slicing.

From the computational standpoint, the average-sliced distance $\underline{W}_p$ is typically computed using Monte Carlo (MC) integration [28, 42]. The accuracy of this approach strongly depends on the variance of the function $\theta \mapsto W_p^p(\mathfrak{p}_\sharp^\theta \mu, \mathfrak{p}_\sharp^\theta \nu)$ when $\theta$ is uniformly distributed on $\mathbb{S}^{d-1}$, which may scale badly with $d$. A bound on the MC integration error in terms of this variance was provided in [42] but without further analysis to control it by basic properties of the population distribution or characterize its dependence on $d$. Accordingly, the accuracy of the MC-based approach for computing $\underline{W}_p$ stands unresolved. Recently, [43] used the conditional central limit theorem [51] to derive a Gaussian approximation of $\underline{W}_2$ that can be computed in closed form. The accuracy of this approximation may improve as $d \to \infty$, contingent on certain weak dependence assumptions on the data distribution. A popular approach for computing the max-sliced distance is the heuristic alternating optimization procedure from [28, 17], which, however, lacks formal convergence guarantees. More recently, computational aspects of the so-called "projection-robust" Wasserstein distance, which considers projections to $k$-dimensional subspaces, were explored in [48, 34, 25].[2] As maximization of projected distance is a non-convex and non-smooth optimization problems, these works considered convex relaxations [48] or entropic regularization [34, 25] to prove approximate convergence to a stationary point.

## 1.2 Contributions

The goal of this paper is to close the aforementioned gaps by (i) deriving fast empirical convergence rates for sliced distances with explicit dimension dependence; (ii) characterizing minimax optimal robust estimation rates with improved dependence on dimension; and (iii) providing formal guarantees for frequently used methods for computing both the average- and max-sliced $W_p$. Focusing on log-concave distributions, we show that both average- and max-sliced empirical distances converge as $n^{-1/\max\{2,p\}}$, which is sharp as it matches lower bounds from [8]. Furthermore, we characterize the constant in terms of $d$ and elementary properties of the population distribution (e.g., mean, moments, covariance matrix). Our derivation leverages the machinery of [8] for analyzing empirical convergence of Wasserstein distances between log-concave measures on $\mathbb{R}$. To that end, we show that log-concavity is preserved under projections and derive lower bounds on the Cheeger constant of the projected distribution. Our results elucidate scaling rates of $d$ with $n$ for which (high-dimensional) empirical convergence holds true, thereby addressing the scalability of empirical estimates question.

---

[1]The two-sample setting, which concerns the convergence $\underline{W}_p(\hat{\mu}_n, \hat{\nu}_n)$ and $\overline{W}_p(\hat{\mu}_n, \hat{\nu}_n)$ towards the corresponding distance between the population measures, is also of interest.

[2]Despite the name "projection-robust", these works do not explore robust estimation.

For robustness guarantees, we formalize minimax risk for robust estimation under sliced $\mathsf{W}_p$ with total variation (TV) contamination and prove that $\overline{\mathsf{W}}_p$ enjoys a dimension-free risk of $\epsilon^{1/p-1/q}$ when clean distributions have bounded $q$th moments for $q > p$ and the corruption level is at most $\epsilon$. $\underline{\mathsf{W}}_p$ admits a strictly smaller risk which scales at the same rate when $q = O(1)$. In contrast, the comparable risk for classic $\mathsf{W}_p$ in this setting acquires an extra $\sqrt{d}$ factor. Using the framework of generalized resilience [61], we extend these guarantees to the finite-sample setting with adversarial corruptions, obtaining matching rates up to an added empirical approximation term. Furthermore, when $p = 1$, we prove equivalence between standard mean resilience [55] and resilience w.r.t. $\overline{\mathsf{W}}_1$, allowing one to lift statistical and algorithmic guarantees for robust mean estimation to the sliced $\mathsf{W}_1$ setting.

Lastly, we provide formal guarantees for popular methods for computing $\underline{\mathsf{W}}_p$ and $\overline{\mathsf{W}}_p$, which were until now lacking. Our analysis relies on showing that $w_p : \theta \mapsto \mathsf{W}_p(\mathsf{p}_\sharp^\theta \mu, \mathsf{p}_\sharp^\theta \nu)$ and its $p$th power are Lipschitz continuous on $\mathbb{S}^{d-1}$ and deriving sharp bounds on their Lipschitz constants. Having that, we analyze the MC estimator for the average-sliced distance, and use concentration of Lipschitz functions on the unit sphere to bound the variance of $w_p^p$. The obtained bound reveals that higher dimension can in fact shrink the MC error when the covariance matrices have bounded operator norms. We numerically verify this surprising observation on synthetic examples.

For the max-sliced distance, we analyze the heuristic algorithm from [17, 28], which utilizes alternating subgradient-based optimization. We observe that in addition to being Lipschitz continuous, the optimization objective for $p = 2$ is weakly convex with easily computable gradients. This lets us cast the algorithm from [17, 28] under the proximal stochastic subgradient optimization framework of [14], from which we obtain local solutions for $w_2(\theta)$ with $O(\epsilon^{-4})$ computational complexity. An empirical comparison with the more advanced approaches of [34, 25] for computing the projection-robust Wasserstein distance (with $k = 1$ to match the sliced framework) based on Riemannian optimization reveals that our subgradient-based method is significantly faster in terms of iteration complexity and computation time. We also consider global optimization by showing that $\overline{\mathsf{W}}_p$ computation matches the framework of [36] for Lipschitz function optimization over convex domains. Adapting their LIPO algorithm to our problem, we obtain a provably consistent algorithm for computing $\overline{\mathsf{W}}_p$. However, the number of function evaluations that LIPO requires grows exponentially with dimension, which renders the locally optimal subgradient method preferable when dimension is large.

## 2 Background and Preliminaries

**Notation.** We use $\| \cdot \|$ for the Euclidean norm in $\mathbb{R}^d$. The operator norm for matrices is $\| \cdot \|_{\mathrm{op}}$. The unit sphere in $\mathbb{R}^d$ is denoted by $\mathbb{S}^{d-1}$, while $\mathbb{B}^d$ is the unit ball. Let $\mathcal{P}(\mathbb{R}^d)$ denote the space of Borel probability measures on $\mathbb{R}^d$ equipped with the TV metric $\|\mu - \nu\|_{\mathrm{TV}} = \frac{1}{2}|\mu - \nu|(\mathbb{R}^d)$, and set $\mathcal{P}_p(\mathbb{R}^d) := \{\mu \in \mathcal{P}(\mathbb{R}^d) : \int \|x\|^p d\mu(x) < \infty\}$ for $1 \leq p < \infty$. The support of $\mu \in \mathcal{P}(\mathbb{R}^d)$ is denoted as $\mathrm{spt}(\mu)$, and we write $\mu \leq \nu$ for setwise inequality. For for a measurable map $f$, the pushforward of $\mu$ under $f$ is denoted as $f_\sharp \mu = \mu \circ f^{-1}$, i.e., if $X \sim \mu$ then $f(X) \sim f_\sharp \mu$. For two numbers $a$ and $b$, we use the notation $a \wedge b = \min\{a, b\}$ and $a \vee b = \max\{a, b\}$. The distance between a set $S$ and a point $x$ in a metric $(\mathcal{X}, d)$ space is defined as $\mathrm{dist}(x, S) := \inf_{y \in S} d(x, y)$.

Some of our results assume log-concavity of the population distribution. A probability measure $\mu \in \mathcal{P}(\mathbb{R}^d)$ is *log-concave* if for every nonempty compact sets $A, B \subset \mathbb{R}^d$ and $\lambda \in [0, 1]$, we have $\mu(\lambda A + (1 - \lambda)B) \geq \mu(A)^\lambda \mu(B)^{1-\lambda}$. A probability density function $f$ on $\mathbb{R}^d$ is called *log-concave* if for every $x, y \in \mathbb{R}^d$ and $\lambda \in [0, 1]$, it satisfies $f(\lambda x + (1 - \lambda)y) \geq f(x)^\lambda f(y)^{1-\lambda}$. Any non-degenerate distribution is log-concave if and only if it has a log-concave density [11, Theorem 1.1]. For $\beta \in (0, 2]$, let $\psi_\beta(t) = e^{t^\beta} - 1$ for $t \geq 0$, and recall that the corresponding Orlicz (quasi-)norm of a real-valued random variable $X$ is defined as $\|X\|_{\psi_\beta} := \inf\{c > 0 : \mathbb{E}[\psi_\beta(|X|/c)] \leq 1\}$. A Borel probability measure $\mu \in \mathcal{P}(\mathbb{R}^d)$ is called *sub-Gaussian* if $\|\|X\|\|_{\psi_2} < \infty$ for $X \sim \mu$.

**Classic and sliced Wasserstein distances.** For $1 \leq p < \infty$, the $p$-Wasserstein distance between $\mu, \nu \in \mathcal{P}_p(\mathbb{R}^d)$ is $\mathsf{W}_p(\mu, \nu) := \inf_{\pi \in \Pi(\mu,\nu)} \left[ \int_{\mathbb{R}^d \times \mathbb{R}^d} \|x - y\|^p d\pi(x, y) \right]^{1/p}$, where $\Pi(\mu, \nu)$ is the set of couplings of $\mu$ and $\nu$. $\mathsf{W}_p$ is a metric on $\mathcal{P}_p(\mathbb{R}^d)$ and metrizes weak convergence plus convergence of $p$th moments. While for $d > 1$ the definition of $\mathsf{W}_p$ generally amounts to an infinite-dimensional optimization problem, the expression simplifies when distributions are supported in $\mathbb{R}$. This motivates

the notion of the average- and max-sliced Wasserstein distances from (1). Both sliced distances are also metrics on $\mathcal{P}_p(\mathbb{R}^d)$ that induce the same topology as $\mathsf{W}_p$ [10, 44, 6, 42].

To present the simple one-dimensional formulae for $\mathsf{W}_p$, for $\mu \in \mathcal{P}(\mathbb{R}^d)$ and $\theta \in \mathbb{S}^{d-1}$, let $F_\mu(t; \theta) := \mu\big(\{x \in \mathbb{R}^d : \theta^\intercal x \leq t\}\big)$ be the distribution function of $\mathfrak{p}_\sharp^\theta \mu$, and $F_\mu^{-1}(\tau; \theta) = \inf\{t \in \mathbb{R} : F_\mu(t; \theta) \geq \tau\}$, for $\tau \in (0, 1)$, be the quantile function. The $\mathsf{W}_p$ between measures on $\mathbb{R}$ amounts to the $L^p$ distance between their quantile functions: $\mathsf{W}_p^p(\mathfrak{p}_\sharp^\theta \mu, \mathfrak{p}_\sharp^\theta \nu) = \int_0^1 \big|F_\mu^{-1}(\tau; \theta) - F_\nu^{-1}(\tau; \theta)\big|^p d\tau$. For $p = 1$, the expression further simplifies to $\mathsf{W}_1(\mathfrak{p}_\sharp^\theta \mu, \mathfrak{p}_\sharp^\theta \nu) = \int_\mathbb{R} \big|F_\mu(t; \theta) - F_\nu(t; \theta)\big| \, dt$.

Sliced Wasserstein distances between empirical distributions can be computed via order statistics. Let $\hat{\mu}_n := n^{-1} \sum_{i=1}^n \delta_{X_i}$ and $\hat{\nu}_n := n^{-1} \sum_{i=1}^n \delta_{Y_i}$ be the empirical distributions of samples $X_1, \dots, X_n$ and $Y_1, \dots, Y_n$. For each $\theta \in \mathbb{S}^{d-1}$, denote $X_i(\theta) = \theta^\intercal X_i$, and let $X_{(1)}(\theta) \leq \cdots \leq X_{(n)}(\theta)$ be the order statistics; define $Y_{(1)}(\theta) \leq \cdots \leq Y_{(n)}(\theta)$ analogously. By Lemma 4.2 in [8], we have $\mathsf{W}_p^p(\mathfrak{p}_\sharp^\theta \hat{\mu}_n, \mathfrak{p}_\sharp^\theta \hat{\nu}_n) = n^{-1} \sum_{i=1}^n \big|X_{(i)}(\theta) - Y_{(i)}(\theta)\big|^p$. The sliced distances $\underline{\mathsf{W}}_p$ and $\overline{\mathsf{W}}_p$ are computed by integrating or maximizing the above over $\theta \in \mathbb{S}^{d-1}$.

## 3 Empirical Convergence Rates

We study empirical convergence rates of sliced Wasserstein distances for log-concave distributions. The next result gives sharp one-sample rates with explicit dependence on the effective dimension.

**Theorem 1** (Empirical convergence rates). *Let $1 \leq p < \infty$ and $n \geq 2$. Suppose that $\mu \in \mathcal{P}(\mathbb{R}^d)$ is log-concave with covariance matrix $\Sigma$ and set $k = \operatorname{rank}(\Sigma)$. Then,*

$$\mathbb{E}\big[\underline{\mathsf{W}}_p(\hat{\mu}_n, \mu)\big] \lesssim_p \frac{\|\Sigma\|_{\mathrm{op}}^{1/2} \sqrt{(\log n)^{\mathbb{1}_{\{p=2\}}}}}{n^{1/(2\vee p)}}, \tag{2a}$$

$$\mathbb{E}\big[\overline{\mathsf{W}}_p(\hat{\mu}_n, \mu)\big] \lesssim_p \frac{\|\Sigma\|_{\mathrm{op}}^{1/2} k \log n}{n^{1/p}} + \frac{\|\Sigma\|_{\mathrm{op}}^{1/2} \sqrt{k \log n}}{n^{1/(2\vee p)}} + \frac{\|\Sigma\|_{\mathrm{op}}^{1/2} \sqrt{(\log n)^{\mathbb{1}_{\{p=2\}}}}}{n^{1/(2\vee p)}}. \tag{2b}$$

The proof of Theorem 1, in Appendix D.1, employs the machinery of [8] for analyzing empirical convergence of log-concave distributions on $\mathbb{R}$ based on their Cheeger constant (see Appendix A). For (2a), we show that log-concavity is preserved under projections and lower bound the Cheeger constant of the projected distribution by $c/\|\Sigma\|_{\mathrm{op}}$, uniformly in $\theta \in \mathbb{S}^{d-1}$. For (2b), concentration and covering arguments enables approximating the expected max-sliced distance by $\sup_{\theta \in \mathbb{S}^{d-1}} \mathbb{E}\big[\mathsf{W}_p(\mathfrak{p}_\sharp^\theta \hat{\mu}_n, \mathfrak{p}_\sharp^\theta \mu)\big]$, for which the aforementioned (uniform in $\theta$) bounds are applicable.

**Remark 1** (Lower bounds). *The rate in (2a) is sharp up to log factors over the log-concave class. Corollary 6.14 in [8] implies that $\mathbb{E}[\underline{\mathsf{W}}_p(\hat{\mu}_n, \mu)]^p \geq c_p (\operatorname{tr}(\Sigma)/d)^{p/2}/(n(\log n)^{p/2})$, for $\mu = \mathcal{N}(0, \Sigma) \in \mathcal{P}(\mathbb{R}^d)$ and any $p > 2$. For $p = 2$, a similar computation yields $\mathbb{E}[\underline{\mathsf{W}}_2(\hat{\mu}_n, \mu)]^2 \gtrsim (\operatorname{tr}(\Sigma)/d) \log \log n/n$, while for $p \in [1, 2)$, $\mathbb{E}[\underline{\mathsf{W}}_p(\hat{\mu}_n, \mu)] \geq \mathbb{E}[\underline{\mathsf{W}}_1(\hat{\mu}_n, \mu)] \gtrsim \sqrt{\tau(\Sigma)^2/n}$ where $\tau(\Sigma) = \frac{1}{d} \sum_{i=1}^d \sqrt{\lambda_i(\Sigma)}$ is the average of root eigenvalues of $\Sigma$. Since $\overline{\mathsf{W}}_p(\hat{\mu}_n, \mu) \gtrsim \underline{\mathsf{W}}_p(\hat{\mu}_n, \mu)$, this also yields a lower bound for $\overline{\mathsf{W}}_p$, while [47] gives a $\sqrt{d/n}$ lower bound under the $T_p$ inequality.*

**Remark 2** (Comparison with [47, 35]). *In [47], empirical rates for $\overline{\mathsf{W}}_p$ were derived under a high-level $T_{p'}(\sigma^2)$ assumption on $\mu$. Our rate of decay from (2b) is faster, while replacing their entropy-transport inequality condition with log-concavity. The bounds for $\overline{\mathsf{W}}_p$ in [35, Theorem 3.6] assume the projection Poincaré inequality and $M_q := (\mu\|x\|^q)^{1/q} < \infty$ for $q > p$, and matches (2a) as $q \to \infty$ in terms of the dependence on $n$. However, their dependence on $d$ is implicit through $M_q$ which typically grows prohibitively with $q$ and $d$. Our log-concavity assumption is strictly stronger than the Poincaré inequality, but yields a bound in terms of $\|\Sigma\|_{\mathrm{op}}$ which, for example, is constant in $d$ when $\Sigma = I_d$.[3] Finally, we note that the bound for $\overline{\mathsf{W}}_p$ in (2b) adapts to the effective dimensionality $k$ of the data, contrasting previously available bounds that depend on the ambient dimension $d$.*

**Remark 3** (Concentration bounds). *Combining the expectation bounds from Theorem 1 with [35, Theorem 3.8] yields concentration bounds for empirical sliced distances. These are presented in Appendix B and are later used to derive formal guarantees for computing $\overline{\mathsf{W}}_p$.*

---

[3]A recent preprint [5], that was posted on arXiv after this paper was submitted, shows that an improved estimate holds with high probability (compared to the convergence in expectation studied herein) for isotropic log-concave random vectors; cf. Equation (1.13) therein.

When $p > 2$, the rates in Theorem 1 are slower than parametric. Nevertheless, in the two-sample case with $\mu \neq \nu$, parametric rates are attainable uniformly in $p$ for compactly supported distributions.

**Proposition 1** (Parametric rates under the alternative). *Let $1 \leq p < \infty$, and suppose that $\mu, \nu$ have compact supports with $\mathrm{diam}\big(\mathrm{spt}(\mu)\big) \vee \mathrm{diam}\big(\mathrm{spt}(\nu)\big) \leq R$. Then,*

$$\mathbb{E}\big[\big|\underline{\mathsf{W}}_p^p(\hat{\mu}_n, \hat{\nu}_n) - \underline{\mathsf{W}}_p^p(\mu, \nu)\big|\big] \lesssim_{p,R} n^{-1/2} \quad and \quad \mathbb{E}\big[\big|\overline{\mathsf{W}}_p^p(\hat{\mu}_n, \hat{\nu}_n) - \overline{\mathsf{W}}_p^p(\mu, \nu)\big|\big] \lesssim_{p,R} d n^{-1/2}.$$

*If further $\mu \neq \nu$, then the same (parametric) rate also holds for empirical $\underline{\mathsf{W}}_p$ and $\overline{\mathsf{W}}_p$.*

Proposition 1 is proven in Appendix D.2 using a comparison inequality between $\mathsf{W}_p$ and $\mathsf{W}_1$ and elementary bounds for $\mathsf{W}_1$ using its integral representation and KR duality.

**Remark 4** (Comparison to [37]). *Theorem 2 of [37] establishes a bound of $(\log n/n)^{1/2}$ on the two-sample average-sliced Wasserstein distance, but under bounded moment assumptions instead of compact support.*

# 4 Robust Estimation

We examine robustness of sliced Wasserstein distances to outliers, showing that slicing enables dimension-free risk bounds that avoid $\mathrm{poly}(d)$ factors present for classic $\mathsf{W}_p$ (cf. [46]). We consider TV corruptions, where an unknown "clean" distribution $\mu$ is contaminated to obtain $\tilde{\mu}$ with $\|\mu - \tilde{\mu}\|_{\mathrm{TV}} \leq \epsilon$. Upon observing $\tilde{\mu}$, the goal is to return a distribution $T(\tilde{\mu})$ such that the error $\mathsf{D}\big(T(\tilde{\mu}), \mu\big)$ is small, where $\mathsf{D} \in \{\underline{\mathsf{W}}_p, \overline{\mathsf{W}}_p\}$. Without further assumptions, this error can be unbounded, so we require that $\mu$ belongs to a family $\mathcal{G} \subset \mathcal{P}(\mathbb{R}^d)$ encoding standard moment bounds. We consider the minimax risk for robust estimation under $\mathsf{D}$ with TV contamination, defined by

$$R(\mathsf{D}, \mathcal{G}, \epsilon) = \inf_{T:\mathcal{P}(\mathbb{R}^d) \to \mathcal{P}(\mathbb{R}^d)} \sup_{(\mu, \tilde{\mu}) \in \mathcal{G} \times \mathcal{P}(\mathbb{R}^d); \, \|\tilde{\mu} - \mu\|_{\mathrm{TV}} \leq \epsilon} \mathsf{D}\big(T(\tilde{\mu}), \mu\big).$$

Fix $q > p$ and let $\mathcal{G}_q(\sigma) := \big\{\mu \in \mathcal{P}_q(\mathbb{R}^d) : \sup_{\theta \in \mathbb{S}^{d-1}} \mu|\theta^{\mathsf{T}}(x - \mu x)|^q \leq \sigma^q\big\}$ contain all distributions whose projections have bounded central $q$th moments. In particular, $\mathcal{G}_2(\sigma) = \{\mu \in \mathcal{P}_2(\mathbb{R}^d) : \|\Sigma_\mu\|_{\mathrm{op}} \leq \sigma\}$. The next theorem characterizes minimax robust estimation risk over this class.

**Theorem 2** (Population-limit robust estimation). *Fix $1 \leq p < q$, $\sigma \geq 0$, and $0 \leq \epsilon \leq 0.49$.[4] We have $R\big(\underline{\mathsf{W}}_p, \mathcal{G}_q(\sigma), \epsilon\big) \asymp \sigma\sqrt{(1 \vee d/q)(1 \wedge p/d)}\,\epsilon^{1/p-1/q} \quad and \quad R\big(\overline{\mathsf{W}}_p, \mathcal{G}_q(\sigma), \epsilon\big) \asymp \sigma\epsilon^{1/p-1/q}.$*

Note that the $\sqrt{(1 \vee d/q)(1 \wedge p/d)}$ prefactor in the first bound is always less than 1. The proof in Appendix D.4 controls the risk via $\sup_{\mu,\nu \in \mathcal{G}_q(\sigma), \|\mu-\nu\|_{\mathrm{TV}} \leq \epsilon} \mathsf{D}(\mu, \nu)$, a modulus of continuity that captures the sensitivity of $\mathsf{D}$ to small perturbations that preserve membership to the clean family. We employ techniques based on generalized resilience [61, 55] to relate this modulus to similar quantities arising in the robust estimation of $p$th moment tensors, giving the above rates. The procedure that achieves these rates projects the observed contaminated distribution onto the corresponding family of clean distributions in TV norm.

**Remark 5** (Comparison to [46]). *A related framework [46] considers robust estimation of $\mathsf{W}_p$ under input measure contamination. They obtain a rate of $\sigma\sqrt{d}\epsilon^{1/p-1/2}$ using similar methods under the weaker Huber $\epsilon$-contamination model when $q = 2$ (see Corollary 1 therein). Evidently, slicing eliminates a $\sqrt{d}$ factor from the minimax estimation risk. In Appendix D.4, we interpolate between these regimes, proving that $k$-dimensional sliced distances admit risks bounded by $\sigma\sqrt{1 \vee k/q}\,\epsilon^{1/p-1/q}$.*

Theorem 2 characterizes population-limit robust estimation, i.e., when data is abundant. The next result, proven in Appendix D.5, extends to the finite-sample regime. For a radius $R > 0$, we write $\mu_R$ to denote the distribution of $X \sim \mu$ conditioned on $\|X - \mu x\| \leq R$.

**Proposition 2** (Finite-sample robust estimation). *Fix $1 \leq p < q, \sigma \geq 0$, and $0 < \epsilon \leq 0.49$, and let $\mathsf{D} \in \{\underline{\mathsf{W}}_p, \overline{\mathsf{W}}_p\}$. Then there exists a radius $R \asymp \sqrt{d/\epsilon}$ and a procedure which, given $n \geq (R^p + \epsilon^{-2})\,d\log(d/\epsilon)$ samples with at least $(1-\epsilon)n$ drawn i.i.d. from any $\mu \in \mathcal{G}_q(\sigma)$, returns $\nu \in \mathcal{P}(\mathbb{R}^d)$ such that $\mathsf{D}(\nu, \mu) \lesssim R(\mathsf{D}, \mathcal{G}_q(\sigma), \epsilon) + \mathbb{E}\big[\mathsf{D}\big((\hat{\mu}_R)_n, \mu_R\big)\big]$ with probability at least 0.99[5].*

---

[4]The upper bound on $\epsilon$ of 0.49 can be substituted with an any constant bounded away from 1/2.

[5]See Appendix D.5 for precise high-probability bounds and extension to the strong contamination model.

Evidently, the finite-sample error bound comprises the population-limit robust estimation risk (which is necessary) plus the empirical estimation error associated with the truncated distribution $\mu_R$. The lower bounds on $n$ ensures that the empirical distribution $(\widehat{\mu}_R)_n$ satisfies the same generalized resilience property appearing in the population-limit analysis. The truncated empirical convergence term can typically be bounded by the corresponding untruncated version. For example, when $\mu \in \mathcal{G}_2(\sigma)$ is log-concave and $\mathsf{D} = \overline{\mathsf{W}}_1$, we can bound this term by $O\big(\sigma d \log n / n + \sigma \sqrt{d \log n / n}\big)$, which follows from Theorem 1 and the fact that $\mu_R$ is also log-concave with $\|\Sigma_{\mu_R}\|_{\mathrm{op}} \leq \|\Sigma_{\mu}\|_{\mathrm{op}} \leq \sigma$ for any $R > 0$.

When $p = 1$, we prove in Appendix D.6 a precise connection to resilience, a sufficient condition for robust mean estimation, which may be of independent interest.

**Proposition 3** (Connection to mean resilience). *For $0 \leq \epsilon < 1$, $\mu \in \mathcal{P}_1(\mathbb{R}^d)$ is $(\rho, \epsilon)$-resilient, i.e. $\|\mu x - \nu x\| \leq \rho$ for all $\nu \leq \frac{1}{1-\epsilon}\mu$, if and only if $\overline{\mathsf{W}}_1(\mu, \nu) \leq \Theta(\rho)$ for all $\nu \leq \frac{1}{1-\epsilon}\mu$.*

This suggests borrowing from the existing family of robust mean estimation algorithms, primarily developed for the bounded covariance setting ($q = 2$). In Appendix D.7, we inspect an efficient spectral reweighting procedure and apply it for both $\overline{\mathsf{W}}_p$ and $\underline{\mathsf{W}}_p$ when $1 \leq p < 2$.

**Proposition 4** (Efficient computation via spectral reweighting). *If $1 \leq p < q = 2$ and $0 \leq \epsilon \leq 1/12$, the guarantee of Proposition 2 is achieved by an $\widetilde{O}(nd^2)$-time spectral reweighting algorithm.*

## 5 Formal Computational Guarantees

The computational tractability of empirical sliced Wasserstein distances relies on the simplified expressions for $\mathsf{W}_p$ between distribution on $\mathbb{R}$. However, even then, evaluating $\underline{\mathsf{W}}_p$ and $\overline{\mathsf{W}}_p$ requires computing the average or the maximum of one-dimensional distances over projection directions $\theta \in \mathbb{S}^{d-1}$. This section provides formal guarantees for two such popular computational methods: MC integration for $\underline{\mathsf{W}}_p$ and alternating subgradient-based optimization for $\overline{\mathsf{W}}_p$. Our analysis relies on the observation that $w_p(\theta) := \mathsf{W}_p(\mathfrak{p}_\sharp^\theta \mu, \mathfrak{p}_\sharp^\theta \nu)$ and its $p$th power are Lipschitz functions on $\mathbb{S}^{d-1}$.

**Lemma 1** (Lipschitz continuity). *The functions $w_p$ and $w_p^p$ are Lipschitz with constants bounded by $L_{\mu,\nu}^p = \sup_{\theta \in \mathbb{S}^{d-1}} \big[ (\mu|\theta^\mathsf{T}x|^p)^{1/p} + (\nu|\theta^\mathsf{T}x|^p)^{1/p} \big]$ and $M_{\mu,\nu}^p = 3p2^p \sup_{\theta \in \mathbb{S}^{d-1}} (\mu|\theta^\mathsf{T}x|^p + \nu|\theta^\mathsf{T}x|^p)$, respectively.*

Lemma 1 (proven in Appendix D.8) sharpens the Lipschitz constants derived in [47, Lemma 2], which correspond to bounding $|\theta^\mathsf{T}x|$ by $\|x\|$ in the above expressions. The projected moments $(\mu|\theta^\mathsf{T}x|^p)^{1/p}$ typically has a milder dependence on $d$ than $(\mu\|x\|^p)^{1/p}$, which is crucial for the subsequent analysis.

### 5.1 Average-Slicing: Monte Carlo Integration

The typical approach for computing the integral over the unit sphere in $\underline{\mathsf{W}}_p$ is MC averaging. Fix $\mu, \nu \in \mathcal{P}_p(\mathbb{R}^d)$ and let $\hat{\mu}_n$ and $\hat{\nu}_n$ be the associated empirical measures. Take $\Theta \sim \mathrm{Unif}(\mathbb{S}^{d-1})$ and consider i.i.d. copies thereof $\Theta_1, \ldots, \Theta_m$. The MC based estimate of $\underline{\mathsf{W}}_p^p$ is given by

$$\widehat{\underline{\mathsf{W}}}_{\mathsf{MC}}^p := \frac{1}{m}\sum_{j=1}^m \mathsf{W}_p^p(\mathfrak{p}_\sharp^{\Theta_j}\hat{\mu}_n, \mathfrak{p}_\sharp^{\Theta_j}\hat{\nu}_n) = \frac{1}{mn}\sum_{j=1}^m\sum_{i=1}^n \big|X_{(i)}(\Theta_j) - Y_{(i)}(\Theta_j)\big|^p,$$

where $X_{(1)}(\theta) \leq \cdots \leq X_{(n)}(\theta)$ is the order statistics, which is readily evaluated using sorting algorithms with $O(n \log n)$ average/worst-case complexity (e.g., `quick_sort` or `merge_sort`).

The next result bounds the effective error of $\widehat{\underline{\mathsf{W}}}_{\mathsf{MC}}^p$ in approximating the population distance $\underline{\mathsf{W}}_p^p(\mu, \nu)$.

**Proposition 5** (Monte Carlo error bound). *Let $1 \leq p < \infty$, and assume $\mu, \nu \in \mathcal{P}_p(\mathbb{R}^d)$ are log-concave with covariance matrices $\Sigma_\mu$ and $\Sigma_\nu$, respectively. The MC estimate above satisfies*

$$\mathbb{E}\Big[\big|\widehat{\underline{\mathsf{W}}}_{\mathsf{MC}}^p - \underline{\mathsf{W}}_p^p(\mu, \nu)\big|\Big] \lesssim_p \frac{\|\mu x - \nu x\|^p + \|\Sigma_\mu\|_{\mathrm{op}}^{p/2} + \|\Sigma_\nu\|_{\mathrm{op}}^{p/2}}{\sqrt{md}} + \frac{\big(\|\Sigma_\nu\|_{\mathrm{op}}^{p/2} + \|\Sigma_\mu\|_{\mathrm{op}}^{p/2}\big)(\log n)^{\mathbb{1}_{\{p=2\}}}}{n^{(p \wedge 2)/2}}$$

*where the hidden constant depends only on $p$.*

---

**Algorithm 1** Projected subgradient method for $\tilde{w}_2^2$

---

    **Input:** $\theta_0 \in \mathbb{B}^d$, a sequence $\{\alpha_t\}_{t \geq 0} \subset \mathbb{R}_+$, and iteration count $T$
    **for** $t = 0, \ldots, T$ **do**
        Calculate $\xi_t \in \partial \tilde{w}_2^2(\theta_t)$
        Set $x_{t+1} = \text{Proj}_{\mathbb{B}^d}(x_t - \alpha_t \xi_t)$
    Sample $t^* \in \{0, \ldots, T\}$ according to the probability distribution $\mathbb{P}(t^* = t) = \frac{\alpha_t}{\sum_{t=0}^{T} \alpha_t}$.
    **Return** $x_{t^*}$

---

Proposition 5 is proven in Appendix D.9 by separately bounding the MC and the empirical approximation errors. For the former, we use the Lipschitzness of $w_p^p$ on $\mathbb{S}^{d-1}$ to show that it concentrates about its median. This enables controlling the variance of $\frac{1}{m} \sum_{j=1}^{m} w_p^p(\Theta_j)$, which, in turn, bounds the MC error. For the empirical approximation error, we reduce the analysis to one-sample empirical convergence under $\mathsf{W}_p^p$ for measures on $\mathbb{R}$ and obtain explicit rates by drawing upon the results of [8].

**Remark 6** (Comparison to [42]). *Error bounds for the MC estimate $\widehat{\mathsf{W}}_{\mathsf{MC}}$ were also provided in [42], but their results differ from ours in two key ways: they use implicit empirical approximation bounds and leave their MC error in terms of* $\text{Var}\big(w_p^p(\Theta)\big)$ *without further analysis. Proposition 5 provides an explicit convergence rates and bounds the said variance in terms of basic characteristics of the population distributions, providing precise rates in* $n, m, d,$ *and* $p$.

**Remark 7** (Blessing of dimensionality). *A cruder approximation of the Lipschitz constant that stems from [47, Lemma 2] would yield* $\mu \|x\|^p + \nu \|x\|^p$ *as the numerator of the first term. However, such a bound can have a significantly worse dimension dependence. Indeed, if, for instance,* $\mu$ *and* $\nu$ *are both mean zero log-concave with identity covariance matrices, then* $\mu \|x\|^p + \nu \|x\|^p$ *is* $O_d(d^{p/2})$ *while the numerator in our bound is* $O_d(1)$. *For such* $\mu$ *and* $\nu$, *the bound decays to 0 as* $d \to \infty$.

## 5.2 Max-Slicing: Subgradient Methods and the LIPO Algorithm

Maximization of projected Wasserstein distance is a non-convex and non-smooth optimization problem. Therefore, past works that studied $k$-dimensional subspace projections relied on convex relaxations [48] or entropic regularization [34, 25] to prove approximate convergence to a stationary point. We show that regularization is not needed in the one-dimensional case of $\overline{\mathsf{W}}_p$ by proving an $O(\epsilon^{-4})$ computational complexity bound for convergence to stationarity of the simple subgradient-based optimization routine from [28, 17]. We also note that global solutions are attainable via generic algorithms for optimizing Lipschitz functions, but with rates that deteriorate exponentially with $d$.

**Local guarantees for subgradient methods.** First note that we may relax the $\overline{\mathsf{W}}_p$ optimization domain from $\mathbb{S}^{d-1}$ to the unit ball $\mathbb{B}^d$ without changing the value (indeed, for any $\theta \in \mathbb{B}^d$, $w_p(\theta) = \|\theta\| w_p(\theta/\|\theta\|)$). Together with [8, Lemma 4.2], we express the empirical max-sliced distance as:

$$\overline{\mathsf{W}}_p^p(\hat{\mu}_n, \hat{\nu}_n) = \max_{\theta \in \mathbb{B}^d} \min_{\pi \in \Pi(\hat{\mu}_n, \hat{\nu}_n)} \mathbb{E}_\pi \left[|\theta^\mathsf{T}(X - Y)|^p\right] = -\min_{\theta \in \mathbb{B}^d} \max_{\sigma \in S_n} \left(-\frac{1}{n} \sum_{i=1}^{n} |\theta^\mathsf{T}(X_i - Y_{\sigma(i)})|^p\right),$$

where $S_n$ is the symmetric group. Here we used the fact that the optimal coupling is given by the order statistics, and hence it suffices to optimize over permutations. Denote $\rho(\sigma, \theta) := -\frac{1}{n} \sum_{i=1}^{n} |\theta^\mathsf{T}(X_i - Y_{\sigma(i)})|^p$ and $\tilde{w}_p^p(\theta) := \max_{\sigma \in S_n} \rho(\sigma, \theta)$. The subgradient of $\tilde{w}_p^p$ has the closed form $\partial \tilde{w}_p^p(\theta) = \text{Conv}\left(\{\partial_\theta \rho(\sigma^*, \theta) : \sigma^* \in \text{argmax}_{\sigma \in S_n} \hat{\rho}(\sigma, \theta)\}\right)$. We can compute an optimal $\sigma^* \in S_n$ via order statistics and evaluate the corresponding subgradient vector in $\partial_\theta \rho(\sigma^*, \theta)$. This gives direct access to subgradients of $\tilde{w}_p^p$ without approximation arguments or regularization.

A heuristic description of Algorithm 1 was given in [28, 17], but without formal guarantees. Proposition 6 below can be viewed as closing that gap by providing said guarantees. In particular, for $p = 2$ the objective function $\tilde{w}_2^2$ is weakly convex [34, Lemma 2.2] and Lipschitz (Lemma 1). Together with the computable subgradients, this enables applying the proximal stochastic subgradient method from [14]. Algorithm 1 describes the adaptation of this method to our problem, after replacing the stochastic subgradient sampling step therein with the direct subgradient calculation described above. The following proposition provides convergence guarantees for Algorithm 1.

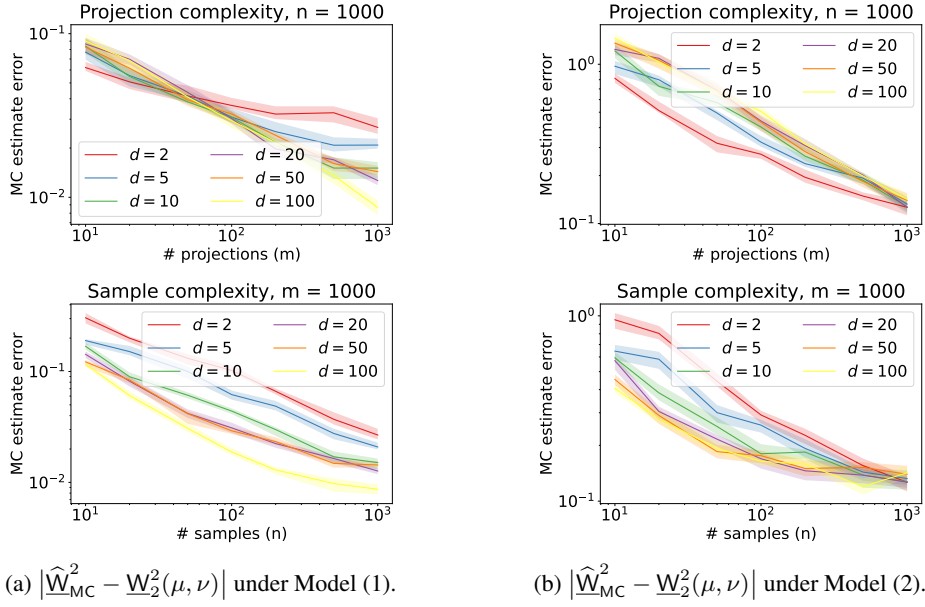

(a) $\left|\widehat{\underline{W}}_{MC}^2 - \underline{W}_2^2(\mu, \nu)\right|$ under Model (1).  (b) $\left|\widehat{\underline{W}}_{MC}^2 - \underline{W}_2^2(\mu, \nu)\right|$ under Model (2).

Figure 1: Projection and sample complexity for $\overline{W}_2$.

**Proposition 6** (Computational complexity of subgradient method). *Fix any $\epsilon > 0$ and $n, d \in \mathbb{N}$ such that $d \geq (\log n)^2$. Let $\mu, \nu \in \mathcal{P}_2(\mathbb{R}^d)$ be log-concave with covariance matrices $\Sigma_\mu$ and $\Sigma_\nu$, respectively, and consider $M_{\mu,\nu}^2$ as defined in Lemma 1. Then, there exist universal constants $c_1, c_2, c_3, c_4$ such the following holds: Algorithm 1 for the objective $\varphi(\theta) = \tilde{w}_2^2 + \delta_{\mathbb{B}^d}$, where $\delta_{\mathbb{B}^d} = -\infty \mathbb{1}_{(\mathbb{B}^d)^c}$, with step size $\alpha_t \propto \frac{1}{\sqrt{t+1}}$, outputs a point $\theta_{t^*}$ that is close to a near-stationary point $\theta^*$, in the sense that $\mathbb{E}_{t^*}[\|\theta^* - \theta_{t^*}\|] \leq \frac{\epsilon}{2\rho_{\mu,\nu}}$, for $\rho_{\mu,\nu} = \|\mu x - \nu x\|^2 + c_1 d (\|\Sigma_\mu\|_{op} + \|\Sigma_\nu\|_{op})$, and $\mathrm{dist}(0, \partial \tilde{w}_2^2(\theta^*)) \leq \epsilon$, within a number of computations $N \leq C_{\mu,\nu} \epsilon^{-4} n \log n$, where $C_{\mu,\nu} := c_2 \rho_{\mu,\nu}^2 (M_{\mu,\nu}^2 + c_3(\|\Sigma_\mu\|_{op} + \|\Sigma_\nu\|_{op}))^2$, with probability at least $1 - \frac{c_4}{n}$.*

Proposition 6 is proven in Appendix D.10 via the complexity bound from [14, Corollary 2]. As the algorithm is tuned for the empirical objective $\tilde{w}_p^p(\theta)$, the bound depends on the random Lipschitz and weak convexity constants $M_n = 4 \sup_{\theta \in \mathbb{S}^{d-1}} (\hat{\mu}_n|\theta^\intercal x|^2 + \hat{\nu}_n|\theta^\intercal x|^2)$ and $\rho_n = 2 \max_{i,j=1,\ldots,n} \|X_i - Y_j\|^2$. We use concentration bounds for $M_n$ and $\rho_n$ to obtain the deterministic bound above.

**Remark 8** (Comparison to past works). *Computation of projection-robust Wasserstein distances (i.e., when projections are $k$-dimensional) was studied in [48] and [34, 25] using a convex relaxations and entropic regularization, respectively. A similar $O(\epsilon^{-4})$ convergence rate is proven in [34] for their regularized method. Proposition 6 shows that regularization in not necessary to achieve this rate when projections are one-dimensional. The result of [34] was improved to $O(\epsilon^{-3})$ in [25] using Riemannian block coordinate descent (still with entropic regularization). While this rate is faster than in Proposition 6, our goal was to couple the simpler and abundantly used subgradient ascent approach with formal guarantees. In addition, the next section shows that empirically, our algorithm is much faster than those of [34, 25] for the $\overline{W}_2$ in terms of complexity and computation time.*

**Remark 9** (The non-quadratic case). *For $p \neq 2$, we still have Lipschitzness of the objective function in $\theta$ (Lemma 1). Recent work on finding stationary points for non-smooth, non-convex, Lipschitz functions, such as [15], provide convergence guarantees for these cases. These guarantees appear to be of the same $\epsilon^{-1/4}$ order (cf. [15, Theorem 3.2]), but we leave a full exploration for future work.*

**Remark 10** (Global guarantees via LIPO). *We can attain global optimality, i.e., compute $\overline{W}_p(\hat{\mu}_n, \hat{\nu}_n)$ itself, via the LIPO algorithm [36]. LIPO performs global optimization of Lipschitz functions over convex domains based on function evaluations, which are readily accessible in our problem via sorting. In Appendix C, we adapt LIPO to the max-sliced distance, prove consistency, and derive its complexity. While this approach attains global optimality, the number of evaluation grows exponentially with dimension. Hence, the subgradient method described above is preferable when dimension is large.*

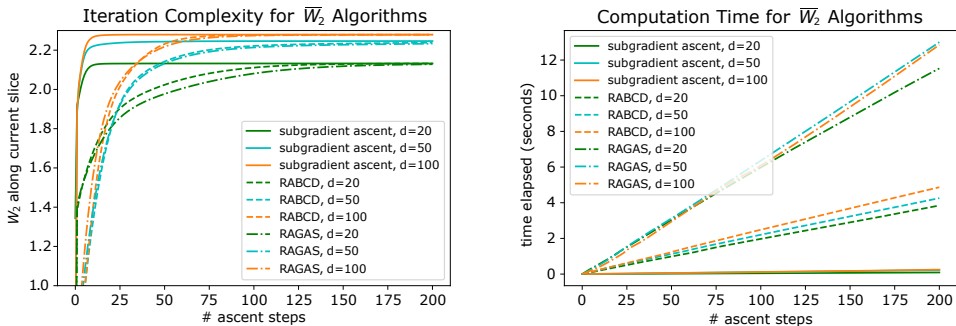

Figure 2: Errors and runtime versus step count for $\overline{W}_2$ computation algorithms.

## 6 Empirical Results

**Projection and sample complexity for $\underline{W}_2$.** We validate the convergence rates of the MC-based estimate of $\underline{W}_2$ predicted by Propositions 5 in the following two models: (1) $\mu = \mathcal{N}(0, I_d)$, $\nu = 0.5\mathcal{N}(0, I_d) + 0.5\mathcal{N}(0, I_d + 0.5\mathbf{1}_d\mathbf{1}_d^\mathsf{T}/d)$, and (2) $\mu = \mathcal{N}(0, I_d)$, $\nu = \mathcal{N}(2\mathbf{1}_d, I_d)$, where $\mathbf{1}_d$ is a vector with all coordinates equal to 1. For Model (1), Proposition 5 predicts a decreasing error with dimension, and inverse square root decay in number of projections and number of samples. For Model (2), on the other hand, the errors should increase with $d$ for sufficiently large $n$. Plots of the projection and sample complexities for each model (averaged over 100 runs) are given in Figure 1 at the top of the previous page, and are in line with the above discussion and our theoretical results. Additionally, confidence bands are plotted representing top and bottom 10% quantiles among 20 bootstrappped means from the same 100 runs. An additional experimental setup, comparing 10 component normal mixtures with different means and variances, can be found in Appendix E.

**Comparison of $\overline{W}_2$ algorithms.** We compare the performance of the subgradient-based Algorithm 1 and the Riemannian optimization methods of [34, 25]. Consider the setup from [25, Section 6.1], where $\mu = \mathrm{Unif}([-1, 1]^d)$ and $\nu = T_\sharp\mu$, with $T(x) = x + \sum_{i=1}^{10} \mathrm{sign}(x_i)e_i$, is the fragmented hypercube distribution with $k^* = 10$. Figure 2 shows the errors and runtime by step count of Algorithm 1 (with a constant step size) and the Riemannian algorithms from [34, 25] (abbreviated RAGAS and RABCD, respectively) for different ambient dimensions. For these algorithms, we used the code from `https://github.com/mhhuang95/PRW_RBCD` with their default choice of parameters; we also tried optimizing these parameters but the observed trends remained the same. Sample size is fixed at $n = 500$ and computation times are averaged over 10 trials. Evidently, the subgradient ascent algorithm converges significantly faster and within fewer iterations than the other two methods, for all considered values of $d$. Despite our $O(\epsilon^{-4})$ iteration complexity bound, which is slower than the best known rates [25], this favorable empirical performance may be attributed to the fact that Algorithm 1 relies on the cheap sorting operation, as opposed to the burdensome computation of regularization operations in [34, 25]. It may also be the case that our bound can be improved, which we plan to explore in future work.

**Robust estimation.** To support Proposition 3, we perform robust estimation via a standard iterative filtering algorithm developed for mean estimation [19]. For $d \in \{10, 20, \dots, 200\}$, we take $n = 10d\epsilon^{-2}$ samples, with $(1 - \epsilon)n$ drawn i.i.d. from $\mathcal{N}(0, I_d)$ and $\epsilon n$ from a product noise distribution used in [19], with $\epsilon = 0.1$. For each $d$, iterative filtering returns a candidate subset of clean samples, and Figure 3 (left) compares these subsets to the true clean samples both in $\overline{W}_1$ (estimated via projected subgradient ascent) and in $\ell_2$ distance between means. Note that the error in the latter never exceeds that in the former by more than a factor of 2 (Proposition 3 implies that mean and $\underline{W}_1$ risk are equivalent up to constant factors). In Figure 3 (right), we set $\mu = (1 - \epsilon)\delta_0 + \epsilon \, \mathrm{Unif}(\sqrt{d/\epsilon}\, \mathbb{S}^{d-1})$ with null contamination, and take $n, d$ ranging as before. In this case, since many samples are 0, we can efficiently compute a lower bound on classic $W_1$ between the filtered and clean samples in high dimensions. As predicted by Theorem 2, we observe the $\sqrt{d}$ separation in estimation error between $W_1$ and $\overline{W}_1$. In this case, only errors for the filtered samples are plotted, since the unfiltered samples have no contamination. See Appendix E for additional experiments on generative modeling with contaminated datasets, along with full details and code for all experiments.

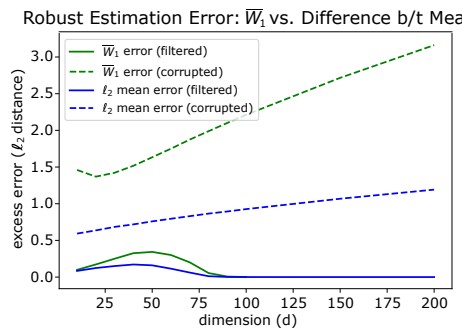 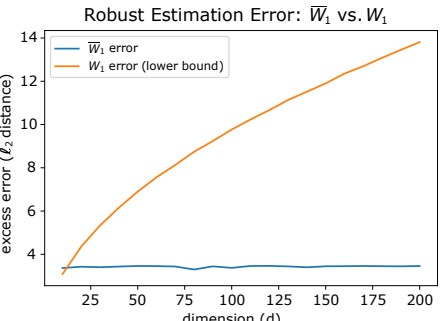

Figure 3: Robust estimation errors for the iterative filtering estimate in two scenarios: (left) comparing $\overline{W}_1$ with difference between means and (right) comparing $\overline{W}_1$ to $W_1$. Mean and $\overline{W}_1$ errors are bounded as $d \to \infty$, while $W_1$ error scales like $\sqrt{d}$.

# 7 Summary and Concluding Remarks

This paper provided a quantitative study of the scalability of sliced Wasserstein distances to high dimensions. Three key aspects were covered:

- **Empirical convergence rates:** We established sharp, dimension-free rates for $\underline{W}_p$ and $\overline{W}_p$ with explicitly characterized dimension-dependent constants. Our bounds reveal the interplay between the number of samples $n$, dimension $d$, and the order of the distance $p$.

- **Robust estimation:** The minimax optimal robust estimation rate of $\underline{W}_p$ and $\overline{W}_p$, under contamination level $\epsilon$, was characterized as $O(\sigma \epsilon^{1/p - 1/q})$. This rate is dimension-free and improves upon corresponding results for $W_p$ by a $\sqrt{d}$ factor. We showed that robust estimation of $\overline{W}_1$ is equivalent to robust mean estimation, which enables lifting statistical/algorithmic results from means to $\overline{W}_1$.

- **Computational guarantees:** The error of a MC-based estimator for $\underline{W}_p$ was derived, showing that it can improve as $d \to \infty$, depending on the growth-rate of the mean and the operator norm of the covariance matrix. For $\overline{W}_p$, we analyzed the subgradient-based local optimization algorithm from [28, 17], and proved $O(\epsilon^{-4})$ complexity using Lipschitzness and weak convexity of the objective.

In all three aspects, the benefit of slicing in terms of dependence on dimension was clearly evident, thus providing rigorous justification the perceived scalability of sliced distance. Going forward, we plan to pursue improved complexity bounds for the subgradient ascent algorithm for computing $\overline{W}_2$, as our empirical results suggest it converges faster than Proposition 6 predicts. We are also interested in understanding conditions on $\mu, \nu$ under which faster global guarantees for $\overline{W}_p$ can be provided, e.g., by precluding the existence of nontrivial local optima for $W_p(\mathfrak{p}_\sharp^\theta \hat{\mu}_n, \mathfrak{p}_\sharp^\theta \hat{\nu}_n)$ on $\mathbb{S}^{d-1}$, or matching the conditions of [36, Theorem 15], which results in polynomial and even exponential rates for LIPO. Extensions of our results to projection-robust Wasserstein distance, which considers projections to $k$-dimensional subspaces, are of interest, aiming to understand the effect of $k$ on the results.

## Acknowledgments and Disclosure of Funding

The authors thank Jason Gaitonde for helpful discussion on high-dimensional probability. S. Nietert was supported by a NSF Graduate Research Fellowship under Grant DGE-1650441. Z. Goldfeld is partially supported by NSF grants CCF-1947801, CCF-2046018, and DMS-2210368, and the 2020 IBM Academic Award. K. Kato is supported by NSF grants DMS-1952306, DMS-2014636, and DMS-2210368.

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
