# A  Cheeger Constant

Our empirical convergence rate analysis for the proof of Theorem 1 relies on controlling the Cheeger (isoperimetric) constant of the projected distributions. This section collects basic definitions and facts about Cheeger constants.

For $\mu \in \mathcal{P}(\mathbb{R}^d)$, define the boundary measure of a Borel subset $A \subset \mathbb{R}^d$ as

$$\mu^+(\partial A) := \liminf_{\epsilon \downarrow 0} \frac{\mu(A^\epsilon) - \mu(A)}{\epsilon},$$

where $A^\epsilon = \{x \in \mathbb{R}^d : d(x, A) \leq \epsilon\}$ is the $\epsilon$-blowup of $A$, with $d(x, A) := \inf\{\|x - y\| : y \in A\}$. The *Cheeger constant* $h(\mu)$ of $\mu$ is defined as

$$h(\mu) := \inf_{A \subset \mathbb{R}^d} \frac{\mu^+(\partial A)}{\min\{\mu(A), \mu(A^c)\}},$$

which serves as a measure of bottleneckedness for $\mu$. Indeed, a small $h(\mu)$ indicates the existence of a measurable $A \subset \mathbb{R}^d$ whose boundary measure is much smaller than the measure of $A$ and $A^c$ themselves. If $\mu$ has density $f$, then we also write $h(f) = h(\mu)$.

If $d = 1$ and $\mu$ has density $f$ with distribution function $F$, then the Cheeger constant admits the simplified expression [7, Theorem 1.3]

$$h(\mu) = \operatorname{essinf}_{x \in \mathbb{R}} \frac{f(x)}{\min\{F(x), 1 - F(x)\}}.$$

Furthermore, if $F$ is strictly increasing around $x$, then for $t = F(x)$, we have

$$\frac{f(x)}{\min\{F(x), 1 - F(x)\}} = \frac{f(F^{-1}(t))}{\min\{t, 1 - t\}}$$

The numerator on the right-hand side (RHS) is denoted by $I(t) := f(F^{-1}(t))$; lower bounding this function plays a key role in our empirical convergence rate analysis. The main observation in that regard is that if $f$ is log-concave in $d = 1$, then $\{x : 0 < F(x) < 1\}$ is an interval and $f$ is positive on the interval, which implies $I(t) \geq h(f) \min\{t, 1 - t\}$ for $t \in (0, 1)$.

Consequently, lower bounding $I(t)$ reduces to controlling $h(f)$ from below. In general, it is known from [26] that if $f$ is a log-concave density on $\mathbb{R}^d$ with covariance matrix $\Sigma$, then there exists a constant $c_d > 0$ that depends only on $d$ such that

$$h(f) \geq \frac{c_d}{\|\Sigma\|_{\mathrm{op}}^{1/2}}. \tag{3}$$

The KLS conjecture [26, 33] states that $c_d$ can be chosen to be independent of $d$. The best available result up to date is due to [13], which shows that $c_d = 1/d^{o_d(1)}$ as $d \to \infty$.

The proof of the concentration inequalities in Proposition 7 below requires another property of log-concave distributions, namely the fact that they satisfy Poincaré inequalities. A probability measure $\mu \in \mathcal{P}(\mathbb{R}^d)$ is said to satisfy a Poincaré inequality with constant $M_\mu > 0$ if

$$\operatorname{Var}_\mu(f) \leq M_\mu \mathbb{E}[\|\nabla f\|^2] \tag{4}$$

for any function $f : \mathbb{R}^d \to \mathbb{R}$ such that both sides of the above display are finite. The Maz'ya-Cheeger theorem (Theorem 1.1 in [39]) yields that $1/M_\mu \geq h(\mu)/2 > 0$, so that any (nondegenerate) log-concave distribution automatically satisfies a Poincaré inequality.

# B  Concentration Inequalities

We present concentration bounds for the empirical sliced distances as a corollary of Theorem 1. This result is utilized to provide global guarantees for computing $\overline{W}_p$ via the LIPO algorithm [36] (cf. Proposition 8 in Appendix C).

**Proposition 7** (Concentration inequalities). *Let $1 \leq p < \infty$ and $n \geq 2$, and assume that $\mu \in \mathcal{P}(\mathbb{R}^d)$ is log-concave with non-singular covariance matrix $\Sigma$. Then, for any $t > 0$,*

$$\mathbb{P}\left(\underline{\mathsf{W}}_p(\hat{\mu}_n, \mu) \geq \frac{\left(\|\Sigma\|_{\mathrm{op}}(\log n)^{\mathbb{1}_{\{p=2\}}}\right)^{1/2}}{n^{1/(2 \vee p)}} + t\right) \leq 2 \exp\left(-K \min\left\{n^{1/p}t, n^{2/(2 \vee p)}t^2\right\}\right), \tag{5a}$$

$$\mathbb{P}\left(\overline{\mathsf{W}}_p(\hat{\mu}_n, \mu) \geq \alpha_{n,\mu} + t\right) \leq 2 \exp\left(-K \min\left\{n^{1/p}t, n^{2/(2 \vee p)}t^2\right\}\right), \tag{5b}$$

*where $K \lesssim d^{o_d(1)} \max\{\|\Sigma\|_{\mathrm{op}}^{1/2}, \|\Sigma\|_{\mathrm{op}}\}$ and $\alpha_{n,\mu}$ is defined by the RHS of* (2b) *with $k = d$.*

The proof of Proposition 7 combines the expectation bounds from Theorem 1 with the concentration inequality for empirical sliced Wasserstein distances from [35, Theorem 3.8]. The latter result holds under a Poincaré inequality assumption on the population distribution, which is always satisfied for log-concave measures (cf. [39, Theorem 1.1]), and is hence applicable for our setting.

The proof proceeds by lower bounding the Poincaré constant of $\mu$, and then using a concentration result with expectation centering in [35] that relies on the Poincaré constant, combined with our expectation bounds (Theorem 1). By assumption, $\mu$ is log-concave with covariance matrix $\Sigma_\mu$. This, in particular, implies that (3) holds, with $c_d = d^{o_d(1)}$ (cf. Theorem 1 in [13]). Combined with the Maz'ya-Cheeger inequality (Theorem 1.1 in [39]), this gives the following bound for the Poincaré constant $M_\mu$ of $\mu$:

$$\frac{1}{M_\mu} \geq \frac{h(\mu)}{2} \geq \frac{1}{2d^{o_d(1)}\|\Sigma\|_{\mathrm{op}}}.$$

Now, by Theorem 3.8 in [35], we have

$$\mathbb{P}\left(|\rho(\hat{\mu}_n, \mu) - \mathbb{E}[\rho(\hat{\mu}_n, \mu)]| \geq t\right) \leq 2 \exp\left(-K \min\left\{n^{1/p}t, n^{2/(2 \vee p)}t^2\right\}\right), \quad t > 0,$$

where $\rho = \underline{\mathsf{W}}_p$ or $\overline{\mathsf{W}}_p$, and $K$ depends only on $M_\mu$. A careful review of the proof of Theorem 3.8 and intermediate results in [31] yields that $1/\min\{2\sqrt{M_\mu}, 6e^5 M_\mu\}$ is a valid choice of $K$ in the above display, so that $K \lesssim d^{o_d(1)} \max\{\|\Sigma\|_{\mathrm{op}}^{1/2}, \|\Sigma\|_{\mathrm{op}}\}$. Plugging (2a) and (2b) into the above display completes the proof.

## C  Global Guarantees for Max-Sliced $\mathsf{W}_p$ Computation via LIPO

We can compute $\overline{\mathsf{W}}_p(\hat{\mu}_n, \hat{\nu}_n) = \max_{\theta \in \mathbb{B}^d} \mathsf{W}_p(\mathfrak{p}_\sharp^\theta \hat{\mu}_n, \mathfrak{p}_\sharp^\theta \hat{\nu}_n)$ itself via the LIPO algorithm [36], which performs global optimization of Lipschitz functions over convex domains based on function evaluations. LIPO sequentially chooses the next evaluation point only if it can increase the function value, based on the Lipschitz condition. Setting $\hat{w}_p(\theta) := \mathsf{W}_p(\mathfrak{p}_\sharp^\theta \hat{\mu}_n, \mathfrak{p}_\sharp^\theta \hat{\nu}_n)$ and tuning LIPO to the (empirical) Lipschitz constant $\hat{L}_n := \sup_{\theta \in \mathbb{S}^{d-1}} \left[(\hat{\mu}_n|\theta^\intercal x|^p)^{1/p} + (\hat{\nu}_n|\theta^\intercal x|^p)^{1/p}\right]$ (see Lemma 1), if $\Theta_1, \ldots, \Theta_t$ are the $t$ previous evaluation points, the next evaluation will be at $\Theta_{t+1}$ provided that

$$\min_{1 \leq i \leq t}\left\{\hat{w}_p(\Theta_i) + \hat{L}_n^p\|\Theta_{t+1} - \Theta_i\|\right\} \geq \max_{1 \leq i \leq t}\hat{w}_p(\Theta_i).$$

The output after $k$ steps is $\max_{1 \leq i \leq k}\hat{w}_p(\Theta_i)$. See [36, Figure 1] for the full pseudo-algorithm. We have the following global guarantee for the performance of LIPO.

**Proposition 8** (LIPO error bound). *Let $1 \leq p < \infty$ and assume that $\mu, \nu \in \mathcal{P}_p(\mathbb{R}^d)$ are log-concave with non-singular covariance matrices $\Sigma_\mu$ and $\Sigma_\nu$, respectively. Let $\Theta_1, \ldots \Theta_k$ be a sequence of points generated by the LIPO for computing $\max_{\theta \in \mathbb{S}^{d-1}}\hat{w}_p(\theta)$. Then for any $t > 0$ and $n \geq C_p d^{p/2}$, we have*

$$\mathbb{P}\left(\left|\overline{\mathsf{W}}_p(\mu, \nu) - \max_{1 \leq i \leq k}\hat{w}_p(\Theta_i)\right| \leq 2L_{\mu,\nu}\left(\frac{\log(1/\delta)}{k}\right)^{1/d} + \alpha_n + 2t\right) \geq 1 - \delta - \beta - \gamma_n(t)$$

*where $\alpha_n = \alpha_{n,\mu} + \alpha_{n,\nu}$ with $\alpha_{n,\mu}$ given by the RHS of* (2b) *with $k = d$, $\alpha_{n,\nu}$ defined analogously,*

$$L_{\mu,\nu} = (\|\Sigma_\mu\|_{\mathrm{op}}^{1/2} + \|\Sigma_\nu\|_{\mathrm{op}}^{1/2})\left((2p)^{1/p} \vee 2 + \frac{1}{2}\right) + \|\mu x\| + \|\nu x\|,$$

$$\beta = \exp(-c_p \sqrt{d}), \quad and$$

$$\gamma_n(t) = 2\exp\left(-K_\mu \min\left(n^{1/p}t, n^{2/(2\vee p)}t^2\right)\right) + 2\exp\left(-K_\nu \min\left(n^{1/p}t, n^{2/(2\vee p)}t^2\right)\right),$$

*with $K_\mu \lesssim d^{o_d(1)} \max\{\|\Sigma_\mu\|_{\mathrm{op}}^{1/2}, \|\Sigma_\mu\|_{\mathrm{op}}\}$ and $K_\nu \lesssim d^{o_d(1)} \max\{\|\Sigma_\nu\|_{\mathrm{op}}^{1/2}, \|\Sigma_\nu\|_{\mathrm{op}}\}$.*

The proof of Proposition 8 is given in Appendix D.11. The analysis separately bounds the empirical approximation error of the max-sliced objective and the error due to LIPO. The empirical error is treated using the concentration inequality from Proposition 7. For the LIPO analysis, we first argue that the (random) Lipschitz constant $\hat{L}_n^p$ concentrated about its mean and bound the latter by the population Lipschitz constant $L_{\mu,\nu}^p$. With this deterministic bound, the result follows from [36, Corollary 13]. Evidently, while Proposition 8 provides a global optimality guarantee, the resulting rate depends exponentially on dimension, which is too conservative in high-dimensional settings.

# D  Proofs of Results in the Main Text

**Additional notation**: We use $N(\epsilon, \mathcal{F}, d)$ to denote the $\epsilon$-covering number of a function class or set $\mathcal{F}$ with respect to (w.r.t.) a metric $d$ on $\mathcal{F}$, and $N_{[]}(\epsilon, \mathcal{F}, d)$ denotes the corresponding bracketing number.

## D.1  Proof of Theorem 1

The proof relies on [8, Theorem 6.6], restated below, that bounds empirical convergence rates for $\mathsf{W}_p$ between distributions on $\mathbb{R}$.

**Lemma 2** (Theorem 6.6 in [8]). *Fix $1 \leq p < \infty$ and $n \geq 2$. Let $\mu \in \mathcal{P}(\mathbb{R})$ have log-concave density $f$ with distribution function $F$. Set $I(t) = f\big(F^{-1}(t)\big)$ for $t \in (0,1)$, where $F^{-1}$ is the quantile function of $F$. Then,*

$$\mathbb{E}\big[\mathsf{W}_p^p(\hat{\mu}_n, \mu)\big] \leq \left(\frac{Cp^2}{n}\right)^{p/2} \int_{1/(n+1)}^{n/(n+1)} \frac{\big(t(1-t)\big)^{p/2}}{I^p(t)}\, dt, \tag{6}$$

*where $C$ is a universal constant.*

We will apply Lemma 2 to $\mathsf{W}_p(\mathfrak{p}_\sharp^\theta \hat{\mu}_n, \mathfrak{p}_\sharp^\theta \mu)$ and bound the corresponding $I$-function from below *uniformly* over the projection parameter $\theta \in \mathbb{S}^{d-1}$. Recall that the distribution function of $\mathfrak{p}_\sharp^\theta \mu$ is denoted by $F_\mu(\cdot; \theta)$, which we abbreviate as $F_\theta$ throughout this proof and denote the corresponding density by $f_\theta$. We first observe that since $\mu \in \mathcal{P}(\mathbb{R}^d)$ is log-concave, then so is $\mathfrak{p}_\sharp^\theta \mu$ for any $\theta \in \mathbb{S}^{d-1}$.

Let $h_\theta := h(\mathfrak{p}_\sharp^\theta \mu)$ denote the Cheeger constant of the projected distribution. From the discussion in Appendix A, we know that $1/\big(f_\theta\big(F_\theta^{-1}(t)\big)\big) \geq h_\theta \min\{t, 1-t\}$ for $t \in (0,1)$. Given that, the proof for the $\underline{\mathsf{W}}_p$ case is relatively straightforward from Lemma 2. Bounding $\mathbb{E}[\overline{\mathsf{W}}_p(\hat{\mu}_n, \mu)]$, however, requires extra work to treat the supremum over $\theta$ that appears inside the expectation.

**$\underline{\mathsf{W}}_p$ case.**  Suppose that $\theta \in \mathbb{S}^{d-1}$ is such that $\theta^\mathsf{T} \Sigma \theta = 0$. Then, $\mathfrak{p}_\sharp^\theta \mu$ degenerates to a point mass, so that $\mathsf{W}_p^p\big(\mathfrak{p}_\sharp^\theta \hat{\mu}_n, \mathfrak{p}_\sharp^\theta \mu\big) = 0$.

Suppose $\theta^\mathsf{T} \Sigma \theta > 0$. Then, $\mathfrak{p}_\sharp^\theta \mu$ is nondegerate log-concave, so it has a log-concave density. Observe that $h_\theta \gtrsim 1/(\theta^\mathsf{T} \Sigma \theta)^{1/2}$. If $1 \leq p < 2$, then

$$\int_0^1 \frac{\big(t(1-t)\big)^{p/2}}{t^p \wedge (1-t)^p}\, dt < \infty,$$

so that by Lemma 2, we have

$$\mathbb{E}\big[\mathsf{W}_p^p\big(\mathfrak{p}_\sharp^\theta \hat{\mu}_n, \mathfrak{p}_\sharp^\theta \mu\big)\big] \lesssim \left(\frac{\theta^\mathsf{T} \Sigma \theta}{n}\right)^{p/2}.$$

If $2 \le p < \infty$, then by Lemma 2, dividing $\int_{1/(n+1)}^{n/(n+1)}$ into $\int_{1/(n+1)}^{1/2} + \int_{1/2}^{n/(n+1)}$ and using the symmetry, we have

$$\mathbb{E}\big[\mathsf{W}_p^p(\mathfrak{p}_\sharp^\theta \hat{\mu}_n, \mathfrak{p}_\sharp^\theta \mu)\big] \lesssim \left(\frac{\theta^\mathsf{T}\Sigma\theta}{n}\right)^{p/2} \int_{1/(n+1)}^{1/2} t^{-p/2}\, dt.$$

Here

$$\int_{1/(n+1)}^{1/2} t^{-p/2}\, dt = \begin{cases} \big[\log(n+1) - \log 2\big] & p = 2 \\ \frac{1}{p/2-1}\big[(n+1)^{p/2-1} - 2^{p/2-1}\big] & p > 2 \end{cases},$$

so that

$$\mathbb{E}\big[\mathsf{W}_p^p(\mathfrak{p}_\sharp^\theta \hat{\mu}_n, \mathfrak{p}_\sharp^\theta \mu)\big] \lesssim \frac{(\theta^\mathsf{T}\Sigma\theta)^{p/2}(\log n)^{\mathbb{1}\{p=2\}}}{n}.$$

The result follows by noting that $\theta^\mathsf{T}\Sigma\theta \le \|\Sigma\|_{\mathrm{op}}$, integrating the display over $\theta \in \mathbb{S}^{d-1}$ and applying Fubini's theorem.

**Remark 11** (Better bound for $\underline{\mathsf{W}}_2$). *The above calculation actually yields the slightly better bound*

$$\mathbb{E}[\underline{\mathsf{W}}_2^2(\mu, \nu)] \lesssim \frac{k\|\Sigma\|_{\mathrm{op}} \log n}{nd}$$

*for $p = 2$ by using the spectral decomposition $\Sigma = \sum_{i=1}^k \lambda_i a_i a_i^\mathsf{T}$, as follows:*

$$\int_{\mathbb{S}^{d-1}} \theta^\mathsf{T}\Sigma\theta\, d\sigma(\theta) = \sum_{i=1}^k \int_{\mathbb{S}^{d-1}} (a_i^\mathsf{T}\theta)^2\, d\sigma(\theta) = \frac{1}{d}\sum_{i=1}^k \lambda_i \le \frac{k\|\Sigma\|_{\mathrm{op}}}{d}.$$

$\overline{\mathsf{W}}_p$ **case.** We divide the proof into two steps. In Step 1, we will prove the claim of the theorem when $k = d$, i.e., $\Sigma$ is of full rank. In Step 2, we reduce the general case to the $d = k$ case.

Step 1. Assume $k = \mathrm{rank}(\Sigma) = d$. The main idea is to approximate $\mathbb{E}\big[\overline{\mathsf{W}}_p(\hat{\mu}_n, \mu)\big] = \mathbb{E}\big[\sup_{\theta \in \mathbb{S}^{d-1}} \mathsf{W}_p(\mathfrak{p}_\sharp^\theta \hat{\mu}_n, \mathfrak{p}_\sharp^\theta \mu)\big]$ by the maximum expected projected distance (roughly speaking, switch the expectation and the supremum). To that end we will employ a covering argument of the unit sphere along with Lipschitz continuity of $\mathsf{W}_p(\mathfrak{p}_\sharp^\theta \hat{\mu}_n, \mathfrak{p}_\sharp^\theta \mu)$ w.r.t. the samples and $\theta$. These technical results are collected in the following lemma.

**Lemma 3.** *The following hold:*

(i) *For any $\epsilon \in (0, 1)$, we have $N(\epsilon, \mathbb{S}^{d-1}, \|\cdot\|) \le (5/\epsilon)^d$.*

(ii) *For any $\gamma \in \mathcal{P}_p(\mathbb{R})$, the map $u \mapsto \mathsf{W}_p(n^{-1}\sum_{i=1}^n \delta_{u_i}, \gamma)$ with $u = (u_1, \ldots, u_n)$ is $n^{-1/(2\vee p)}$-Lipschitz. Further, it is partially differentiable a.e. w.r.t. each $u_i$, and its partial derivative w.r.t. $u_i$ is bounded by $n^{-1/p}$*

(iii) *The map $\theta \mapsto \mathsf{W}_p(\mathfrak{p}_\sharp^\theta \hat{\mu}_n, \mathfrak{p}_\sharp^\theta \mu)$ is $L$-Lipschitz with $L = (\sup_\theta \hat{\mu}_n|\theta^\mathsf{T}x|^p)^{1/p} + (\sup_\theta \mu|\theta^\mathsf{T}x|^p)^{1/p}$. If $\mu \in \mathcal{P}(\mathbb{R}^d)$ is centered and log-concave with non-singular covariance matrix $\Sigma$, then*

$$\mathbb{E}[L] \le 2p\|\Sigma\|_{\mathrm{op}}^{1/2}\sqrt{d}$$

*Proof of Lemma 3.* (i) Follows from an elementary volumetric argument, which is omitted for brevity.

(ii) By the triangle inequality and definition of $\mathsf{W}_p$, we have

$$\left|\mathsf{W}_p\left(n^{-1}\sum_{i=1}^n \delta_{u_i}, \gamma\right) - \mathsf{W}_p\left(n^{-1}\sum_{i=1}^n \delta_{u_i'}, \gamma\right)\right| \le \left(n^{-1}\sum_{i=1}^n |u_i - u_i'|^p\right)^{1/p}.$$

To bound the RHS by $n^{-1/(2\vee p)}\|u - u'\|$ we apply Jensen's inequality when $p \le 2$, and using the fact that $\sum_{i=1}^n a_i^{p/2} \le (\sum_{i=1}^n a_i)^{p/2}$ when $p \ge 2$.

The second statement follows from the fact that when coordinates other than $u_i$ are kept fixed, the RHS of the above display is bounded by $\|u_i - u_i'\|$.

(iii) A simpler version is proven in [47, Lemma 2], but we include the argument for completeness. Applying Lemma 1 with $\nu = \hat{\mu}_n$, we obtain Lipschitz continuity with constant $L := \left( \sup_\theta \mu |\theta^\mathsf{T} x|^p \right)^{1/p} + \left( \sup_\theta \hat{\mu}_n |\theta^\mathsf{T} x|^p \right)^{1/p} \leq (\mu \|x\|^p)^{1/p} + (\hat{\mu}_n \|x\|^p)^{1/p}$ so that $\mathbb{E}[L] \leq 2(\mu \|x\|^p)^{1/p}$

Since $\mu$ is centered and log-concave with covariance matrix $\Sigma$, in particular

$$(\mu \|x\|^p)^{1/p} \leq p(\mathbb{E}[\|X_1\|^2])^{1/2} = p\sqrt{\mathrm{tr}(\mathrm{Cov}(X_1))} \leq p\sqrt{d}\|\Sigma\|_{\mathrm{op}}^{1/2}.$$

See, for example, Remark 1 after Theorem 3.1 in [1]. □

We are ready to prove the empirical convergence rate of the max-sliced distance. For the remainder of the proof we will assume, without loss of generality, that $\mu$ has mean 0, since $\overline{\mathrm{W}}_p(t^v_\sharp \mu, t^v_\sharp \nu) = \overline{\mathrm{W}}_p(\mu, \nu)$ for any location shift $t^v : x \mapsto x + v$ and any probability measures $\mu$ and $\nu$, and any location shifted log-concave distribution is also log-concave with the same covariance matrix. Let $\tilde{w}_p(\theta) = \mathrm{W}_p(\mathfrak{p}^\theta_\sharp \hat{\mu}_n, \mathfrak{p}^\theta_\sharp \mu)$ and observe that

$$\mathbb{E}\left[\overline{\mathrm{W}}_p(\hat{\mu}_n, \mu)\right] \leq \sup_{\theta \in \mathbb{S}^{d-1}} \mathbb{E}[\tilde{w}_p(\theta)] + \mathbb{E}\left[ \sup_{\theta \in \mathbb{S}^{d-1}} \left( \tilde{w}_p(\theta) - \mathbb{E}[\tilde{w}_p(\theta)] \right) \right].$$

From the proof for the average-sliced case, we have

$$\sup_{\theta \in \mathbb{S}^{d-1}} \mathbb{E}[\tilde{w}_p(\theta)] \lesssim_p \frac{\left( \|\Sigma\|_{\mathrm{op}} (\log n)^{\mathbb{1}_{\{p=2\}}} \right)^{1/2}}{n^{1/(2 \vee p)}}. \tag{7}$$

Let $\theta_1, \ldots, \theta_{N_\epsilon}$ be a minimal $\epsilon$-net of $\mathbb{S}^{d-1}$, where $N_\epsilon = N(\epsilon, \mathbb{S}^{d-1}, \|\cdot\|)$. Using Lemma 3, we have

$$\mathbb{E}\left[ \sup_{\theta \in \mathbb{S}^{d-1}} \left( \tilde{w}_p(\theta) - \mathbb{E}[\tilde{w}_p(\theta)] \right) \right] \leq \inf_{\epsilon > 0} \mathbb{E}\left[ \max_{1 \leq j \leq N_\epsilon} \left( \tilde{w}_p(\theta_j) - \mathbb{E}[\tilde{w}_p(\theta_j)] \right) + 2\epsilon L \right], \tag{8}$$

where $L$ is a random variable with $\mathbb{E}[L] \leq c_p \|\Sigma\|_{\mathrm{op}}^{1/2} d^{1/2}$.

To control the maximum inside the expectation on the RHS of (8), we use an approach based on maximal inequalities for sub-exponential random variables, similar to Theorem 3.5 in [35]. Briefly, we will first show that for each $\theta$, $\tilde{w}_p(\theta)$ is a Lipschitz function of the projected observations $(\theta^\mathsf{T} X_1, \ldots, \theta^\mathsf{T} X_n)$, with bounded gradient in each coordinate, which will imply sub-exponential concentration for each $\tilde{w}_p(\theta)$. The term $\max_{1 \leq j \leq N_\epsilon} \left( \tilde{w}_p(\theta_j) - \mathbb{E}[\tilde{w}_p(\theta_j)] \right)$ will then be bounded via a maximal inequality as a direct consequence of this concentration (cf. Exercise 2.8 in [57]).

We will use the following refined concentration inequality for Lipschitz functions of random variables satisfying a Poincaré inequality, stated in [31]. Since explicit constants are not derived there, a proof is provided in Appendix D.3,

**Lemma 4** (Concentration from Poincaré inequality; Corollary 4.6 in [31]). *Let $\mu \in \mathcal{P}(\mathbb{R}^d)$ satisfy the Poincaré inequality (4) with constant $M_\mu$ and $f : \mathbb{R}^{nd} \to \mathbb{R}$ be $\alpha$-Lipschitz. For $x_1, \ldots, x_n \in \mathbb{R}^d$, define the functions*

$$f_i(\cdot | x_1, \ldots, x_{i-1}, x_{i+1}, \ldots, x_n) := f(x_1, \ldots, x_{i-1}, \cdot, x_{i+1}, \ldots, x_n), \quad i = 1, \ldots, n,$$

*and assume that $\max_{1 \leq i \leq n} \sup_{x \in \mathbb{R}^d} \|\nabla f_i(x | X_1, \ldots, X_{i-1}, X_{i+1}, \ldots, X_n)\| \leq \beta$ a.s. Then,*

$$\mu^{\otimes n}(f \geq \mu^{\otimes n} f + t) \leq \exp\left( -\min\left\{ \frac{t}{2\beta M_\mu^{1/2}}, \frac{t^2}{6e^5 \alpha^2 M_\mu} \right\} \right), \ t > 0.$$

The random vector $(\theta^\mathsf{T} X_1, \ldots, \theta^\mathsf{T} X_n)$ in $\mathbb{R}^n$ has i.i.d. coordinates with law $(\mathfrak{p}^\theta_\sharp \mu)^{\otimes n}$. The distribution $\mathfrak{p}^\theta_\sharp \mu$ is log-concave with variance $\theta^\mathsf{T} \Sigma \theta > 0$, which is bounded above by $\|\Sigma\|_{\mathrm{op}}$. This yields that $h_\theta := h(\mathfrak{p}^\theta_\sharp \mu) \gtrsim \|\Sigma\|_{\mathrm{op}}^{-1}$ for each $\theta \in \mathbb{S}^{d-1}$. By item (ii) in Lemma 3, the partial derivatives of $\tilde{w}_p(\theta)$ w.r.t. $\theta^\mathsf{T} X_i$, denoted $\nabla_i \tilde{w}_p(\theta)$, satisfy $\max_i \|\nabla_i \tilde{w}_p(\theta)\| \leq n^{-1/p}$ a.s., and $\tilde{w}_p(\theta)$ is

$n^{-1/(2\vee p)}$-Lipschitz in $(\theta^\intercal X_1, \ldots, \theta^\intercal X_n)$. By the Maz'ya-Cheeger Theorem (cf. Theorem 1.1 in [39]), $M_{\mathfrak{p}_\sharp^\theta \mu}^{-1} \geq h_\theta/2 \geq \|\Sigma_\mu\|_{\mathrm{op}}^{-1}/2$. Combining these facts and applying Lemma 4, we have

$$\mathbb{P}\big(\tilde{w}_p(\theta) - \mathbb{E}[\tilde{w}_p(\theta)] > t\big) \leq \exp\left(-\min\left(\frac{t}{\sqrt{2}n^{-1/p}\|\Sigma\|_{\mathrm{op}}^{1/2}}, \frac{t^2}{3e^5 n^{-2/(2\vee p)}\|\Sigma\|_{\mathrm{op}}}\right)\right)$$

$$\leq \exp\left(-\frac{t^2}{\sqrt{2}n^{-1/p}\|\Sigma\|_{\mathrm{op}}^{1/2}t + 3e^5 n^{-2/(2\vee p)}\|\Sigma\|_{\mathrm{op}}}\right), \quad t > 0.$$

A simple union bound then gives

$$\mathbb{P}\Big(\max_{1\leq j \leq N_\epsilon}\big(\tilde{w}_p(\theta_j) - \mathbb{E}[\tilde{w}_p(\theta_j)]\big) > t\Big) \leq N_\epsilon \exp\left(-\frac{t^2}{\sqrt{2}n^{-1/p}\|\Sigma\|_{\mathrm{op}}^{1/2}t + 3e^5 n^{-2/(2\vee p)}\|\Sigma\|_{\mathrm{op}}}\right),$$

which, by an expectation bound for sub-exponential random variables (cf. Exercise 2.8 in [57]), yields

$$\mathbb{E}\left[\max_{1\leq j \leq N_\epsilon}\big(\tilde{w}_p(\theta_j) - \mathbb{E}[\tilde{w}_p(\theta_j)]\big)\right]$$

$$\leq \sqrt{6e^5 n^{-2/(2\vee p)}\|\Sigma\|_{\mathrm{op}}}(\sqrt{\pi} + \sqrt{\log N_\epsilon}) + 2\sqrt{2}n^{-1/p}\|\Sigma\|_{\mathrm{op}}^{1/2}(1 + \log N_\epsilon)$$

$$\lesssim \|\Sigma\|_{\mathrm{op}}^{1/2} n^{-1/(2\vee p)}(1 + \sqrt{\log N_\epsilon}) + \|\Sigma\|_{\mathrm{op}}^{1/2} n^{-1/p}(1 + \log N_\epsilon). \tag{9}$$

By Lemma 3 (i), $\log N_\epsilon \leq d\log(5/\epsilon)$. Thus, setting $\epsilon = n^{-1/2}$ and plugging (9) into (8), we have

$$\mathbb{E}\big[\overline{\mathsf{W}}_p(\hat{\mu}_n, \mu)\big] \lesssim_p \|\Sigma\|_{\mathrm{op}}^{1/2}\left(\frac{(\log n)^{\mathbb{1}_{\{p=2\}}}}{n^{1/(2\vee p)}} + \frac{(1 + \sqrt{d\log n})}{n^{1/(2\vee p)}} + \frac{(1 + d\log n)}{n^{1/p}} + \frac{d^{1/2}}{n^{1/2}}\right).$$

The last term on the RHS of the above display is of smaller order in $n$ and $d$ than the other two terms. Further, for $n \geq 2$, we have $(1 + \sqrt{d\log n}) \lesssim \sqrt{d\log n}$ and $(1 + d\log n) \lesssim d\log n$. This leads to the bound stated in Theorem 1 when $k = d$.

Step 2. Suppose now that $1 \leq k < d$. Again assume without loss of generality that the mean of $\mu$ is zero. Observe that $\overline{\mathsf{W}}_p$ is invariant under common orthogonal transformations, i.e., for any $d \times d$ orthogonal matrix $Q$, $\overline{\mathsf{W}}_p(Q_\sharp \mu, Q_\sharp \nu) = \overline{\mathsf{W}}_p(\mu, \nu)$. With this in mind, we see that we may assume without loss of generality that $\Sigma$ is diagonal whose first $k$ diagonal entries are nonzero. Then, for $X = (X_1, \ldots, X_d)^\intercal \sim \mu$ and $\theta = (\theta^1, \ldots, \theta^d)^\intercal \in \mathbb{S}^{d-1}$, $\theta^\intercal X = \sum_{j=1}^k \theta^j X_j$ a.s. Thus, we have

$$\sup_{\theta \in \mathbb{S}^{d-1}} \mathsf{W}_p(\mathfrak{p}_\sharp^\theta \hat{\mu}_n, \mathfrak{p}_\sharp^\theta \mu) = \sup_{\substack{\theta = (\theta^1, \ldots, \theta^d)^\intercal \in \mathbb{S}^{d-1} \\ \theta^{k+1} = \cdots = \theta^d = 0}} \mathsf{W}_p(\mathfrak{p}_\sharp^\theta \hat{\mu}_n, \mathfrak{p}_\sharp^\theta \mu) \quad \text{a.s.}$$

The bound stated in Theorem 1 follows by the argument in Step 1 with $d$ replaced by $k$. $\qquad \square$

## D.2 Proof of Proposition 1

**Upper bound for $\underline{\mathsf{W}}_p^p$.** Let $F_\mu(t, \theta) = \mathbb{P}(\theta^\intercal X \leq t)$ for $X \sim \mu$, and analogously define $F_\nu(t, \theta)$. Then,

$$\mathbb{E}\big[\big|\underline{\mathsf{W}}_p^p(\hat{\mu}_n, \hat{\nu}_n) - \underline{\mathsf{W}}_p^p(\mu, \nu)\big|\big] \leq \mathbb{E}\left[\int_{\mathbb{S}^{d-1}}\left|\mathsf{W}_p^p(\mathfrak{p}_\sharp^\theta \hat{\mu}_n, \mathfrak{p}_\sharp^\theta \hat{\nu}_n) - \mathsf{W}_p^p(\mathfrak{p}_\sharp^\theta \mu, \mathfrak{p}_\sharp^\theta \nu)\right| d\sigma(\theta)\right]$$

$$\leq C_{p,R}\,\mathbb{E}\left[\int_{\mathbb{S}^{d-1}}\left(\mathsf{W}_1(\mathfrak{p}_\sharp^\theta \hat{\mu}_n, \mathfrak{p}_\sharp^\theta \mu) + \mathsf{W}_1(\mathfrak{p}_\sharp^\theta \hat{\nu}_n, \mathfrak{p}_\sharp^\theta \nu)\right) d\sigma(\theta)\right]$$

$$\leq \frac{C_{p,R}}{\sqrt{n}}\left(\int_{\mathbb{S}^{d-1}}\int_{-\infty}^{\infty}\sqrt{F_\mu(t, \theta)(1 - F_\mu(t, \theta))}\, dt\, d\sigma(\theta)\right.$$

$$\left. + \int_{\mathbb{S}^{d-1}}\int_{-\infty}^{\infty}\sqrt{F_\nu(t, \theta)(1 - F_\nu(t, \theta))}\, dt\, d\sigma(\theta)\right)$$

$$\leq \frac{RC_{p,R}}{\sqrt{n}},$$

where the second inequality follows from a comparison between $\mathsf{W}_p$ and $\mathsf{W}_1$ for compactly supported distributions (Lemma 4 in [21]), the third from the integral representation of $\mathsf{W}_1$, and the final inequality from truncating the inner integrals to $[-R, R]$ and observing that $p(1-p) \leq 1/4$ for $p \in [0, 1]$.

**Upper bound for $\overline{\mathsf{W}}_p^p$.**     As earlier, observe that

$$\mathbb{E}\big[\big|\overline{\mathsf{W}}_p^p(\hat{\mu}_n, \hat{\nu}_n) - \overline{\mathsf{W}}_p^p(\mu, \nu)\big|\big] \leq \mathbb{E}\left[\sup_{\theta \in \mathbb{S}^{d-1}} \left|\mathsf{W}_p^p(\mathfrak{p}_\sharp^\theta \hat{\mu}_n, \mathfrak{p}_\sharp^\theta \hat{\nu}_n) - \mathsf{W}_p^p(\mathfrak{p}_\sharp^\theta \mu, \mathfrak{p}_\sharp^\theta \nu)\right|\right]$$

$$\leq C_{p,R}\, \mathbb{E}\left[\sup_{\theta \in \mathbb{S}^{d-1}} \left(\mathsf{W}_1(\mathfrak{p}_\sharp^\theta \hat{\mu}_n, \mathfrak{p}_\sharp^\theta \mu) + \mathsf{W}_1(\mathfrak{p}_\sharp^\theta \hat{\nu}_n, \mathfrak{p}_\sharp^\theta \nu)\right)\right]$$

$$\leq C_{p,R}\left(\mathbb{E}\left[\sup_{\theta \in \mathbb{S}^{d-1}} \mathsf{W}_1(\mathfrak{p}_\sharp^\theta \hat{\mu}_n, \mathfrak{p}_\sharp^\theta \mu)\right] + \mathbb{E}\left[\sup_{\theta \in \mathbb{S}^{d-1}} \mathsf{W}_1(\mathfrak{p}_\sharp^\theta \hat{\nu}_n, \mathfrak{p}_\sharp^\theta \nu)\right]\right).$$

Now, $\mathbb{E}\left[\sup_{\theta \in \mathbb{S}^{d-1}} \mathsf{W}_1(\mathfrak{p}_\sharp^\theta \hat{\mu}_n, \mathfrak{p}_\sharp^\theta \mu)\right]$ admits the following dual representation via KR duality:

$$\mathbb{E}\left[\sup_{\theta \in \mathbb{S}^{d-1}} \mathsf{W}_1(\mathfrak{p}_\sharp^\theta \hat{\mu}_n, \mathfrak{p}_\sharp^\theta \mu)\right] = \mathbb{E}\left[\sup_{f \in \mathsf{Lip}_{1,0}(\mathbb{R}),\, \theta \in \mathbb{S}^{d-1}} (\hat{\mu}_n - \mu)(f \circ \mathfrak{p}^\theta)\right], \tag{10}$$

where $\mathsf{Lip}_{1,0}(\mathbb{R}) = \{f : \mathbb{R} \to \mathbb{R} : |f(x) - f(y)| \leq |x - y| \,\forall x, y \in \mathbb{R}, f(0) = 0\}$. By Lemma 8 in [21], the function class $\mathcal{G} = \{f \circ \mathfrak{p}^\theta : \theta \in \mathbb{S}^{d-1}, f \in \mathsf{Lip}_{1,0}(\mathbb{R})\}$ is $\mu$-Donsker, and we have

$$\log N_{[]}(\epsilon, \mathcal{G}, L^2(\mu)) \lesssim R^{5/3}\epsilon^{-3/2} + 5d \log(R/\epsilon),$$

which, by the global maximal inequality (Theorem 2.14.2 in [56]), gives,

$$\mathbb{E}\left[\sup_{f \in \mathsf{Lip}_{1,0}(\mathbb{R}),\, \theta \in \mathbb{S}^{d-1}} (\hat{\mu}_n - \mu)(f \circ \mathfrak{p}^\theta)\right] \lesssim \frac{R^{5/4} + d \log R}{\sqrt{n}}. \tag{11}$$

Combining this with (10), and repeating the same argument for $\nu$, we have the second statement.

The final statement of the theorem on $\underline{\mathsf{W}}_p$ and $\overline{\mathsf{W}}_p$ follows from the first two upper bounds combined with the elementary inequality $|a - b| \leq b^{1-p}|a^p - b^p|$ for $a \geq 0, b > 0, p \geq 1$. $\qquad\square$

### D.3   Proof of Lemma 4

The proof of this lemma essentially recovers constants in Corollary 4.6 in [31], but a full argument is included for completeness. With some abuse of notation, let $\|\nabla f\|_\infty = \sup_x \|\nabla f(x)\|$. By Theorem 4.5 in [31], for any $\lambda$-Lipschitz function $f$ with $\lambda \leq 2/\sqrt{M_\mu}$, we have

$$\mathsf{Ent}_\mu(e^f) \leq B(\lambda)\mathbb{E}\big[\|\nabla f\|_\infty^2 e^f\big], \tag{12}$$

where $\mathsf{Ent}_\mu(f) := \mathbb{E}[f \log f]$ is the entropy functional of $f$, and

$$B(\lambda) \leq \frac{M_\mu}{2}\left(\frac{2 + \lambda\sqrt{M_\mu}}{2 - \lambda\sqrt{M_\mu}}\right)e^{\sqrt{5M_\mu}\lambda}.$$

Each function $f_i$ in the statement of the proposition is $\beta$-Lipschitz. From (12) together with the tensorization property of the entropy functional (cf. Proposition 2.2 in [31]), we obtain

$$\mathsf{Ent}_{\mu^{\otimes n}}\left(\frac{\lambda f}{\beta}\right) \leq \frac{\lambda^2}{\beta^2}\sum_{i=1}^n \mathbb{E}\left[\mathsf{Ent}_\mu\left(\frac{\lambda f_i}{\beta}\right)\right] \leq \frac{\lambda^2 B(\lambda)}{\beta^2}\sum_{i=1}^n \mathbb{E}\big[\|\nabla f_i\|_\infty^2 e^f\big], \quad \forall \lambda \in \big(0, 2/\sqrt{M_\mu}\big).$$

Further, we have $B(\lambda) \leq \frac{3e^5 M_\mu}{2}$ for $\lambda \leq 1/\sqrt{M_\mu}$ and $\sum_{i=1}^n \|\nabla f_i\|_\infty^2 \leq \beta \, \mu^{\otimes n}$-a.e. by assumption. Therefore

$$\mathsf{Ent}_{\mu^{\otimes n}}\left(\frac{\lambda f}{\beta}\right) \leq \frac{3e^5 M_\mu \alpha^2}{2\beta^2}\lambda^2 \mathbb{E}\left[e^{\frac{\lambda f}{\beta}}\right], \quad \forall \lambda \in \big(0, 1/\sqrt{M_\mu}\big).$$

By Corollary 2.11 in [31], this yields

$$\mu^{\otimes n}\left(\frac{f}{\beta} > \mu\left(\frac{f}{\beta}\right) + r\right) \le \exp\left(-\min\left\{\frac{r}{2\sqrt{M_\mu}}, \frac{r^2\beta^2}{6M_\mu e^5\alpha^2}\right\}\right), \ r > 0,$$

from which the result follows by replacing $r$ with $r/\beta$. $\qquad\square$

### D.4 Proof of Theorem 2

Denoting by $\wedge$ the setwise minimum of two measures, we first recall some useful facts. Throughout we write $\Theta \sim \mathrm{Unif}(\mathbb{S}^{d-1})$ for a random direction on the sphere sampled independently of any other randomness.

**Fact 1.** *For $\mu, \nu \in \mathcal{P}(\mathbb{R}^d)$ and $p \ge 1$, we have $\mathsf{W}_p(\mu, \nu) \le \mathsf{W}_p(\mu - \mu \wedge \nu, \nu - \mu \wedge \nu)$.*

This follows by infimizing over transport plans which leave the shared mass $\mu \wedge \nu$ unmoved.

**Fact 2.** *For $\mu, \nu \in \mathcal{P}(\mathbb{R}^d)$, $p \ge 1$, and $c \ge 0$, we have $\mathsf{W}_p(c\mu, c\nu) = c^{1/p}\mathsf{W}_p(\mu, \nu)$.*

It is easy to check that these properties extend to $\underline{\mathsf{W}}_p$ and $\overline{\mathsf{W}}_p$. We also employ the following.

**Lemma 5.** *Fixing $p \ge 1$ and $\mu = \mathrm{Unif}(\mathbb{S}^{d-1})$, we have*

$$(\mu|x_1|^p)^{1/p} \asymp \sqrt{1 \wedge p/d}.$$

**Lemma 6.** *Fixing $p \ge 1$ and $\mu \in \mathcal{P}_p(\mathbb{R}^d)$, we have*

$$\mu(\|x\|^p)^{1/p} = \mathsf{W}_p(\mu, \delta_0) \asymp \sqrt{1 \vee d/p} \ \underline{\mathsf{W}}_p(\mu, \delta_0).$$

We defer proofs of the previous lemmas to Appendix D.4.4. In what follows, we will refer to any $\mathsf{D} : \mathcal{P}(\mathbb{R}^d)^2 \to [0, \infty]$ as a *statistical distance*, specifying additional properties as needed. Our risk bounds use the following standard lemma (see e.g. [20]), with a proof provided for completeness.

**Lemma 7.** *For any statistical distance $\mathsf{D}$, corruption fraction $\epsilon \in [0, 1]$, and clean family $\mathcal{G} \subseteq \mathcal{P}(\mathbb{R}^d)$, define the modulus of continuity*

$$\mathfrak{m}(\mathsf{D}, \mathcal{G}, \epsilon) = \sup_{\substack{\mu, \nu \in \mathcal{G} \\ \|\mu - \nu\|_{\mathrm{TV}} \le \epsilon}} \mathsf{D}(\mu, \nu). \tag{13}$$

*We then have*

$$\frac{1}{2}\mathfrak{m}(\mathsf{D}, \mathcal{G}, \epsilon) \le R(\mathsf{D}, \mathcal{G}, \epsilon) \le \mathfrak{m}(\mathsf{D}, \mathcal{G}, 2\epsilon).$$

*Proof.* For the lower bound, take any $\mu, \nu$ feasible for (13). Then, if the statistician observes $\epsilon$-contaminated measure $\tilde{\kappa} = \mu$, the clean measure could potentially be either $\kappa = \mu$ or $\kappa = \nu$. Hence any estimate $T(\tilde{\kappa})$ for the clean measure $\kappa$ must incur error at least $D(\mu, \nu)/2$ in the worst case. For the upper bound, consider $T$ which projects $\tilde{\kappa}$ onto $\mathcal{G}$ in TV. Then, $\|T(\tilde{\kappa}) - \kappa\|_{\mathrm{TV}} \le \|T(\tilde{\kappa}) - \tilde{\kappa}\|_{\mathrm{TV}} + \|\tilde{\kappa} - \kappa\|_{\mathrm{TV}} \le 2\epsilon$, and so $D(T(\tilde{\kappa}), \kappa) \le \mathfrak{m}(\mathsf{D}, \mathcal{G}, 2\epsilon)$ by definition. $\qquad\square$

By rescaling $\mathbb{R}^d$ appropriately, it is easy to check that $\mathfrak{m}(\mathsf{D}, \mathcal{G}_q(\sigma), \epsilon) = \sigma\mathfrak{m}(\mathsf{D}, \mathcal{G}_q(1), \epsilon)$ for our choices of $\mathsf{D}$, so we will assume $\sigma = 1$ and write $\mathcal{G}_q = \mathcal{G}_q(1)$ from now on.

#### D.4.1 Lower bounds

Immediately, we can apply Lemma 7 to obtain the lower bounds of Theorem 2.

**Proposition 9.** *Fix $1 \le p < q$ and corruption fraction $\epsilon \in [0, 1/2]$. Then we have*

$$R(\underline{\mathsf{W}}_p, \mathcal{G}_q, \epsilon) \gtrsim \sqrt{(1 \vee d/q)(1 \wedge p/d)} \ \epsilon^{1/p - 1/q}$$
$$R(\overline{\mathsf{W}}_p, \mathcal{G}_q, \epsilon) \gtrsim \epsilon^{1/p - 1/q}.$$

*Proof.* For $\overline{W}_p$, we consider $\mu = \delta_0$ and $\nu = (1 - \epsilon)\delta_0 + \epsilon\delta_y$ where $\|y\| = (2\epsilon)^{-1/q}$. Trivially, $\mu \in \mathcal{G}_q$, and

$$
\begin{aligned}
\sup_{\theta \in \mathbb{S}^{d-1}} \nu|\theta^{\mathsf{T}}(x - \nu y)|^q &= \sup_{\theta \in \mathbb{S}^{d-1}} (1 - \epsilon)\epsilon^q|\theta^{\mathsf{T}}y|^q + \epsilon(1 - \epsilon)^q|\theta^{\mathsf{T}}y|^q \\
&= \|y\|^q \left[(1 - \epsilon)\epsilon^q + \epsilon(1 - \epsilon)^q\right] \\
&\leq \frac{1}{2}\epsilon^{-1}2\epsilon(1 - \epsilon)^q \leq 1,
\end{aligned}
$$

so $\nu \in \mathcal{G}_q$ as well. Moreover, we have

$$
\overline{W}_p(\mu, \nu) = \epsilon^{1/p}\overline{W}_p(\delta_0, \delta_y) = 2^{-1/q}\epsilon^{1/p - 1/q} \geq \frac{1}{2}\epsilon^{1/p - 1/q},
$$

and so Lemma 7 gives the desired risk bound for $\overline{W}_p$.

For $\underline{W}_p$, we fix $\mu = \delta_0$ and set $\nu = (1-\epsilon)\delta_0 + \epsilon\,\mathrm{Unif}(r\mathbb{S}^{d-1})$ with $r = \epsilon^{-1/q}\overline{W}_q(\mathrm{Unif}(\mathbb{S}^{d-1}), \delta_0)^{-1}$. As before $\mu, \nu \in \mathcal{G}_q$, since

$$
\begin{aligned}
\overline{W}_q(\nu, \delta_{\nu x}) &= \epsilon^{1/q}\,\overline{W}_q(\mathrm{Unif}(r\mathbb{S}^{d-1}), \delta_0) \\
&= r\epsilon^{1/q}\,\overline{W}_q(\mathrm{Unif}(\mathbb{S}^{d-1}), \delta_0) = 1.
\end{aligned}
$$

Furthermore, we have

$$
\begin{aligned}
\underline{W}_p(\mu, \nu) &= \epsilon^{1/p}\,\overline{W}_p(\mathrm{Unif}(r\mathbb{S}^{d-1}), \delta_0) \\
&= \epsilon^{1/p - 1/q}\overline{W}_q(\mathrm{Unif}(\mathbb{S}^{d-1}), \delta_0)^{-1}\,\overline{W}_p(\delta_0, \mathrm{Unif}(\mathbb{S}^{d-1})) \\
&\asymp \sqrt{(1 \vee d/q)(1 \wedge p/d)}\,\epsilon^{1/p - 1/q},
\end{aligned}
$$

where the last relation uses Lemma 5. Again, we obtain the desired risk bound via Lemma 7. $\qquad\square$

### D.4.2  Upper bounds

Next, we introduce an important notion of *(generalized) resilience* [55, 61]. We say that a distribution $\mu \in \mathcal{P}(\mathbb{R}^d)$ is $(\rho, \epsilon)$-resilient w.r.t. a statistical distance D if $D(\mu, \nu) \leq \rho$ for all distributions $\nu \leq \frac{1}{1-\epsilon}\mu$ (i.e. for all $\epsilon$-deletions of $\mu$). Standard (mean) resilience refers to resilience w.r.t. $D_{\mathrm{mean}}(\mu, \nu) = \|\mu x - \nu x\|$. Writing $\mathcal{G}^{\mathsf{D}}_{\rho,\epsilon} \subset \mathcal{P}(\mathbb{R}^d)$ for the family of $\mu \in \mathcal{P}(\mathbb{R}^d)$ which are $(\rho, \epsilon)$-resilient w.r.t. D, we have the following standard result.

**Proposition 10.** *Fix $0 \leq \epsilon < 1/2$, $\rho \geq 0$, and D satisfying the triangle inequality. Then, we have $R(\mathsf{D}, \mathcal{G}^{\mathsf{D}}_{\rho,2\epsilon}, \epsilon) \leq 2\rho$.*

*Proof.* Fix $\mu, \nu \in \mathcal{G}^{\mathsf{D}}_{\rho,2\epsilon}$ with $\|\mu - \nu\|_{\mathrm{TV}} \leq 2\epsilon$. We consider the midpoint $\gamma = \frac{1}{1 - \|\mu - \nu\|_{\mathrm{TV}}}\mu \wedge \nu \in \mathcal{P}(\mathbb{R}^d)$ and compute

$$
D(\mu, \nu) \leq D(\mu, \gamma) + D(\nu, \gamma) \leq 2\rho,
$$

implying the desired risk bound via Lemma 7. $\qquad\square$

We will also use the following standard result for one-dimensional (mean) resilience (see e.g. [55, Proposition 23]), which is a consequence of Markov's inequality.

**Lemma 8.** *Fix $0 \leq \epsilon \leq 1/2$ and $\mu \in \mathcal{P}(\mathbb{R})$ with $\mu|x - x_0|^p \leq \sigma^p$ for some $x_0 \in \mathbb{R}$. Then, for all distributions $\nu \leq \frac{1}{1-\epsilon}\mu$, we have $|\mu x - \nu x| \lesssim \sigma\epsilon^{1-1/p}$.*

Next, for $\mathsf{D} \in \{\underline{W}_p, \overline{W}_p\}$, we show that it suffices to prove resilience with respect to the simpler distances defined by

$$
\underline{D}_p(\mu, \nu) = \underline{D}_p(\mu - \nu) := \left|\mathbb{E}[(\mu - \nu)(|\Theta^{\mathsf{T}}x|^p)]\right| = \left|\underline{W}_p^p(\mu, \delta_0) - \underline{W}_p^p(\nu, \delta_0)\right|
$$

$$
\text{and} \quad \overline{D}_p(\mu, \nu) = \overline{D}_p(\mu - \nu) := \sup_{\theta \in \mathbb{S}^{d-1}} \left|(\mu - \nu)(|\theta^{\mathsf{T}}x|^p)\right|,
$$

respectively. These distances encode a certain similarity of moment tensors, with $\overline{D}_2(\mu, \nu) = \|\Sigma_\mu + (\mu x)(\mu x)^{\mathsf{T}} - \Sigma_\nu - (\nu x)(\nu x)^{\mathsf{T}}\|_{\mathrm{op}}$.

Recall that $\mathsf{D} = \mathsf{D}_{\mathcal{F}}$ is an integral probability metric (IPM) w.r.t. a class $\mathcal{F}$ of measurable functions on $\mathbb{R}^d$ if $D(\mu, \nu) = \sup_{f \in \mathcal{F}}(\mu - \nu)(f)$. By design, we have the following.

**Lemma 9.** *The statistical distances $\underline{D}_p$ and $\overline{D}_p$ are IPMs w.r.t. the function classes $\underline{\mathcal{F}}_p = \{x \mapsto c_p s\|x\|^p : s \in \{\pm 1\}\}$ and $\overline{\mathcal{F}}_p = \{x \mapsto s|\theta^\mathsf{T} x|^p : s \in \{\pm 1\}, \theta \in \mathbb{S}^{d-1}\}$, respectively, where $c_p = \mathbb{E}[|\Theta_1|^p] \asymp \sqrt{1 \wedge p/d}$. Moreover, $\underline{D}_p(\mu, \delta_0) = \underline{W}_p^p(\mu, \delta_0)$ and $\overline{D}_p(\mu, \delta_0) = \overline{W}_p^p(\mu, \delta_0)$.*

*Proof.* For $\underline{D}_p$, we compute

$$
\begin{aligned}
\underline{D}_p(\mu, \nu) &= \left| \mathbb{E}[(\mu - \nu)(|\Theta^\mathsf{T} x|^p)] \right| \\
&= \left| (\mu - \nu)(\mathbb{E}|\Theta^\mathsf{T} x|^p) \right| \\
&= \left| (\mu - \nu)(c_p\|x\|^p) \right| \\
&= \sup_{s \in \{\pm 1\}} (\mu - \nu)(c_p s\|x\|^p).
\end{aligned}
$$

Likewise, for $\overline{D}_p$, we check

$$
\begin{aligned}
\overline{D}_p(\mu, \nu) &= \sup_{\theta \in \mathbb{S}^{d-1}} \left| (\mu - \nu)(|\theta^\mathsf{T} x|^p) \right| \\
&= \sup_{s \in \{\pm 1\}, \theta \in \mathbb{S}^{d-1}} (\mu - \nu)(s|\theta^\mathsf{T} x|^p).
\end{aligned}
$$

Computations when $\nu = \delta_0$ are trivial, since there is a single coupling between $\mu$ and $\nu$. $\square$

The third property is particularly relevant to resilience.

**Lemma 10.** *Let $D = D_{\mathcal{F}}$ be an IPM. Then $\mu$ is $(\rho, \epsilon)$-resilient w.r.t. $D$ if and only if $\mu$ is $(\epsilon(1-\epsilon)^{-1}\rho, 1 - \epsilon)$-resilient w.r.t. $D$.*

*Proof.* Writing $\mu = (1 - \epsilon)\nu + \epsilon\alpha$ for some $\nu, \alpha \in \mathcal{P}(\mathbb{R}^d)$, we have

$$
\begin{aligned}
D(\nu, \mu) &= D'(\epsilon^{-1}[\mu - (1 - \epsilon)\alpha] - \mu) \\
&= \frac{1 - \epsilon}{\epsilon} D(\mu, \alpha) \qquad\qquad\qquad \text{(homogeneity).} \qquad \square
\end{aligned}
$$

We now formally translate resilience w.r.t. $\underline{D}_p$ and $\overline{D}_p$ to that which we desire.

**Proposition 11.** *Fix $0 \le \epsilon < 1$, $\rho \ge 0$, and $(D, D') \in \{(\overline{W}_p, \overline{D}_p), (\underline{W}_p, \underline{D}_p)\}$. If $\mu \in \mathcal{P}(\mathbb{R}^d)$ with $\mu x = 0$ is $(\rho, \epsilon)$-resilient w.r.t. $D'$, then $\mu$ is $(2\rho^{1/p} + 2\epsilon^{1/p}D(\mu, \delta_0), \epsilon)$-resilient w.r.t. $D$.*

*Proof.* Fixing such $\mu$ and taking $\nu \le \frac{1}{1-\epsilon}\mu$, write $\mu = (1-\epsilon)\nu + \epsilon\alpha$ for some $\alpha \in \mathcal{P}(\mathbb{R}^d)$ and write $\tau = \epsilon \wedge (1-\epsilon)$, so that $\nu, \alpha \le \tau^{-1}\mu$. Then, we bound

$$
\begin{aligned}
D(\mu, \nu)^p &= D((1-\epsilon)\nu + \epsilon\alpha, \nu)^p \\
&\le \epsilon D(\alpha, \nu)^p && \text{(Facts 1 and 2)} \\
&\le 2^p \epsilon \sup_{\kappa \le \tau^{-1}\mu} D(\kappa, \delta_0)^p && \text{(triangle inequality for D)} \\
&= 2^p \epsilon \sup_{\kappa \le \tau^{-1}\mu} D'(\kappa, \delta_0) && \text{(Lemma 9)} \\
&\le 2^p \epsilon \sup_{\kappa \le \tau^{-1}\mu} D'(\kappa, \mu) + 2^p \epsilon D(\mu, \delta_0)^p && \text{(triangle inequality for D')} \\
&\le 2^p \rho + 2^p \epsilon D(\mu, \delta_0)^p, && \text{(Lemma 10)}
\end{aligned}
$$

giving the desired bound after taking $p$th roots. $\square$

Equipped with this result, we are prepared to prove the upper bounds of Theorem 2. Given $\mu \in \mathcal{G}_q$, we must provide bounds on $D(\mu, \delta_0)$ as well as the resilience of $\mu$ w.r.t. $D'$.

**Lemma 11.** *Fixing $1 \le p < q$ and $\mu \in \mathcal{G}_q$ with $\mu x = 0$, we have*

$$
\begin{aligned}
\underline{W}_p(\mu, \delta_0) &\lesssim \sqrt{(1 \vee d/q)(1 \wedge p/d)} \\
\overline{W}_p(\mu, \delta_0) &\lesssim 1.
\end{aligned}
$$

**Lemma 12.** *Fix $1 \leq p < q$, corruption fraction $0 \leq \epsilon \leq 1/2$, and $\mu \in \mathcal{G}_q$ with $\mu x = 0$. Then, $\mu$ is $(C\sqrt{(1 \vee d/q)(1 \wedge p/d)}\epsilon^{1/p-1/q}, \epsilon)$-resilient w.r.t. $\underline{D}_p^{1/p}$ and $(C\epsilon^{1/p-1/q}, \epsilon)$-resilient w.r.t. $\overline{D}_p^{1/p}$, for some absolute constant $C > 0$.*

Together, these give the desired risk bounds.

**Proposition 12.** *Fix $1 \leq p < q$ and corruption fraction $0 \leq \epsilon \leq 0.49$. Then we have*

$$R(\underline{W}_p, \mathcal{G}_q, \epsilon) \lesssim \sqrt{(1 \vee d/q)(1 \wedge p/d)} \, \epsilon^{1/p-1/q}$$
$$R(\overline{W}_p, \mathcal{G}_q, \epsilon) \lesssim \epsilon^{1/p-1/q}.$$

*Proof.* Fixing $\mu \in \mathcal{G}_q$, it suffices by Proposition 10 to prove that $\mu$ is $(C\epsilon^{1/p-1/q}, \epsilon)$-resilient w.r.t. $\overline{W}_p$ and $(C\sqrt{(1 \vee d/q)(1 \wedge p/d)} \, \epsilon^{1/p-1/q}, \epsilon)$-resilient w.r.t. $\underline{W}_p$ for all $0 \leq \epsilon \leq 0.98$, where $C > 0$ is some absolute constant. Since these distances are translation invariant, we can assume without loss of generality that $\mu x = 0$. By Lemmas 11 and 12, we know that $\mu$ is $(C\epsilon^{1/p-1/q}, \epsilon)$-resilient w.r.t. $\overline{D}_p^{1/p}$ and $(C\sqrt{(1 \vee d/q)(1 \wedge p/d)} \, \epsilon^{1/p-1/q}, \epsilon)$-resilient w.r.t. $\underline{D}_p^{1/p}$ for all $0 \leq \epsilon \leq 1/2$ and some absolute constant $C > 0$. For $1/2 \leq \epsilon \leq 0.98$, the same resiliency bounds are implied by Lemma 10, since $1 - \epsilon \geq 0.02 \geq \epsilon/49$. Finally, we apply Proposition 11 to obtain the desired risk bounds. $\square$

We now prove the preceding lemmas.

*Proof of Lemma 11.* Fixing $\mu \in \mathcal{G}_q$ with $\mu x = 0$, we bound

$$
\begin{aligned}
\underline{W}_p(\mu, \delta_{\mu x}) &\asymp \sqrt{1 \wedge p/d} \, W_p(\mu, \delta_{\mu x}) & \text{(Lemma 6)} \\
&\leq \sqrt{1 \wedge p/d} \, W_q(\mu, \delta_{\mu x}) & (q > p) \\
&= \sqrt{1 \wedge p/d} \, \frac{W_q(\mu, \delta_{\mu x})}{\underline{W}_q(\mu, \delta_{\mu x})} \underline{W}_q(\mu, \delta_{\mu x}) & \\
&\asymp \sqrt{(1 \wedge p/d)(1 \vee d/q)} \, \underline{W}_q(\mu, \delta_{\mu x}) & \text{(Lemma 6)} \\
&\leq \sqrt{(1 \wedge p/d)(1 \vee d/q)} & \mu \in \mathcal{G}_q. \quad (14)
\end{aligned}
$$

Similarly, we obtain

$$\overline{W}_p(\mu, \delta_{\mu x}) \leq \overline{W}_q(\mu, \delta_{\mu x}) \leq 1. \qquad \square$$

*Proof of Lemma 12.* By Lemma 9, $\underline{D}_p$ and $\overline{D}_p$ are IPMs with respect to the stated function classes $\underline{\mathcal{F}}_p$ and $\overline{\mathcal{F}}_p$, respectively. Note that if $D = D_{\mathcal{F}}$ is an IPM for any symmetric $\mathcal{F} = -\mathcal{F}$, then $\mu$ is $(\rho, \epsilon)$-resilient w.r.t. D if and only if $f_\sharp \mu$ is $(\rho, \epsilon)$-resilient (in mean) for all $f \in \mathcal{F}$.

Now, fix $\mu \in \mathcal{G}_q$ with $\mu x = 0$. For $\underline{D}_p$, we observe that

$$
\begin{aligned}
\mu((\|x\|^p)^{q/p}) &= \mu(\|x\|^q) & \\
&\lesssim C^q (1 \vee d/q)^{q/2} \sup_{\theta \in \mathbb{S}^{d-1}} \mu(|\theta^\mathsf{T} x|^q) & \text{(Lemma 6)} \\
&\leq C^q (1 \vee d/q)^{q/2} & (\mu \in \mathcal{G}_q) \\
&= \left[ C^p (1 \vee d/q)^{p/2} \right]^{q/p},
\end{aligned}
$$

for some absolute constant $C > 0$. For $f \in \underline{\mathcal{F}}_p$, we then have that $f_\sharp \mu$ has $q/p$-th moments bounded by $O(C^p (1 \wedge p/d)^{p/2}(1 \vee d/q)^{p/2})$, and is thus $(O(C^p(1 \wedge p/d)^{p/2}(1 \vee d/q)^{p/2}\epsilon^{1-p/q}), \epsilon)$-resilient, by Lemma 8. Taking $p$th roots gives the claim.

For $\overline{D}_p$, note that for $\theta \in \mathbb{S}^{d-1}$, we have

$$\mu((|\theta^\mathsf{T} x|^p)^{q/p}) = \mu(|\theta^\mathsf{T} x|^q) \leq 1.$$

For $f \in \overline{\mathcal{F}}_p$, we then have that $f_\sharp \mu$ has $q/p$-th moments bounded by 1 and is thus $O(\epsilon^{1-p/q}, \epsilon)$-resilient. Taking $p$th roots gives the claim. $\square$

### D.4.3 Higher-dimensional slicing

We now extend Proposition 12 to the $k$-dimensional sliced distances defined by

$$\underline{\mathsf{W}}_{p,k}(\mu,\nu) := \left[\int_{\mathrm{Gr}_k(\mathbb{R}^d)} \mathsf{W}_p^p(\mathfrak{p}_\sharp^E\mu, \mathfrak{p}_\sharp^E\nu)d\sigma_k(E)\right]^{1/p} \quad \text{and} \quad \overline{\mathsf{W}}_{p,k}(\mu,\nu) := \sup_{E\in\mathrm{Gr}_k(\mathbb{R}^d)} \mathsf{W}_p(\mathfrak{p}_\sharp^E\mu, \mathfrak{p}_\sharp^E\nu),$$

where $\mathrm{Gr}_k(\mathbb{R}^d)$ is the Grassmannian of $k$-dimensional linear subspaces of $\mathbb{R}^d$, $\sigma_k$ is its standard Haar measure, and $\mathfrak{p}^E$ is the orthogonal projection onto $E \in \mathrm{Gr}_k(\mathbb{R}^d)$. These coincide with $\underline{\mathsf{W}}_p$ and $\overline{\mathsf{W}}_p$ when $k = 1$, and both equal $\mathsf{W}_p$ when $k = d$. We focus here on $\overline{\mathsf{W}}_p$, with $\underline{\mathsf{W}}_p$ inheriting the same risk bound, although stronger guarantees can be obtained in a similar manner to the proof of Proposition 12. First, we extend $\overline{\mathsf{D}}_p$ to this regime as

$$\overline{\mathsf{D}}_{p,k} := \sup_{\substack{U\in\mathbb{R}^{d\times k} \\ U^\intercal U=I_k}} \left|(\mu-\nu)(\|U^\intercal x\|^p)\right|,$$

and observe that all of the properties from Lemma 9 still hold. Moreover, for $\mu \in \mathcal{G}_q$ with $\mu x = 0$, we obtain the needed analog of Lemma 11, bounding

$$\overline{\mathsf{W}}_{p,k}(\mu,\delta_0)^{1/p} = \sup_{\substack{U\in\mathbb{R}^{d\times k} \\ U^\intercal U=I_k}} \mu(\|U^\intercal x\|^p)^{1/p}$$

$$\leq \sup_{\substack{U\in\mathbb{R}^{d\times k} \\ U^\intercal U=I_k}} \mu(\|U^\intercal x\|^q)^{1/q} \qquad (q > p)$$

$$\leq \sqrt{1\vee k/q} \sup_{\substack{U\in\mathbb{R}^{d\times k} \\ U^\intercal U=I_k}} \sup_{\theta\in\mathbb{S}^{k-1}} \mu(|\theta^\intercal U^\intercal x|^q)^{1/q} \qquad (\text{Lemma 6})$$

$$\leq \sqrt{1\vee k/q} \sup_{\theta\in\mathbb{S}^{d-1}} \mu(|\theta^\intercal x|^q)^{1/q} \qquad (\mathbb{S}^{k-1}\subset\mathbb{S}^{d-1})$$

$$\leq \sqrt{1\vee k/q} \qquad (\mu\in\mathcal{G}_q).$$

In the same way, we can extract this factor of $\sqrt{1\vee k/q}$ for the resiliency of $\mu$ w.r.t. $\overline{\mathsf{D}}_{p,k}^{1/p}$ to prove the needed analog of Lemma 12. Combining these results gives that $R(\underline{\mathsf{W}}_{p,k}, \mathcal{G}_q, \epsilon) \leq R(\overline{\mathsf{W}}_{p,k}, \mathcal{G}_q, \epsilon) \lesssim \sqrt{1\vee k/q}\, R(\overline{\mathsf{W}}_p, \mathcal{G}_q, \epsilon) \asymp \sqrt{1\vee k/q}\, \epsilon^{1/p-1/q}$ for $0 \leq \epsilon \leq 0.49$, as desired.

### D.4.4 Proofs of auxiliary lemmas

*Proof of Lemma 5.* Let $\Theta \sim \mathrm{Unif}(\mathbb{S}^{d-1})$. When $d = 1$, we have $\mathbb{E}[|\Theta_1|^p] = 1$. Otherwise, we use that the probability density function of $\Theta_1$ at $s \in [-1,1]$ is proportional to $(1-s^2)^{\frac{d-3}{2}}$ [52]. Equivalently, $(\Theta_1 + 1)/2 \sim \mathrm{Beta}\left(\frac{d-1}{2}, \frac{d-1}{2}\right)$. We will first prove the desired statement for even integer $p = 2m$, where

$$\mathbb{E}\left[|\Theta_1|^{2m}\right] = \frac{(2m)!}{2^{2m}m!} \frac{\Gamma(d-1)\Gamma(\frac{d-1}{2}+m)}{\Gamma(\frac{d-1}{2})\Gamma(d-1+2m)}$$

(see, e.g., [38]). Simplifying, we obtain

$$\mathbb{E}\left[|\Theta_1|^{2m}\right] = \frac{\Gamma(2m+1)}{2^{2m}\Gamma(m+1)} \frac{\Gamma(d-1)\Gamma(\frac{d-1}{2}+m)}{\Gamma(\frac{d-1}{2})\Gamma(d-1+2m)}$$

$$= \frac{\Gamma(d/2)\Gamma(m+1/2)}{\sqrt{\pi}2^{2m}\Gamma(m+d/2)}.$$

Employing Stirling's formula, we compute

$$2^{2m}\mathbb{E}\left[|\Theta_1|^{2m}\right] \asymp \frac{\Gamma(d/2)\Gamma(m+1/2)}{\Gamma(m+d/2)}$$

$$\asymp \frac{(d/2)^{d/2-1/2}e^{-d/2}(m+1/2)^m e^{-m-1/2}}{(m+d/2)^{m+d/2-1/2}e^{-m-d/2}}$$

$$\asymp \frac{(d/2)^{d/2-1/2}(m+1/2)^m}{(m+d/2)^{m+d/2-1/2}}$$

$$= \left(\frac{d/2}{m+d/2}\right)^{\frac{d-1}{2}} \left(\frac{m+1/2}{m+d/2}\right)^m.$$

Consequently, we have

$$\mathbb{E}\left[|\Theta_1|^{2m}\right]^{1/2m} \asymp \left(1+\frac{m}{d/2}\right)^{-\frac{d-1}{4m}} \sqrt{\frac{m+1/2}{m+d/2}}$$

$$\asymp \sqrt{\frac{m+1/2}{m+d/2}}$$

$$\asymp 1 \wedge \sqrt{m/d},$$

as desired. When $p \geq 2$ is not an even integer, we use that $\mathbb{E}\left[|\Theta_1|^p\right]$ is monotonically increasing in $p$ to obtain matching bounds by rounding $p$ up and down to the nearest even integers. To obtain the needed lower bound when $p \in [1, 2)$, we derive

$$\mathbb{E}\left[|\Theta_1|\right] = \frac{4\Gamma(d-1)}{\Gamma\left(\frac{d-1}{2}\right)^2} \frac{\left(\frac{d-1}{2}\right)^{d-1}}{(d-1)^d},$$

using the formula for the mean absolute deviation of the beta distribution [23]. Applying Stirling's formula once more, we obtain

$$\mathbb{E}\left[|\Theta_1|\right] \asymp \frac{(d-1)^{d-3/2}\left(\frac{d-1}{2}\right)^{d-1}}{(\frac{d-1}{2})^{d-2}(d-1)^d} = \frac{(d-1)/2}{(d-1)^{3/2}} \asymp d^{-1/2},$$

as desired. $\qquad\square$

*Proof of Lemma 6.* Taking $X \sim \mu$ and $\Theta \sim \mathrm{Unif}(\mathbb{S}^{d-1})$, we use rotational symmetry of the sphere to compute

$$\mathbb{E}\left[|\Theta^\intercal X|^p\right] = \mathbb{E}\left[|\Theta_1|^p\right] \mathbb{E}\left[\|X\|^p\right],$$

giving the first equality via Lemma 5. The inequality follows by comparing an average to a supremum, and the inequality is tight when these coincide, i.e. when $\mu$ is rotationally symmetric about 0. $\qquad\square$

### D.5 Proof of Proposition 2

The high-level structure of our proof follows a standard template for finite-sample robust mean and covariance estimation (see, e.g., [55, 61]). We first prove Proposition 2 under bounded support and then extend our result to the general setting. Throughout, we write $\mathbb{B}_r := \{x \in \mathbb{R}^d : \|x\| \leq r\}$ for the Euclidean ball of radius $r \geq 0$.

**Bounded Support:** For ease of presentation, we slightly extend our notion of resilience in a standard way. We say that $\mu \in \mathcal{P}(\mathbb{R}^d)$ is $(\rho, \epsilon)$-resilient w.r.t. D about $\kappa \in \mathcal{P}(\mathbb{R}^d)$ if $\mathsf{D}(\nu, \kappa) \leq \rho$ for all $\nu \leq \frac{1}{1-\epsilon}\mu$. Namely, we will consider the resilience of an empirical measure $\kappa = \hat{\mu}_n$ about its population measure $\mu$. If $\mathsf{D} = \mathsf{D}_\mathcal{F}$ is an IPM for symmetric $\mathcal{F} = -\mathcal{F}$, note that $\mu$ is $(\rho, \epsilon)$-resilient w.r.t. D about $\kappa$ if and only if $f_\sharp\mu$ is $(\rho, \epsilon)$-resilient (in mean) about $f_\sharp\kappa$ for all $f \in \mathcal{F}$.

We first recall and derive some basic results for finite-sample resilience. The following lemma is a simplification of [55, Proposition 4], specified to the 1-dimensional case.

**Lemma 13** (1-dimensional finite-sample resilience). *Suppose that $\mu \in \mathcal{P}([-M, M])$ is $(\rho, \epsilon)$-resilient in mean for $\epsilon \leq 0.999$. Then, with probability at least $1 - \delta$, the empirical distribution $\hat{\mu}_n$ is $(\rho', \epsilon)$-resilient in mean about $\mu$ with $\rho' = O\left(\rho\left(1 + \sqrt{\frac{\log(1/\delta)}{\epsilon^2 n}}\right) + \frac{M\log(1/\delta)}{n}\right).$*

The result is stated in [55] for $\epsilon < 1/2$, but the proof only uses that $\epsilon$ is bounded away from 1. We then extend this result to IPMs over uniformly bounded function classes.

**Proposition 13** (Finite-sample resilience w.r.t. IPMs). *Let $\mathsf{D}_{\mathcal{F}}$ be the IPM induced by a function class $\mathcal{F} = -\mathcal{F}$ on $\mathbb{R}^d$ with $\|f\|_\infty \le M$ for $f \in \mathcal{F}$, and fix any finite subset $\mathcal{H} \subseteq \mathcal{F}$ such that $\mathsf{D}_{\mathcal{F}}(\mu, \nu) \le \mathsf{D}_{\mathcal{H}}(\mu, \nu) + \rho$. Then if $\mu \in \mathcal{P}(\mathbb{R}^d)$ is $(\rho, \epsilon)$-resilient w.r.t. $\mathsf{D}_{\mathcal{F}}$ for $\epsilon \le 0.999$, we have that $\hat{\mu}_n$ is $(\rho', \epsilon)$-resilient w.r.t. $\mathsf{D}_{\mathcal{F}}$ about $\mu$ with probability $1 - \delta$, where $\rho' = O\left(\rho + \frac{\rho}{\epsilon}\sqrt{\frac{\log(|\mathcal{H}|/\delta)}{n}} + \frac{M\log(|\mathcal{H}|/\delta)}{n}\right)$.*

*Proof.* For any $\nu_n \le (1-\epsilon)\hat{\mu}_n$, we have

$$\mathsf{D}_{\mathcal{F}}(\nu_n, \mu) \le \max_{f \in \mathcal{H}}(\nu_n - \mu)(f) + \rho.$$

Now, fixing any $f \in \mathcal{H}$, resilience of $\mu$ w.r.t. $\mathsf{D}_{\mathcal{F}}$ requires that $f_\sharp \mu$ is $(\rho, \epsilon)$-resilient. Noting that $f_\sharp \nu_n \le \frac{1}{1-\epsilon}f_\sharp\hat{\mu}_n$, Lemma 13 gives that

$$|(\nu_n - \mu)(f)| = |(f_\sharp\nu_n)x - (f_\sharp\mu)x| \le O\left(\rho + \frac{\rho}{\epsilon}\sqrt{\frac{\log(|\mathcal{H}|/\delta)}{n}} + \frac{M\log(|\mathcal{H}|/\delta)}{n}\right)$$

with probability at least $1 - \delta/|\mathcal{H}|$. A union bound over $f \in \mathcal{H}$ gives the desired result. $\qquad\square$

To apply this result, we approximate $\overline{\mathsf{D}}_p$ with an IPM over a finite function class.

**Lemma 14** (Approximating $\overline{\mathsf{D}}_p$). *For each $\gamma > 0$, there exists a net $\mathcal{N} \subset \mathbb{S}^{d-1}$ of size $(10R^p p/\gamma)^d$ such that for all $\mu, \nu \in \mathcal{P}(\mathbb{B}_R)$, we have*

$$\overline{\mathsf{D}}_p(\mu, \nu) = \sup_{\theta \in \mathbb{S}^{d-1}} |(\mu - \nu)(|\theta^\mathsf{T} x|^p)| \le \max_{\theta \in \mathcal{N}} |(\mu - \nu)(|\theta^\mathsf{T} x|^p)| + \gamma.$$

*Proof.* Let $\mathcal{N}$ be a $\gamma(2R^p p)^{-1}$-covering for $\mathbb{S}^{d-1}$ in $\ell_2$ with $|\mathcal{N}| \le (10R^p p\gamma^{-1})^d$, the existence of which is guaranteed by Lemma 3 (i). Then, taking $\theta$ to be a direction achieving the LHS supremum, and $\tilde{\theta} \in \mathcal{N}$ to be its nearest neighbor in $\mathcal{N}$, we have

$$|(\mu - \nu)(|\theta^\mathsf{T} x|^p)| \le |(\mu - \nu)(|\tilde{\theta}^\mathsf{T} x|^p)| + 2\sup_{\kappa \in \mathcal{P}(\mathbb{B}_R)} \kappa||\theta^\mathsf{T} x|^p - |\tilde{\theta}^\mathsf{T} x|^p|$$

$$\le |(\mu - \nu)(|\tilde{\theta}^\mathsf{T} x|^p)| + 2R^p p\|\theta - \tilde{\theta}\|$$

$$\le |(\mu - \nu)(|\tilde{\theta}^\mathsf{T} x|^p)| + \gamma,$$

where the second inequality follows by Lipschitzness. Supremizing over $\theta$ gives the lemma. $\qquad\square$

Combining, we obtain finite-sample resilience w.r.t. our distances of interest. Slightly abusing notation for brevity, we write $\underline{\rho}(\tau) = \underline{\rho}(\tau, p, d, q) = \sqrt{(1 \vee d/q)(1 \wedge p/d)}\tau^{1/p-1/q}$ and $\overline{\rho}(\tau) = \overline{\rho}(\tau, p, q) = \tau^{1/p-1/q}$ for our resilience bounds for the class $\mathcal{G}_q$ w.r.t. $\underline{\mathsf{W}}_p$ and $\overline{\mathsf{W}}_p$.

**Lemma 15** (Finite-sample resilience under bounded support). *Let $0 \le \epsilon \le 0.999$ and $q > p$. If $\mu \in \mathcal{G}_q$ with $\mathrm{diam}(\mathrm{spt}(\mu)) \le R/2$ and $n = \Omega\big((R^p + \epsilon^{-2})(d\log(R/\epsilon) + \log(1/\delta))\big)$, then $\hat{\mu}_n$ is $(O(\underline{\rho}(\epsilon)), \epsilon)$-resilient w.r.t. $\underline{\mathsf{W}}_p$ and $(O(\overline{\rho}(\epsilon)), \epsilon)$-resilient w.r.t. $\overline{\mathsf{W}}_p$ with probability $1 - \delta$.*

*Proof.* Assume without loss of generality that $\mu x = 0$ and $\mu \in \mathcal{G}_q \cap \mathcal{P}(\mathbb{B}_R)$. By Lemma 12 (combined with Lemma 10 if $\epsilon \ge 1/2$), we have that $\mu$ is $(\underline{\rho}(\epsilon), \epsilon)$-resilient w.r.t. $\underline{\mathsf{D}}_p^{1/p}$ and $(\overline{\rho}(\epsilon), \epsilon)$-resilient w.r.t. $\overline{\mathsf{D}}_p^{1/p}$. For $\overline{\mathsf{D}}_p$, observe that for $\|x\| \le R$ and $\theta \in \mathbb{S}^{d-1}$, we have $|\theta^\mathsf{T} x|^p \le R^p$. Thus, applying Proposition 13 with $\mathcal{F} = \overline{\mathcal{F}}_p$, $M = R^p$, and $\mathcal{H}$ induced by the net from Lemma 14 with $\gamma = \overline{\rho}(\epsilon)^p$ gives that $\hat{\mu}_n$ is $(O(2^p\overline{\rho}(\epsilon)^p), \epsilon)$-resilient about $\mu$ w.r.t. $\overline{\mathsf{D}}_p$ with probability at least $1 - \delta/2$ whenever

$$n \ge (\overline{\rho}(\epsilon)^p R^p + \epsilon^{-2})\log(2|\mathcal{H}|/\delta)/2^p$$

$$= (\overline{\rho}(\epsilon)^p R^p + \epsilon^{-2})\log((20R^p p/\overline{\rho}(\epsilon)^p)^d/\delta)/2^p.$$

Plugging in our value for $\overline{\rho}(\epsilon)$ and applying some crude bounds shows the stated sample complexity of $n = \Omega\big((R^p + \epsilon^{-2})(d\log(R/\epsilon) + \log(1/\delta))\big)$ suffices. Since the resilience bound is centered about

$\mu$ and $\mu \in \mathcal{G}_q$, we deduce that $\overline{\mathsf{D}}_p(\hat{\mu}_n, \delta_0) \leq \overline{\mathsf{D}}_p(\hat{\mu}_n, \mu) + \overline{\mathsf{D}}_p(\mu, \delta_0) \leq O(2^p \overline{\rho}(\epsilon)^p) + 1 = O(2^p)$. Thus, by Proposition 11, we have that $\hat{\mu}_n$ is $(O(\overline{\rho}(\epsilon)), \epsilon)$-resilient w.r.t. $\overline{\mathsf{W}}_p$. An analogous argument shows that the same sample complexity suffices for $\underline{\mathsf{W}}_p$ (of course, far fewer samples are actually needed, but we shall not focus on this distinction). Applying a union bound gives that both resilience guarantees hold with probability $1 - \delta$. $\qquad\square$

Finally, we use finite-sample resiliency to bound finite-sample robust estimation risk.

**Proposition 14.** *Let $0 \leq \epsilon \leq 0.499$ and $q > p$ and $\mathsf{D} \in \{\underline{\mathsf{W}}_p, \overline{\mathsf{W}}_p\}$. Then there exists an estimation procedure $\mathcal{A}$ with the following guarantee: for any $\mu \in \mathcal{G}_q$ with $\mathrm{diam}(\mathrm{spt}(\mu)) \leq R/2$ and $n = \Omega\big((R^p + \epsilon^{-2})(d \log(R/\epsilon) + \log(1/\delta))\big)$, after observing any random distribution $\tilde{\mu}_n$ such that $\|\tilde{\mu}_n - \hat{\mu}_n\|_{\mathrm{TV}} \leq \epsilon$ almost surely, $\mathcal{A}$ produces $\nu$ such that $\mathsf{D}(\nu, \mu) \lesssim R(\mathsf{D}, \mathcal{G}_q, \epsilon) + \mathsf{D}(\hat{\mu}_n, \mu)$ with probability $1 - \delta$.*

*Proof.* For $\mathsf{D} = \overline{\mathsf{W}}_p$, define

$$\Pi_{\epsilon, \rho}(\tilde{\mu}_n) = \big\{\kappa \in \mathcal{P}(\mathbb{R}^d) : \|\kappa - \tilde{\mu}_n\|_{\mathrm{TV}} \leq \epsilon \text{ and } \kappa \text{ is } (\rho, 2\epsilon)\text{-resilient w.r.t. } \overline{\mathsf{W}}_p\big\}$$

Write $\rho_\star = \inf\{\rho \geq 0 : \Pi_{\epsilon, \rho}(\tilde{\mu}_n) \neq \emptyset\}$ for the smallest resilience parameter such that this set is non-empty. Since $2\epsilon \leq 0.999$, we know by Lemma 15 that $\hat{\mu}_n$ is $(O(\overline{\rho}(2\epsilon)), 2\epsilon)$-resilient w.r.t. $\overline{\mathsf{W}}_p$ (for an appropriate choice of constant in the sample complexity) with probability $1 - \delta/2$. Noting that $\|\tilde{\mu}_n - \hat{\mu}_n\|_{\mathrm{TV}} \leq \epsilon$, we have $\rho_\star \lesssim \overline{\rho}(2\epsilon) \lesssim \overline{\rho}(\epsilon)$ with probability $1 - \delta/2$.

Now consider any algorithm which returns $\nu \in \Pi_{\epsilon, 2\rho_\star}(\tilde{\mu}_n)$. Then we have $\|\nu - \hat{\mu}_n\|_{\mathrm{TV}} \leq \|\nu - \tilde{\mu}_n\|_{\mathrm{TV}} + \|\tilde{\mu}_n - \hat{\mu}_n\|_{\mathrm{TV}} \leq 2\epsilon$. By considering their midpoint $\kappa = \frac{1}{1 - \|\nu - \hat{\mu}_n\|_{\mathrm{TV}}} \nu \wedge \hat{\mu}_n$ and applying $(O(\overline{\rho}(2\epsilon)), 2\epsilon)$-resilience of $\nu$ and $\hat{\mu}_n$ w.r.t. $\overline{\mathsf{W}}_p$, we deduce that $\overline{\mathsf{W}}_p(\nu, \hat{\mu}_n) \leq \overline{\mathsf{W}}_p(\nu, \kappa) + \overline{\mathsf{W}}_p(\kappa, \hat{\mu}_n) \lesssim \overline{\rho}(\epsilon) \lesssim R(\overline{\mathsf{W}}_p, \mathcal{G}_q, \epsilon)$ with probability $1 - \delta$. By the triangle inequality for $\overline{\mathsf{W}}_p$, we thus have $\overline{\mathsf{W}}_p(\nu, \mu) \lesssim R(\overline{\mathsf{W}}_p, \mathcal{G}_q, \epsilon) + \overline{\mathsf{W}}_p(\hat{\mu}_n, \mu)$ with probability $1 - \delta$. An analogous argument gives the corresponding result for $\underline{\mathsf{W}}_p$. $\qquad\square$

**Reduction to bounded support:** To prove Proposition 2, we provide a reduction from the general case to that of bounded support via Markov's inequality and a coupling argument.

**Lemma 16** (High probability norm bound). *If $X \sim \mu \in \mathcal{G}_q$, there exists $R \lesssim \delta^{-1/q}\sqrt{1 \vee d/q} \leq \delta^{-1/q}\sqrt{d}$ such that $\|X - \mu x\| \leq R$ with probability at least $1 - \delta$.*

*Proof.* Assume without loss of generality that $\mu x = 0$. We compute

$$\mu(\|x\|^q)^{1/q} \lesssim (1 \vee d/q)^{q/2} \sup_{\theta \in \mathbb{S}^{d-1}} \mu(|\theta^\intercal x|^q)^{1/q} \leq (1 \vee d/q)^{q/2},$$

where the first inequality uses Lemma 6 and the second uses $\mu \in \mathcal{G}_q$. Markov's inequality then gives the claim. $\qquad\square$

**Lemma 17** (Switch of base measure). *Fix $\mu \in \mathcal{P}(\mathbb{R}^d)$ and $A \subseteq \mathbb{R}^d$ with $\mu(A) \geq 1 - \epsilon$. Write $\mu_A$ for the distribution of $X \sim \mu$ conditioned on $X \in A$. Consider any random measure $\tilde{\mu}_n$ such that $\|\tilde{\mu}_n - \hat{\mu}_n\| \leq \epsilon'$ almost surely, where $n \geq 3 \log(1/\delta)/\epsilon$. Then there exists a coupling of $(\tilde{\mu}_n, \hat{\mu}_n)$ and $(\widehat{\mu_A})_n$ such that $\|\tilde{\mu}_n - (\widehat{\mu_A})_n\|_{\mathrm{TV}} \leq 2\epsilon + \epsilon'$ with probability at least $1 - \delta$.*

*Proof.* Given $n$ i.i.d. samples $X_1, \ldots, X_n$ from $\mu$, Lemma 17 and a Chernoff bound give that at least $(1 - 2\epsilon)n$ of them satisfy $X_i \in A$, with probability at least $1 - \delta$. Define the coupled set of samples $Y_1, \ldots, Y_n$ by $Y_i = X_i$ if $X_i \in A$ and $Y_i \sim \mu_A$ i.i.d. otherwise, and choose $(\widehat{\mu_A})_n$ as their empirical measure (by design, the marginal distribution of $Y_1, \ldots, Y_n$ coincides with $n$ samples from $\mu_A$). Under this coupling, we then have

$$\|\tilde{\mu}_n - (\widehat{\mu_A})_n\|_{\mathrm{TV}} \leq \|\tilde{\mu}_n - \hat{\mu}_n\|_{\mathrm{TV}} + \|\hat{\mu}_n - (\widehat{\mu_A})_n\|_{\mathrm{TV}} \leq 2\epsilon + \epsilon'$$

with probability at least $1 - \delta$. $\qquad\square$

*Proof of Proposition 2.* By Lemma 16, we have that for $X \sim \mu$, $\|X - \mu x\| \leq R \asymp \sqrt{d/\epsilon}$ with probability at least $1 - \epsilon/400$. Letting $A$ denote the ball of radius $R$ around $\mu x$ and applying Lemma 17 with failure probability 0.001, we can view $\tilde{\mu}_n$ as being a $\frac{201}{200}\epsilon$-corrupted version of $n$ i.i.d. samples from the conditional distribution $\mu_R$, with probability at least 0.999. Thus, applying the procedure from Proposition 14 with $R \asymp \sqrt{d/\epsilon}$, confidence probability 0.999 and corruption fraction $\frac{201}{200}\epsilon < 0.499$, we obtain $\nu$ with $\overline{W}_p(\nu, \mu_R) \lesssim R(\mathsf{D}, \mathcal{G}_q, \epsilon) + \mathsf{D}((\widehat{\mu_R})_n, \mu_R)$ with unconditional probability 0.998. By resilience of $\mu$ and the fact that $\|\mu - \mu_R\|_{\mathrm{TV}} \leq \epsilon/400$, the same recovery guarantees hold with base measure $\mu$. Finally, we bound $\mathsf{D}((\widehat{\mu_R})_n, \mu_R)$ by its expectation via Markov's inequality to obtain $\overline{W}_p(\nu, \mu) \lesssim R(\mathsf{D}, \mathcal{G}_q, \epsilon) + \mathbb{E}[\mathsf{D}((\widehat{\mu_R})_n, \mu_R)]$ with probability 0.99. $\quad\square$

### D.6 Proof of Proposition 3

For any $\mu, \nu \in \mathcal{P}_1(\mathbb{R}^d)$, we have $\overline{W}_p(\mu, \nu) \geq \|\mu x - \nu x\|$ (seen by taking $\theta$ in the direction of $\mu x - \nu x$). Hence, if $\overline{W}_p(\mu, \nu) \leq \rho$ for all $\nu \leq \frac{1}{1-\epsilon}\mu$, then $\mu$ is $(\rho, \epsilon)$-resilient in mean. (This direction holds for all $p \geq 1$). For the other direction, we mirror the proof of Theorem 2, first establishing a simple lemma.

**Lemma 18.** *Fix $X \sim \mu \in \mathcal{P}_1(\mathbb{R})$ and define the quantiles $\tau_\epsilon = \sup\{t \in \mathbb{R} : \Pr(X \geq t) \geq \epsilon\}$ and $\tilde{\tau}_\epsilon = \sup\{t \in \mathbb{R} : \Pr(|X| \geq t) \geq \epsilon\}$. Then, we have*

$$\mathbb{E}[|X| \mid |X| \geq \tilde{\tau}_\epsilon] \leq 4\,\mathbb{E}[X \mid X \geq \tau_\epsilon] \vee \mathbb{E}[-X \mid X \leq \tau_{1-\epsilon}]$$

Simply put, if $|X|$ has large tails, then one of $X$ or $-X$ must have a large tail.

*Proof.* Writing $X_+ = X \vee 0$ and $X_- = -X \vee 0$, we bound

$$
\begin{aligned}
\mathbb{E}\big[|X| \,\big|\, |X| \geq \tilde{\tau}_\epsilon\big] &= \mathbb{E}\big[X_+ \,\big|\, |X| \geq \tilde{\tau}_\epsilon\big] + \mathbb{E}\big[X_- \,\big|\, |X| \geq \tilde{\tau}_\epsilon\big] \\
&\leq \mathbb{E}[X_+ | X \geq \tau_\epsilon] + \mathbb{E}[X_- | X \leq \tau_{1-\epsilon}] \\
&\leq \mathbb{E}[X - (\tau_\epsilon \wedge 0) \mid X \geq \tau_\epsilon] + \mathbb{E}[-X + (\tau_{1-\epsilon} \vee 0) \mid X \leq \tau_{1-\epsilon}] \\
&= \mathbb{E}[X \mid X \geq \tau_\epsilon] + \mathbb{E}[-X \mid X \leq \tau_{1-\epsilon}] - (\tau_\epsilon \wedge 0) + (\tau_{1-\epsilon} \vee 0) \\
&\leq \mathbb{E}[X \mid X \geq \tau_\epsilon] + \mathbb{E}[-X \mid X \leq \tau_{1-\epsilon}] + (-\tau_{1-\epsilon} \vee 0) + (\tau_\epsilon \vee 0) \\
&\leq \mathbb{E}[X \mid X \geq \tau_\epsilon] + \mathbb{E}[-X \mid X \leq \tau_{1-\epsilon}] + (\mathbb{E}[-X \mid X \leq \tau_{1-\epsilon}] \vee 0) + (\mathbb{E}[X \mid X \geq \tau_\epsilon] \vee 0).
\end{aligned}
$$

Now, it is easy to check that each summand is bounded by $\mathbb{E}[X \mid X \geq \tau_\epsilon] \vee \mathbb{E}[-X \mid X \leq \tau_{1-\epsilon}]$ (since this maximum is non-negative), giving the lemma. $\quad\square$

Continuing, we take $\mu \in \mathcal{P}_1(\mathbb{R}^d)$ which is $(\rho, \epsilon)$-mean-resilient and assume without loss of generality that $\mu x = 0$. For all $\nu \leq \frac{1}{1-\epsilon}\mu$, we write $\mu = (1-\epsilon)\nu + \epsilon\alpha$ for $\alpha \in \mathcal{P}(\mathbb{R}^d)$ and bound

$$
\begin{aligned}
\overline{W}_p(\mu, \nu) = \overline{W}_p((1-\epsilon)\nu + \epsilon\alpha, \nu) && \\
&\leq \epsilon^{1/p}\overline{W}_p(\alpha, \nu) && \text{(Fact 2)} \\
&\leq \epsilon^{1/p}(\overline{W}_p(\alpha, \delta_0) + \overline{W}_p(\nu, \delta_0)) && \text{(triangle inequality)} \\
&\leq 2\epsilon^{1/p} \sup_{\kappa \leq \frac{1}{(1-\epsilon)\wedge\epsilon}\mu} \overline{W}_p(\kappa, \delta_0) && \\
&= 2\epsilon \sup_{\kappa \leq \frac{1}{(1-\epsilon)\wedge\epsilon}\mu} \sup_{\theta \in \mathbb{S}^{d-1}} \mathbb{E}_\kappa\left[|\theta^\mathsf{T} X|^p\right]^{1/p} && \\
&= 2\epsilon \sup_{\theta \in \mathbb{S}^{d-1}} \mathbb{E}_\kappa\left[|\theta^\mathsf{T} X|^p \,\big|\, |\theta^\mathsf{T} X| \geq \tilde{\tau}_{\epsilon\wedge(1-\epsilon)}(\theta)\right]^{1/p}.
\end{aligned}
$$

where $\tilde{\tau}_\epsilon(\theta) = \sup\{t \in \mathbb{R} : \mathbb{P}(|\theta^\mathsf{T} X| \geq t) \geq \epsilon\}$ for $X \sim \mu$. (Technically, the final inequality may fail if $\mu$ has a point mass at $\tilde{\tau}_{\epsilon\wedge(1-\epsilon)}(\theta)$; in this case, assume that ties are broken with independent randomness so that the conditioned event has probability $\epsilon \wedge (1 - \epsilon)$). From now on, we will use that $p = 1$. Writing $\tau_\epsilon(\theta) = \sup\{t \in \mathbb{R} : \mathbb{P}(\theta^\mathsf{T} X \geq t) \geq \epsilon\}$ and breaking ties in the same way, we apply Lemma 18 to bound

$$\overline{W}_1(\mu, \nu) \leq 8\epsilon \sup_{\theta \in \mathbb{S}^{d-1}} \mathbb{E}_\mu[\theta^\mathsf{T} X \mid \theta^\mathsf{T} X \geq \tau_{\epsilon\wedge(1-\epsilon)}(\theta)]$$

$$= 8\epsilon \sup_{\theta \in \mathbb{S}^{d-1}} \theta^{\mathsf{T}} \mathbb{E}_{\mu}[X \mid \theta^{\mathsf{T}} X \geq \tau_{\epsilon \wedge (1-\epsilon)}(\theta)]$$

$$= 8\epsilon \left\| \mathbb{E}_{\mu}[X \mid \theta^{\mathsf{T}} X \geq \tau_{\epsilon \wedge (1-\epsilon)}(\theta)] \right\|$$

$$= 8\epsilon \left\| \mathbb{E}_{\mu} X - \mathbb{E}_{\mu}[X \mid \theta^{\mathsf{T}} X \geq \tau_{\epsilon \wedge (1-\epsilon)}(\theta)] \right\|.$$

Now, if $\epsilon \geq 1/2$, we can use resilience of $\mu$ to bound

$$\overline{\mathsf{W}}_1(\mu, \nu) \leq 8\epsilon\rho \leq 8\rho.$$

Otherwise, writing $E$ for the event that $\theta^{\mathsf{T}} X \geq \tau_{\epsilon}(\theta)$, we have

$$\begin{aligned}
\overline{\mathsf{W}}_1(\mu, \nu) &\leq 8\epsilon \left\| \epsilon \mathbb{E}_{\mu}[X|E] + (1-\epsilon)\mathbb{E}_{\mu}[X|E^c] - \mathbb{E}_{\mu}[X|E] \right\| \\
&= 8\epsilon(1-\epsilon) \left\| \mathbb{E}_{\mu}[X|E^c] - \mathbb{E}_{\mu}[X|E] \right\| \\
&= 8\epsilon(1-\epsilon) \left\| \mathbb{E}_{\mu}[X|E^c] - \epsilon^{-1}(\mathbb{E}_{\mu}[X] - (1-\epsilon)\mathbb{E}_{\mu}[X|E^c]) \right\| \\
&= 8(1-\epsilon) \left\| \mathbb{E}_{\mu}[X] - \mathbb{E}_{\mu}[X|E^c] \right\| \\
&\leq 8(1-\epsilon)\rho \leq 8\rho.
\end{aligned}$$

Hence, $\mu$ is $(8\rho, \epsilon)$-resilient w.r.t. $\overline{\mathsf{W}}_1$.

Immediately, this allows $\overline{\mathsf{W}}_1$ to inherit a multitude of (population-limit and finite-sample) risk bounds from the robust mean estimation literature. See [54] for a detailed survey of robust statistics results based on resiliency. For example, $\mu \in \mathcal{G}_q$ is known to be $(O(\epsilon^{1-1/q}), \epsilon)$-mean-resilient, immediately implying Theorem 2 for $\overline{\mathsf{W}}_1$.

### D.7 Proof of Proposition 4

When $q = 2$, we mirror the approach of Proposition 2 but perform projection onto the space of distributions with bounded covariance, instead of onto the space of resilient distributions. We require the following standard result (see, e.g., Lemma A.18 of [19]), establishing finite-sample covariance bounds under bounded support.

**Lemma 19.** *Let $\mu \in \mathcal{P}(\mathbb{R}^d)$ with $\|\Sigma_{\mu}\|_{\mathrm{op}} \leq \sigma^2$ and $\mathrm{diam}(\mathrm{spt}(\mu)) \leq R$. Then the empirical distribution $\hat{\mu}_n$ satisfies $\|\Sigma_{\hat{\mu}_n}\|_{\mathrm{op}} \lesssim \sigma^2$ with probability at least 0.999 for $n \gtrsim R^2 \log(d)$.*

Importantly, there are efficient filtering algorithms for projecting onto the set of distributions with bounded covariance (see, e.g., Theorem 3.1 [24]).

**Lemma 20** (Spectral reweighting). *Let $x_1, \ldots, x_n \in \mathbb{R}^d$ and $0 < \epsilon \leq 1/10$. Suppose the discrete measure $\mu_n = \frac{1}{n} \sum_{i=1}^n \delta_{x_i}$ admits an $\epsilon$-deletion $\nu_n \leq \frac{1}{1-\epsilon} \mu_n$ such that $\|\Sigma_{\mu_n}\|_{\mathrm{op}} \leq \sigma^2$. Then, given $\{x_i\}_{i=1}^n$ and $\epsilon$, there is an algorithm which finds $\nu \leq \frac{1}{1-3\epsilon} \mu_n$ such that $\|\Sigma_{\nu}\|_{\mathrm{op}} \lesssim \sigma^2$ with probability 0.999, in time $\widetilde{O}(nd^2)$.*

Combining, we prove the proposition. We remark that sample complexity is dominated by empirical convergence under $\mathsf{D} \in \{\underline{\mathsf{W}}_p, \overline{\mathsf{W}}_p\}$ of the truncated version of a distribution with bounded second moments. This can be improved significantly in many cases of interest, for example under log-concavity of the clean distribution.

*Proof of Proposition 4.* First, we consider the case of bounded support, when $\mathrm{diam}(\mathrm{spt}(\mu)) \leq R$, and with contamination fraction $\epsilon \in [0, 1/10]$. We mirror the argument of Proposition 14, but project onto the set of distributions with bounded covariance using spectral reweighting. Write $\tilde{\mu}_n$ for the empirical distribution of the $\epsilon$-contaminated samples and $\hat{\mu}_n$ for that of the clean samples, with $\hat{\mu}_n \leq \frac{1}{1-\epsilon} \tilde{\mu}_n$. Combining Lemmas 19 and 20, we find that $\|\Sigma_{\hat{\mu}_n}\|_{\mathrm{op}} \lesssim \sigma^2$ and that the spectral reweighting algorithm returns $\nu \leq \frac{1}{1-3\epsilon} \hat{\mu}_n$ with $\|\Sigma_{\nu}\|_{\mathrm{op}} \lesssim \sigma^2$ in time $\widetilde{O}(nd^2)$, all with probability 0.998. By resilience of the class $\mathcal{G}_2(\sigma)$ w.r.t. $\mathsf{D}$ and Markov's inequality, we have

$$\mathsf{D}(\nu, \mu) \leq \mathsf{D}(\nu, \hat{\mu}_n) + \mathsf{D}(\hat{\mu}_n, \mu) \lesssim R(\mathsf{D}, \mathcal{G}_q(\sigma), \epsilon) + \mathbb{E}[\mathsf{D}(\hat{\mu}_n, \mu)]$$

with probability 0.995. For the unbounded case, we apply Lemma 16 and Lemma 17 as in the proof of Proposition 2 to reduce to $R \asymp \sqrt{d/\epsilon}$ and obtain the desired error bound with probability at least 0.99, so long as $0 < \epsilon \leq 1/12$ (any constant separated from 1/10 will do). $\qquad\square$

### D.8 Proof of Lemma 1

We start by showing that the Lipschitz constant of $w_p$ is upper bounded by $L_{\mu,\nu}^p$. Fix $\theta_1, \theta_2 \in \mathbb{S}^{d-1}$ and observe that

$$
\begin{aligned}
\left| w_p(\theta_1) - w_p(\theta_2) \right| &= \left| \mathsf{W}_p\big(\mathsf{p}_\sharp^{\theta_1}\mu, \mathsf{p}_\sharp^{\theta_1}\nu\big) - \mathsf{W}_p\big(\mathsf{p}_\sharp^{\theta_2}\mu, \mathsf{p}_\sharp^{\theta_2}\nu\big) \right| \\
&\leq \mathsf{W}_p\big(\mathsf{p}_\sharp^{\theta_1}\mu, \mathsf{p}_\sharp^{\theta_2}\mu\big) + \mathsf{W}_p\big(\mathsf{p}_\sharp^{\theta_1}\nu, \mathsf{p}_\sharp^{\theta_2}\nu\big) \\
&\leq \|\theta_1 - \theta_2\| \sup_{\theta \in \mathbb{S}^{d-1}} \left( \big(\mu|\theta^\mathsf{T} x|^p\big)^{1/p} + \big(\nu|\theta^\mathsf{T} x|^p\big)^{1/p} \right),
\end{aligned}
$$

where the last step uses the optimal transportation cost formulation of $\mathsf{W}_p$. The RHS above is $L_{\mu,\nu}^p$ from the lemma, which concludes the proof of the first statement.

Next, we bound the Lipschitz constant of $w_p^p$. For $\theta_1, \theta_2 \in \mathbb{S}^{d-1}$ and $i = 1, 2$, let $(X_i, Y_i)$ be a coupling of $\mu$ and $\nu$ so that $(\theta_i^\mathsf{T} X_i, \theta_i^\mathsf{T} Y_i)$ is optimal for $\mathsf{W}_p\big(\mathsf{p}_\sharp^{\theta_i}\mu, \mathsf{p}_\sharp^{\theta_i}\nu\big)$. These couplings are constructed as follows. For $i = 1, 2$, let $(U_i, V_i)$ be an optimal couplings for $\mathsf{W}_p\big(\mathsf{p}_\sharp^{\theta_i}\mu, \mathsf{p}_\sharp^{\theta_i}\nu\big)$. Take $\mathrm{P}_i \in \mathbb{R}^{d \times d}$ as a unitary matrix whose first row is $\theta_i$, and let $\mathrm{P}_{i,-1} \in \mathbb{R}^{(d-1) \times d}$ denote the matrix obtained by deleting the first row of $\mathrm{P}_i$. Given $u_1, u_2 \in \mathbb{R}$ generate the random variables $W_i(u_i) \sim \mathcal{L}\big(P_{i,-1}X \big| \theta_i^\mathsf{T} X = u_i\big)$, for $i = 1, 2$, where $X \sim \mu$ and $\mathcal{L}(\cdot)$ designates the probability law of a random variable. Setting $\bar{U}_i := \big(U_i, W_i(U_i)\big)$ for $i = 1, 2$, observe that $\bar{U}_i \sim \mathcal{L}\big(P_i X\big)$ and further that $X_i := \mathrm{P}_i^\mathsf{T} \bar{U}_i \sim \mu$. Constructing $\bar{V}_i$, for $i = 1, 2$, in an analogous fashion but with $\nu$ in place of $\mu$, and defining $Y_i$ similarly to $X_i$ above, we obtain the desired $(X_i, Y_i)$ couplings.

Then by optimality of the couplings, we have

$$
\begin{aligned}
w_p^p(\theta_1) - w_p^p(\theta_2) &\leq \mathbb{E}\big[ \big| \theta_1^\mathsf{T}(X_2 - Y_2) \big|^p - \big| \theta_2^\mathsf{T}(X_2 - Y_2) \big|^p \big], \\
w_p^p(\theta_2) - w_p^p(\theta_1) &\leq \mathbb{E}\big[ \big| \theta_2^\mathsf{T}(X_1 - Y_1) \big|^p - \big| \theta_1^\mathsf{T}(X_1 - Y_1) \big|^p \big].
\end{aligned}
$$

Combining these bounds, we obtain

$$
\begin{aligned}
|w_p^p(\theta_1) - w_p^p(\theta_2)| &\leq p\|\theta_1 - \theta_2\| \, \mathbb{E}\left[ \max_{i=1,2} \left| \frac{(\theta_1 - \theta_2)^\mathsf{T}(X_i - Y_i)}{\|\theta_1 - \theta_2\|} \right| \cdot \max_{i,j=1,2} \big| \theta_i^\mathsf{T}(X_j - Y_j) \big|^{p-1} \right] \\
&\leq p\|\theta_1 - \theta_2\| \, \mathbb{E}\left[ \max_{\substack{i=1,2 \\ j=1,2,3}} \big| \theta_j'(X_i - Y_i) \big|^p \right] \\
&\leq 3p2^p \|\theta_1 - \theta_2\| \sup_{\theta \in \mathbb{S}^{d-1}} \mathbb{E}\big[ \big| \theta^\mathsf{T} X_1 \big|^p + \big| \theta^\mathsf{T} Y_1 \big|^p \big],
\end{aligned}
$$

where for the second inequality we have defined $\theta_3 := \frac{\theta_1 - \theta_2}{\|\theta_1 - \theta_2\|}$. This concludes the proof.

**Remark 12** (Alternative Lipschitz constants). *The Lipschitz constant for $w_p^p$ can be alternatively derived as*

$$
\begin{aligned}
|w_p^p(\theta_1) - w_p^p(\theta_2)| &\leq p\|\theta_1 - \theta_2\| \, \mathbb{E}\left[ \max_{i=1,2} \left| \frac{(\theta_1 - \theta_2)^\mathsf{T}(X_i - Y_i)}{\|\theta_1 - \theta_2\|} \right| \cdot \max_{i,j=1,2} \big| \theta_i^\mathsf{T}(X_j - Y_j) \big|^{p-1} \right] \\
&\leq p\|\theta_1 - \theta_2\| \, \mathbb{E}\left[ \max_{\substack{i=1,2 \\ j=1,2,3}} \big| \theta_j'(X_i - Y_i) \big|^p \right] \\
&\lesssim_p \|\theta_1 - \theta_2\| \left( \|\mu x - \nu x\| + \sup_{\theta \in \mathbb{S}^{d-1}} \mathbb{E}\big[ \big| \theta^\mathsf{T}(X_1 - \mu x) \big|^p + \big| \theta^\mathsf{T}(Y_1 - \nu x) \big|^p \big] \right),
\end{aligned}
$$

*where the terms corresponding to mean difference and covariance are separated.*

### D.9 Proof of Proposition 5

We decompose the error by introducing the Monte Carlo average for the population projected distances:

$$
\mathbb{E}\left[ \left| \widehat{\underline{\mathsf{W}}}_{\mathsf{MC}}^p - \underline{\mathsf{W}}_p^p(\mu, \nu) \right| \right]
$$

$$\leq \mathbb{E}\left[\left\| \widehat{\mathsf{W}}^p_{\mathsf{MC}} - \frac{1}{m}\sum_{i=1}^m \mathsf{W}^p_p\big(\mathfrak{p}^{\Theta_i}_\sharp \mu, \mathfrak{p}^{\Theta_i}_\sharp \nu\big)\right\|\right] + \mathbb{E}\left[\left\| \frac{1}{m}\sum_{i=1}^m \mathsf{W}^p_p\big(\mathfrak{p}^{\Theta_i}_\sharp \mu, \mathfrak{p}^{\Theta_i}_\sharp \nu\big) - \underline{\mathsf{W}}^p_p(\mu,\nu)\right\|\right]\Bigg\}.$$

$$(15)$$

For the first term, using the fact that $\Theta_1,\dots,\Theta_n$ are i.i.d., we have

$$\mathbb{E}\left[\left\| \widehat{\mathsf{W}}^p_{\mathsf{MC}} - \frac{1}{m}\sum_{i=1}^m \mathsf{W}^p_p\big(\mathfrak{p}^{\Theta_i}_\sharp \mu, \mathfrak{p}^{\Theta_i}_\sharp \nu\big)\right\|\right] \leq \mathbb{E}\left\{ \mathbb{E}\left[ \left| \mathsf{W}^p_p\big(\mathfrak{p}^{\Theta}_\sharp \hat{\mu}_n, \mathfrak{p}^{\Theta}_\sharp \hat{\nu}_n\big) - \mathsf{W}^p_p\big(\mathfrak{p}^{\Theta}_\sharp \mu, \mathfrak{p}^{\Theta}_\sharp \nu\big) \right| \,\Big|\, \Theta \right]\right\}.$$

$$(16)$$

Denote $(f\oplus g)(x,y) = f(x) + g(y)$, and let $c(x,y) = \|x-y\|^2$. Further, define the c-conjugate of a function $f$ as $f^c(y) = \inf_x c(x,y) - f(x)$. For each $\theta \in \mathbb{S}^{d-1}$, observe that

$$\mathsf{W}^p_p\big(\mathfrak{p}^\theta_\sharp \hat{\mu}_n, \mathfrak{p}^\theta_\sharp \hat{\nu}_n\big) - \mathsf{W}^p_p\big(\mathfrak{p}^\theta_\sharp \mu, \mathfrak{p}^\theta_\sharp \nu\big)$$

$$\leq \sup_{\substack{(\varphi,\psi)\in L^1(\mu)\times L^1(\nu):\\ \varphi\oplus\psi\leq c}} \big\{ (\mathfrak{p}^\theta_\sharp \hat{\mu}_n)\varphi + (\mathfrak{p}^\theta_\sharp \hat{\nu}_n)\psi \big\} - \sup_{\substack{(f,g)\in L^1(\mu)\times L^1(\nu):\\ f\oplus g\leq c}} \big\{ (\mathfrak{p}^\theta_\sharp \mu)f + (\mathfrak{p}^\theta_\sharp \nu)g \big\}$$

$$\leq \sup_{(\varphi,\psi)\in L^1(\mu)\times L^1(\nu)} \big\{ (\mathfrak{p}^\theta_\sharp \hat{\mu}_n)\varphi + (\mathfrak{p}^\theta_\sharp \mu)\varphi^c + (\mathfrak{p}^\theta_\sharp \hat{\nu}_n)\psi + (\mathfrak{p}^\theta_\sharp \nu)\psi^c \big\}$$

$$= \mathsf{W}^p_p\big(\mathfrak{p}^\theta_\sharp \hat{\mu}_n, \mathfrak{p}^\theta_\sharp \mu\big) + \mathsf{W}^p_p\big(\mathfrak{p}^\theta_\sharp \hat{\nu}_n, \mathfrak{p}^\theta_\sharp \nu\big).$$

Repeating this argument for $\mathsf{W}^p_p\big(\mathfrak{p}^\theta_\sharp \mu, \mathfrak{p}^\theta_\sharp \nu\big) - \mathsf{W}^p_p\big(\mathfrak{p}^\Theta_\sharp \hat{\mu}_n, \mathfrak{p}^\Theta_\sharp \hat{\nu}_n\big)$ we obtain

$$\left| \mathsf{W}^p_p\big(\mathfrak{p}^\theta_\sharp \mu, \mathfrak{p}^\theta_\sharp \nu\big) - \mathsf{W}^p_p\big(\mathfrak{p}^\theta_\sharp \hat{\mu}_n, \mathfrak{p}^\theta_\sharp \hat{\nu}_n\big) \right| \leq \mathsf{W}^p_p\big(\mathfrak{p}^\theta_\sharp \hat{\mu}_n, \mathfrak{p}^\theta_\sharp \mu\big) + \mathsf{W}^p_p\big(\mathfrak{p}^\theta_\sharp \hat{\nu}_n, \mathfrak{p}^\theta_\sharp \nu\big).$$

The proof of Theorem 1 implies that, for any $\theta \in \mathbb{S}^{d-1}$,

$$\mathbb{E}\big[\mathsf{W}^p_p\big(\mathfrak{p}^\theta_\sharp \hat{\mu}_n, \mathfrak{p}^\theta_\sharp \mu\big)\big] \leq C_p \frac{(\log n)^{\mathbb{1}_{\{p=2\}}} \|\Sigma_\mu\|^{p/2}_{\mathrm{op}}}{n^{(p\wedge 2)/2}},$$

$$\mathbb{E}\big[\mathsf{W}^p_p\big(\mathfrak{p}^\theta_\sharp \hat{\nu}_n, \mathfrak{p}^\theta_\sharp \nu\big)\big] \leq C_p \frac{(\log n)^{\mathbb{1}_{\{p=2\}}} \|\Sigma_\nu\|^{p/2}_{\mathrm{op}}}{n^{(p\wedge 2)/2}}.$$

Inserting this back into (16), we have

$$\mathbb{E}\left[\left\| \widehat{\mathsf{W}}^p_{\mathsf{MC}} - \frac{1}{m}\sum_{i=1}^m \mathsf{W}^p_p\big(\mathfrak{p}^{\Theta_i}_\sharp \mu, \mathfrak{p}^{\Theta_i}_\sharp \nu\big)\right\|\right] \leq \frac{C_p\big(\|\Sigma_\nu\|^{p/2}_{\mathrm{op}} + \|\Sigma_\mu\|^{p/2}_{\mathrm{op}}\big)(\log n)^{\mathbb{1}_{\{p=2\}}}}{n^{(p\wedge 2)/2}}$$

$$(17)$$

For the second term in (15), recall that $w^p_p(\theta) := \mathsf{W}^p_p\big(\mathfrak{p}^\theta_\sharp \mu, \mathfrak{p}^\theta_\sharp \nu\big)$ for $\theta \in \mathbb{S}^{d-1}$, and bound

$$\mathbb{E}\left[\left\| \frac{1}{m}\sum_{i=1}^m \mathsf{W}^p_p\big(\mathfrak{p}^{\Theta_i}_\sharp \mu, \mathfrak{p}^{\Theta_i}_\sharp \nu\big) - \underline{\mathsf{W}}^p_p(\mu,\nu)\right\|\right] \leq \sqrt{\frac{1}{m}\mathrm{Var}\big(w^p_p(\Theta)\big)}$$

To control the variance we use concentration of Lipschitz functions on the unit sphere. By Remark 12 following the proof of Lemma 1, $w^p_p$ is $\tilde{M}^p_{\mu,\nu}$-Lipschitz, with $\tilde{M}^p_{\mu,\nu} \lesssim_p \|\mu x - \nu x\|^p + \sup_{\theta\in\mathbb{S}^{d-1}}(\mu|\theta^\mathsf{T}(x-\mu x)|^p + \nu|\theta^\mathsf{T}(x-\nu x)|^p)$. Denoting the median by $\mathrm{med}(\cdot)$, we have for $d \geq 3$ (cf. e.g., [32, Chapter 1])

$$\mathbb{P}\Big( \big| w^p_p(\Theta) - \mathrm{med}\big(w^p_p(\theta)\big) \big| \geq t \Big) \leq 8\exp\left( -\frac{(d-2)t^2}{2(\tilde{M}^p_{\mu,\nu})^2} \right).$$

Consequently,

$$\mathrm{Var}\big(w^p_p(\Theta)\big) \leq \mathbb{E}\Big[\big(w^p_p(\Theta) - \mathrm{med}\big(w^p_p(\Theta)\big)\big)^2\Big]$$

$$= \int_0^\infty \mathbb{P}\Big( \big| w^p_p(\Theta) - \mathrm{med}\big(w^p_p(\Theta)\big) \big| \geq \sqrt{t} \Big)\, dt$$

$$\leq \frac{16(\tilde{M}_{\mu,\nu}^p)^2}{d-2}.$$

Alternatively, for $d \leq 2$, letting $\Theta, \Theta'$ be independent samples drawn uniformly from $\mathbb{S}^{d-1}$, we have

$$\mathrm{Var}\big(w_p^p(\Theta)\big) = \frac{1}{2}\mathbb{E}\big[\big|w_p^p(\Theta) - w_p^p(\Theta')\big|^2\big] \leq \frac{(\tilde{M}_{\mu,\nu}^p)^2}{2}\mathbb{E}\big[|\Theta - \Theta'|^2\big] \leq (\tilde{M}_{\mu,\nu}^p)^2.$$

Combining the two variance bounds, for any $d \geq 1$, we obtain

$$\mathbb{E}\left[\left\|\frac{1}{m}\sum_{i=1}^m \mathsf{W}_p^p\big(\mathfrak{p}_\sharp^{\Theta_i}\mu, \mathfrak{p}_\sharp^{\Theta_i}\nu\big) - \underline{\mathsf{W}}_p^p(\mu,\nu)\right\|\right] \lesssim \frac{4\tilde{M}_{\mu,\nu}^p}{\sqrt{md}}, \tag{18}$$

where the hidden constant is universal.

We now focus on bounding $\tilde{M}_{\mu,\nu}^p$, leveraging log-concavity of $\mu$ and $\nu$. We present the derivation for $\sup_\theta \mu|\theta^\mathsf{T}(x - \mu x)|$; the one corresponding to $\nu$ is analogous. To control this term we use exponential concentration for 1-Lipschitz functions of log-concave random variables. Recalling that $\mu$ and $\nu$ being log-concave implies that so are $\mathfrak{p}_\sharp^\theta \mu$ and $\mathfrak{p}_\sharp^\theta \nu$, Theorem 1.2 in [39] yields

$$\mathbb{P}\big(\big|\theta^\mathsf{T}X - \mu(\theta^\mathsf{T}x)\big| > t\big) \leq e\exp(-D_\mu t),$$

where $X \sim \mu$ and $D_\mu \geq c/\sqrt{\|\Sigma_\mu\|_{\mathrm{op}}}$, with a universal constant $c$. Then,

$$\begin{aligned}
\sup_{\theta \in \mathbb{S}^{d-1}} \mu|\theta^\mathsf{T}(x - \mu x)|^p &\leq \int_0^\infty \mathbb{P}\big(\big|\theta^\mathsf{T}x - \mu(\theta^\mathsf{T}x)\big|^p > t\big)dt \\
&\leq \int_0^\infty e\exp(-D_\mu t^{1/p})\,dt \\
&= \Gamma(p+1)D_\mu^{-p} \\
&\leq \Gamma(p+1)\left(\frac{\sqrt{\|\Sigma_\mu\|_{\mathrm{op}}}}{c}\right)^p \\
&\leq C_p\|\Sigma_\nu\|_{\mathrm{op}}^{p/2},
\end{aligned}$$

for a constant $C_p$ depending only on $p$. Similarly, we obtain

$$\sup_{\theta \in \mathbb{S}^{d-1}} \nu|\theta^\mathsf{T}(x - \nu x)|^p \leq C_p\|\Sigma_\nu\|_{\mathrm{op}}^{p/2},$$

which together implies

$$\tilde{M}_{\mu,\nu}^p \leq C_p'\left(\|\mu x - \nu x\|^p + \|\Sigma_\mu\|_{\mathrm{op}}^{p/2} + \|\Sigma_\nu\|_{\mathrm{op}}^{p/2}\right).$$

Inserting the above bound into (18) and combining with (17) yields the result.

### D.10 Proof of Proposition 6

Observe that $\tilde{w}_2^2(\theta)$ is $M_n$-Lipschitz by Lemma 1 and $\rho_n$-weakly convex by Lemma 2.2 in [34], where $M_n = 4\sup_\theta(\hat{\mu}_n|\theta^\mathsf{T}x|^2 + \hat{\nu}_n|\theta^\mathsf{T}x|^2)$ and $\rho_n = 2\max_{i,j}\|X_i - Y_j\|^2$. By equation (2.10) in [14], there exists a choice of step sizes $\alpha_t = \frac{c_{\rho_n, M_n}}{\sqrt{t+1}}$, such that Algorithm 1 for the objective $\varphi(\theta) = \tilde{w}_2^2 + \delta_{\mathbb{B}^d}$, where $\delta_{\mathbb{B}^d} = \infty\mathbb{1}_{(\mathbb{B}^d)^c}$, outputs a point $\theta_{t^*}$ that is close to a near-stationary point $\theta^*$, in the sense that $\mathbb{E}_{t^*}[\|\theta^* - \theta_{t^*}\|] \leq \frac{\epsilon}{2\rho_n}$ and $\mathrm{dist}\big(0, \partial\tilde{w}_2^2(\theta^*)\big) \leq \epsilon$, in number of steps

$$T \leq \left\lceil \frac{64\rho_n^2 M_n^2\big(1 \wedge \frac{M_n}{2\rho_n}\big)}{\epsilon^4}\right\rceil.$$

We derive high probability upper bounds on $M_n$ and $\rho_n$ to obtain a non-stochastic bound on the computational complexity of our algorithm.

We will first reduce our problem to the case where $\mu$ and $\nu$ are isotropic log-concave, where our assumptions will lead to concentration inequalities on the above quantities. Assume first that $\Sigma_\mu$

and $\Sigma_\nu$ have rank $d$. Let $T^\mu(x) = \Sigma_\mu^{-1/2}(x - \mu x)$, and define $T^\nu$ analogously. Then, $\tilde{\mu} = T^\mu_\sharp \mu$ and $\tilde{\nu} = T^\nu_\sharp \nu$ are isotropic log-concave. Let $\tilde{\mu}_n$ and $\tilde{\nu}_n$ be empirical measures corresponding to $\tilde{\mu}$ and $\tilde{\nu}$, obtained by applying $T_\mu$ and $T_\nu$ to samples from $\mu$ and $\nu$, respectively. For the first, we have

$$M_n \leq M_{\mu,\nu}^2 + 24 \sup_{\theta \in \mathbb{S}^{d-1}} |\hat{\mu}_n \theta^\mathsf{T}(x - \mu x)|^2 + 24 \sup_{\theta \in \mathbb{S}^{d-1}} \hat{\nu}_n |\theta^\mathsf{T}(x - \nu x)|^2, \quad \text{and}$$

$$\rho_n \leq 6\Big( \|\mu x - \nu x\|^2 + \|\Sigma_\mu\|_{\text{op}} \max_i \|\Sigma_\mu^{-1/2}(X_i - \mu x)\|^2 + \|\Sigma_\nu\|_{\text{op}} \max_j \|\Sigma_\nu^{-1/2}(Y_j - \nu x)\|^2 \Big).$$

Further, assuming that $\mu$ and $\nu$ are centered, we have

$$\sup_{\theta \in \mathbb{S}^{d-1}} \left| \hat{\mu}_n |\theta^\mathsf{T} x|^2 - \mu |\theta^\mathsf{T} x|^2 \right|$$

$$\leq \sup_{\theta \in \mathbb{S}^{d-1}} \left| \hat{\mu}_n |\theta^\mathsf{T}_{\Sigma_\mu} x|^2 - \mu |\theta^\mathsf{T} x|^2 \right| \qquad \left[ \theta_{\Sigma_\mu} = \frac{\Sigma_\mu^{-1/2}\theta}{\|\Sigma_\mu^{-1/2}\theta\|} \right]$$

$$\leq \|\Sigma_\mu\|_{\text{op}} \sup_{\theta \in \mathbb{S}^{d-1}} \left| \tilde{\mu}_n |\theta^\mathsf{T} x|^2 - \tilde{\mu} |\theta^\mathsf{T} x|^2 \right|,$$

and similarly,

$$\sup_{\theta \in \mathbb{S}^{d-1}} \left| \hat{\nu}_n |\theta^\mathsf{T} x|^2 - \nu |\theta^\mathsf{T} x|^2 \right| \leq \|\Sigma_\nu\|_{\text{op}} \sup_{\theta \in \mathbb{S}^{d-1}} \left| \tilde{\nu}_n |\theta^\mathsf{T} x|^2 - \tilde{\nu} |\theta^\mathsf{T} x|^2 \right|.$$

For isotropic $\tilde{\mu}$ and $\tilde{\nu}$, we have (cf. Theorem 4.2 in [2])

$$\mathbb{P}\left( \sup_{\theta \in \mathbb{S}^{d-1}} \left| \tilde{\mu}_n |\theta^\mathsf{T} x|^2 - \tilde{\mu} |\theta^\mathsf{T} x|^2 \right| \leq \epsilon \right) \geq 1 - \exp\left( -cn^{1/4}\epsilon\sqrt{d} \right).$$

Choosing $\epsilon = 1/c$ above and noting that $\tilde{\mu}|\theta^\mathsf{T} x|^2 = \tilde{\nu}|\theta^\mathsf{T} x|^2 = 1$, we have

$$M_n \leq M_{\mu,\nu}^2 + 4(1 + 1/c)\Big( \|\Sigma_\mu\|_{\text{op}} + \|\Sigma_\nu\|_{\text{op}} \Big) \tag{19}$$

with probability at least $1 - \frac{2}{n}$. Additionally, by Lemma 3.1 in [2], if $d \geq \big( \log n \big)^2$, there exists a universal constant $C > 0$ such that

$$\max\left\{ \max_i \|\Sigma_\mu^{-1/2}(X_i - \mu x)\|^2, \max_i \|\Sigma_\mu^{-1/2}(X_i - \mu x)\|^2 \right\} \leq Cd,$$

implying

$$\rho_n \leq 6\Big( \|\mu x - \nu x\|^2 + Cd\left( \|\Sigma_\mu\|_{\text{op}} + \|\Sigma_\nu\|_{\text{op}} \right) \Big) \tag{20}$$

with probability at least $1 - \frac{2}{n}$.

If $\Sigma_\mu$ and $\Sigma_\nu$ are not full rank, then the above results hold for $\mu * \text{Unif}(B_d(0, \sigma))$ and $\nu * \text{Unif}(B_d(0, \sigma))$ instead, which are log-concave measures with covariance matrices $\Sigma_\mu + \sigma^2 I_d/(d+1)$ and $\Sigma_\nu + \sigma^2 I_d/(d+1)$, respectively. Letting $M_n^\sigma$, $\rho_n^\sigma$ and $M_{\mu,\nu}^{2;\sigma}$ denote $M_n$, $\rho_n$ and $M_{\mu,\nu}^2$ for these perturbed measures, we observe that $|\rho_n - \rho| \leq \sigma^2$, $|M_n^\sigma - M_n| \leq 96\sigma^2$, and $|M_{\mu,\nu}^{2,\sigma} - M_{\mu,\nu}^2| \leq 48\sigma^2$. Choosing $\sigma^2 = \|\Sigma_\mu\|_{\text{op}} + \|\Sigma_\nu\|_{\text{op}}$, we see that (19) and (20) hold for non-full dimensional $\mu$ and $\nu$ as well with adjustments to $c$ and $C$.

Combining (19) and (20), and noting that sorting to obtain the optimal permutation $\sigma^*$ and computing the subdifferential $\partial\rho(\sigma^*, \theta) = \nabla_\theta \rho(\sigma^*, \theta)$ takes $O(n \log n)$ operations, we have the result. $\qquad \square$

### D.11 Proof of Proposition 8

By Lemma 1, we have that $\hat{w}_p(\theta)$ is $\hat{L}_n$-Lipschitz with $\hat{L}_n := \sup_{\theta \in \mathbb{S}^{d-1}} \big\{ (\hat{\mu}_n |\theta^\mathsf{T} x|^p)^{1/p} + (\hat{\nu}_n |\theta^\mathsf{T} x|^p)^{1/p} \big\}$. This yields, via the LIPO convergence guarantee (Corollary 13 in [36]), that

$$\max_{\theta \in \mathbb{S}^{d-1}} \hat{w}_p(\theta) - \max_{1 \leq i \leq k} \hat{w}_p(\Theta_i) \leq 2\hat{L}_n \left( \frac{\log(1/\delta)}{k} \right)^{1/d} \tag{21}$$

with probability at least $1 - \delta$.

As in the previous section, we will first reduce our problem to the case where $\mu$ and $\nu$ are isotropic log-concave. For $1 \leq p \leq 2$, $\hat{L}_n \leq \sup_{\theta \in \mathbb{S}^{d-1}} \{(\hat{\mu}_n |\theta^\mathsf{T} x|^2)^{1/2} + (\hat{\nu}_n |\theta^\mathsf{T} x|^2)^{1/2}\}$, so that it suffices to bound $\hat{L}_n$ for $p \geq 2$. We have

$$
\begin{aligned}
\hat{L}_n &= \sup_{\theta \in \mathbb{S}^{d-1}} \left\{ (\hat{\mu}_n |\theta^\mathsf{T} x|^p)^{1/p} + (\hat{\nu}_n |\theta^\mathsf{T} x|^p)^{1/p} \right\} \\
&\leq \sup_{\theta \in \mathbb{S}^{d-1}} \left\{ (\hat{\mu}_n |\theta^\mathsf{T} (x - \mu x)|^p)^{1/p} + (\hat{\nu}_n |\theta^\mathsf{T} (x - \nu x)|^p)^{1/p} \right\} \\
&\quad + \sup_\theta |\mu(\theta^\mathsf{T} x)| + \sup_\theta |\nu(\theta^\mathsf{T} x)| \\
&\leq \|\Sigma_\mu\|_{\mathrm{op}}^{1/2} \left[ \sup_{\theta \in \mathbb{S}^{d-1}} (\tilde{\mu} |\theta^\mathsf{T} x|^p)^{1/p} + \sup_{\theta \in \mathbb{S}^{d-1}} \left| (\tilde{\mu}_n |\theta^\mathsf{T} x|^p)^{1/p} - (|\tilde{\mu}(\theta^\mathsf{T} x)|^p)^{1/p} \right| \right] \\
&\quad + \|\Sigma_\nu\|_{\mathrm{op}}^{1/2} \left[ \sup_{\theta \in \mathbb{S}^{d-1}} (\tilde{\nu} |\theta^\mathsf{T} x|^p)^{1/p} + \sup_{\theta \in \mathbb{S}^{d-1}} \left| (\tilde{\nu}_n |\theta^\mathsf{T} x|^p)^{1/p} - (|\tilde{\nu}(\theta^\mathsf{T} x)|^p)^{1/p} \right| \right] \\
&\quad + \sup_\theta |\mu(\theta^\mathsf{T} x)| + \sup_\theta |\nu(\theta^\mathsf{T} x)| \\
&\leq \|\Sigma_\mu\|_{\mathrm{op}}^{1/2} \left[ (2p)^{1/p} + \sup_{\theta \in \mathbb{S}^{d-1}} \left| (\tilde{\mu}_n |\theta^\mathsf{T} x|^p)^{1/p} - (|\tilde{\mu}(\theta^\mathsf{T} x)|^p)^{1/p} \right| \right] \\
&\quad + \|\Sigma_\nu\|_{\mathrm{op}}^{1/2} \left[ (2p)^{1/p} + \sup_{\theta \in \mathbb{S}^{d-1}} \left| (\tilde{\nu}_n |\theta^\mathsf{T} x|^p)^{1/p} - (|\tilde{\nu}(\theta^\mathsf{T} x)|^p)^{1/p} \right| \right] \\
&\quad + \sup_\theta |\mu(\theta^\mathsf{T} x)| + \sup_\theta |\nu(\theta^\mathsf{T} x)|
\end{aligned}
$$

Now, applying [2, Theorem 4.2] with $\epsilon = 1/2$ and $t = 1$, we get

$$
\mathbb{P}\left( \sup_{\theta \in \mathbb{S}^{d-1}} \left| (\tilde{\mu}_n |\theta^\mathsf{T} x|^p)^{1/p} - (|\tilde{\mu}(\theta^\mathsf{T} x)|^p)^{1/p} \right| > \frac{1}{2} \right) \leq \mathbb{P}\left( \sup_{\theta \in \mathbb{S}^{d-1}} \left| \tilde{\mu}_n |\theta^\mathsf{T} x|^p - \tilde{\mu}(\theta^\mathsf{T} x)|^p \right| > \frac{1}{2p} \right)
$$
$$
\leq 1 - e^{-c_p \sqrt{d}} \tag{22}
$$

under assumed constraints on $n$ in the statement. An analogous bound holds for $\nu$, which yields that

$$
\mathbb{P}\left( \hat{L}_n \geq (\|\Sigma_\mu\|_{\mathrm{op}}^{1/2} + \|\Sigma_\nu\|_{\mathrm{op}}^{1/2}) \left( (2p)^{1/p} + \frac{1}{2} \right) + \sup_\theta |\mu(\theta^\mathsf{T} x)| + \sup_\theta |\nu(\theta^\mathsf{T} x)| \right) \leq e^{-c_p \sqrt{d}}
$$

Recall that $\beta = \exp(-c_p \sqrt{d})$. Plugging (22) back into (21), we get

$$
\max_{\theta \in \mathbb{S}^{d-1}} \hat{w}_p(\theta) - \max_{1 \leq i \leq k} \hat{w}_p(\Theta_i) \leq L_{\mu,\nu} \left( \frac{\log(1/\delta)}{k} \right)^{1/d} \tag{23}
$$

with probability $1 - \delta - \beta$.

Finally, we have $\max_{\theta \in \mathbb{S}^{d-1}} \hat{w}_p(\theta) = \overline{W}_p(\hat{\mu}_n, \hat{\nu}_n)$, and

$$
|\overline{W}_p(\hat{\mu}_n, \hat{\nu}_n) - \overline{W}_p(\mu, \nu)| \leq \overline{W}_p(\hat{\mu}_n, \mu) + \overline{W}_p(\hat{\nu}_n, \nu).
$$

By (5b) in Proposition 7, for any $t > 0$,

$$
\mathbb{P}\left( \overline{W}_p(\hat{\mu}_n, \mu) \geq \alpha_{n,\mu} + t \right) \leq 2 \exp\left( -K_\mu \min\left( n^{1/p} t, n^{2/(2 \vee p)} t^2 \right) \right),
$$

$$
\mathbb{P}\left( \overline{W}_p(\hat{\nu}_n, \nu) \geq \alpha_{n,\nu} + t \right) \leq 2 \exp\left( -K_\nu \min\left( n^{1/p} t, n^{2/(2 \vee p)} t^2 \right) \right),
$$

where $K_\mu \lesssim d^{o_d(1)} \max\{\|\Sigma_\mu\|_{\mathrm{op}}^{1/2}, \|\Sigma_\mu\|_{\mathrm{op}}\}$ and $K_\nu \lesssim d^{o_d(1)} \max\{\|\Sigma_\nu\|_{\mathrm{op}}^{1/2}, \|\Sigma_\nu\|_{\mathrm{op}}\}$. Setting

$$
\gamma_n(t) = 2 \exp\left( -K_\mu^{-1} \min\left( n^{1/p} t, n^{2/(2 \vee p)} t^2 \right) \right) + 2 \exp\left( -K_\nu^{-1} \min\left( n^{1/p} t, n^{2/(2 \vee p)} t^2 \right) \right),
$$

we then have

$$
\mathbb{P}\left( |\overline{W}_p(\hat{\mu}_n, \hat{\nu}_n) - \overline{W}_p(\mu, \nu)| > \alpha_{n,\mu} + \alpha_{n,\nu} + 2t \right) \leq \gamma_n(t).
$$

Combining the above display with (23), we get the desired result.

# E  Additional Experiments and Details

Code for reproducing this paper's experiments can be found at `https://github.com/sbnietert/sliced-Wp`. Distance computations and plots for Figure 1 were performed on a cluster machine with 8 CPU cores and 64GB RAM in approximately 6 hours. Distance computations and plots for Figures 2 and 3 were performed on a cluster machine with 4 CPU cores and 20GB RAM in approximately 30 minutes. For Figure 3 (right), the lower bound on $W_1$ is computed by only considering couplings which leave the shared mass at 0 unmoved.

As an additional experimental setup along the lines of Figure 1, we consider Model (3): Gaussian mixtures $\mu = \frac{1}{10} \sum_{i=1}^{10} \mathcal{N}(\mu_{1,i}, \Sigma_{1,i})$ and $\nu = \frac{1}{10} \mathcal{N}(\mu_{2,i}, \Sigma_{2,i})$, where means $\mu_{1,i}$ and $\mu_{2,i}$ are respectively generated from $\mathcal{N}(\mathbf{1}_d, I_d)$ and $\mathcal{N}(3\,\mathbf{1}_d, I_d)$, and the covariance matrices of the mixtures are simulated as $\frac{1}{k} X^\intercal X$, where $X$ is $k \times d$ data matrix generated from $\mathcal{N}(0, I_d)$ and $k$ is a uniformly sampled integer from 1 to $d$. Conditioned on fixed random choices of $\mu$ and $\nu$, we provide the corresponding projection and sample complexity plots in Figure 4, with general trends matching those of Figure 1. For both this experiment and Figure 1 in the main text, the population versions of the distances where no closed forms exist were calculated by setting the number of samples and Monte Carlo directions to 5000 and 2000 respectively. Computations and plots were performed on a cluster machine with 8 CPU cores and 64GB RAM in approximately 12 hours.

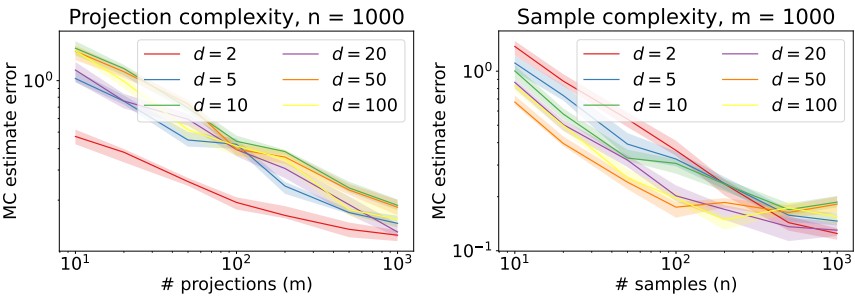

Figure 4: $\left| \widehat{\underline{W}}^2_{MC} - \underline{W}^2_2(\mu, \nu) \right|$ under Model (3).

Finally, we consider how the robustness properties of sliced $W_p$ may impact its application to generative modeling. Minimum distance estimation with respect to classic $W_1$ serves as a theoretical foundation for Wasserstein GANs [3, 22], a successful approach for training generative models. Later work extended this approach to average and max-sliced $W_p$ [18, 17], albeit at a slightly less direct level (in these papers, sliced distances are computed in a feature embedding space rather than raw image space). In Figure 5, we display samples generated from open source implementations of the standard Wasserstein GAN with Gradient Penalty (WGAN-GP) [22] and an average-sliced WGAN [18] trained for 20 epochs over the MNIST dataset [16] of digit images with 10% random noise contamination, using default parameter settings. Computations were performed on a cluster machine with 4 CPU cores, a NVIDIA Tesla T4 GPU, and 20GB RAM in roughly 12 hours. While there are differences between the produced samples, the two GAN architectures seem too distinct to draw any strong conclusions. Moreover, the robustness guarantees from Section 4 hold after preprocessing that appears too expensive to perform for data of this scale, so it is not surprising that the sliced WGAN reproduces random noise. Translating methods and guarantees for standard

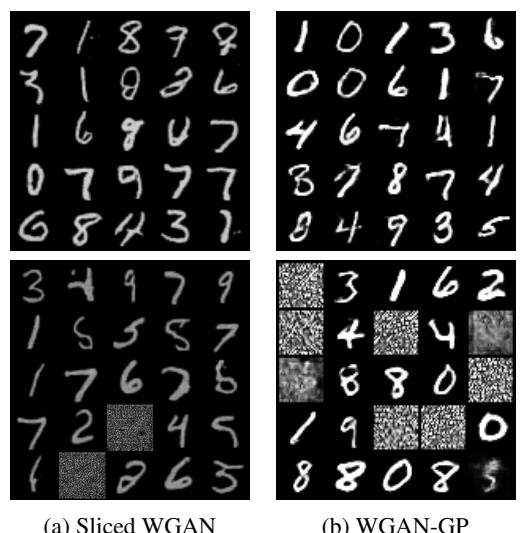

(a) Sliced WGAN  (b) WGAN-GP

Figure 5: Preliminary GAN experiments with uncontaminated (top) vs. contaminated (bottom) MNIST data.

WGAN robustification (e.g., [46]) to the sliced setting and thorough empirical comparisons are an interesting avenue for future research beyond the scope of this paper.