# OpenReview forum: "Statistical, Robustness, and Computational Guarantees for Sliced Wasserstein Distances"
_NeurIPS.cc/2022/Conference — NeurIPS 2022 Accept_

### Official Review · Reviewer_DAJE · 2022-07-06

**Rating:** 7
**Confidence:** 2
**Soundness:** 4 excellent
**Presentation:** 3 good
**Contribution:** 3 good

**Summary:**

The authors' contribution is threefold: computing empirical convergence rates, robustness to data contamination and improving computational methods of SW distances.

**Questions:**

No questions

**Limitations:**

no limitations or negative societal impact

**Strengths And Weaknesses:**

Strengths:
- Multiple new theoretical results are given and rates seem to be sharp.

Weaknesses:
- Empirical results are sparse, more could have been expected, for more complicated distributions/datasets, to show the theory is robust.
- I don't have the feeling that the authors compare Theorem 1 to other convergence rates (for instance Equation 7.120 of https://tel.archives-ouvertes.fr/tel-03533097/document ).
- The authors should develop more on the log-concave assumption: what are the practical implications for ML practitioners?
- I feel like the three parts of the papers do not have much in common and could each deserve a study on their own.
- What about distributional SW? Is it possible to compute such theorems for this distance?

I am willing to increase my score if I find the author's answer satisfying.

---

> ### Author Response · Authors · 2022-07-29
> **Response to Reviewer DAJE**
>
> - **Empirical results:** Indeed, our paper is primarily theoretical, with experiments provided mainly as proof-of-concepts to illustrate/validate the theory. Still, we agree with the reviewer that additional experiments with slightly more complex data models will strengthen the paper and our plan is to add those to the revision. Specifically, we will (i) repeat the experiment in Figure 2 with more complex Gaussian mixture models (e.g., $\mu = \frac{1}{k}\sum_{i=1}^{k} \mathcal{N}(x_i^\mu,\Sigma^\mu_i), \nu = \frac{1}{k}\sum_{i=1}^{k} \mathcal{N}(x_i^\nu,\Sigma^\nu_i)$ with random well-separated means, and random covariance matrices with an even spread of ranks, for different $k$ values); and (ii) add an experiment of generative modeling with contaminated datasets (e.g. 90% CIFAR-10 images, 10% MNIST images), comparing the performance of a standard Wasserstein GAN to a sliced Wasserstein GAN (see, e.g., "Generative Modeling using the Sliced Wasserstein Distance," Deshpande et al., CVPR 2018), along with a discussion of potential observed gains.
> - **Comparison of Theorem 1 to prior results:** Equation 7.120 of mentioned reference (identical to Corollary 2 of [34]) establishes a bound on convergence rates of $\underline{\mathsf{W}}_p$ under only moment assumptions. While we our regularity assumptions are stronger, we establish sharper rates in Theorem 1 and Proposition 1 compared to above. We will update Remark 2 to include discussions on [34] and thank the reviewer for pointing our this gap.
> - **Log-concave distributions:** Log-concavity is a primitive and easy-to-verify assumption that captures a broad class of important distributions. This encompasses distributions such as Gaussian, Laplace, exponential, chi, Dirichlet, Wishart, uniform over convex domains, and many more. Our results also allow for singular covariance matrices, which further broadens the class of distributions. Log-concavity is a much lower-level assumption than those typically imposed to derive empirical convergence rates for sliced $\mathsf{W}_p$, e.g., that the population measure satisfies certain functional inequalities ($T_p$ or Poincaré), finiteness of complicated functionals (e.g., the $\mathsf{SJ}_r(P)$ mentioned in the response to Reviewer 2), etc. For ML practitioners, this means that log-concavity is more workable in practice (to verify or negate), which makes our results easier to apply.
> - **Connection between parts:** Statistical rates, robustness to outliers, and computational efficiency are three important aspects in which classic OT suffers from inscalability to high dimensions. Sliced $\mathsf{W}_p$ mitigates this curse of dimensionality effect, but a comprehensive theoretical account was missing. Our work provides the theory to formally support scalability of both average- and max-sliced $\mathsf{W}_p$ on all three fronts. We acknowledge that covering for all three aspects resulted in the text being dense, but we view all three as critically important---both theoretically and to support principled application of ${\underline{\mathsf{W}}_p}$ and ${\overline{\mathsf{W}}_p}$ in practice.
> - **Distributional sliced $\mathsf{W}_p$:** This is an interesting question! We believe that our theory should extend to distributional sliced Wasserstein distance, given that the distribution over projections is sufficiently regular (e.g., Gaussian). We will mention this as an appealing future direction in the summary section.

---

> > ### Comment · Reviewer_DAJE · 2022-08-04
> > **Answer to the authors**
> >
> > I would like to thank the authors for their detailed answer and their additional work. I choose to increase my score by 1, under the condition that the promised experimental results  and a short vulgarization about log-concave distribution are added to the paper.

---

> > > ### Author Response · Authors · 2022-08-05
> > > **Thank you**
> > >
> > > We thank the reviewer again for their hard work and for raising their score. We can reassure the reviewer that the promised edits will be added to the camera-ready version of the paper.

---

### Official Review · Reviewer_diVV · 2022-07-11

**Rating:** 7
**Confidence:** 3
**Soundness:** 4 excellent
**Presentation:** 4 excellent
**Contribution:** 4 excellent

**Summary:**

This paper provides a comprehensive theoretical analysis on the scalability, convergence rates, and robustness of sliced Wasserstein distances and max-sliced Wasserstein distances. Specifically, the paper established sharp empirical convergence rate for sliced Wasserstein distances which is sharp up to log factors over the log-concave class. In addition, this work also investigated the robustness of (max) sliced Wasserstein distances to data contamination. For the computational guarantees, the Monte Carlo error bound of the sliced Wasserstein distance using finite slices was derived. Regarding the max sliced Wasserstein distances, although without formal computational guarantees, this paper analysed a subgradient-based local optimisation algorithm used in previous literature and proved a computational complexity bound for it. The experiment section validates the above theoretical analysis, including the projection and sample complexity, the computational cost of the subgradient max sliced Wasserstein distance, and the robustness of max sliced Wasserstein distance to data contamination.

**Questions:**

1. In Theorem 1, it is assumed that measure μ is log-concave, is this a common assumption in ML applications? What if this assumption does not hold?

2. What are “one sample” and “two sample” the paper refers to? I guess they refer to the case where the compared distributions are the same and the compared distributions are different, respectively. The author would better explain it to avoid confusion.

3. Is it possible to show the improved robustness of max sliced Wasserstein distances upon classic Wasserstein distances in a ML model?

**Limitations:**

Future work including improved complexity bounds for the subgradient methods for computing max-sliced Wasserstein distances are mentioned, although limitations of the work are barely mentioned.

**Strengths And Weaknesses:**

Strength:
1. This paper is well-written and well-structured, provided an in-depth analysis of the statistical properties, robustness to data contamination, and computational guarantees for sliced Wasserstein distances and max sliced Wasserstein distances.

2. The derivations are theoretically sound and provides a good reference for the community working on sliced Wasserstein distances.

Weakness:
Although not entirely a weakness due to the nature of theoretical work, the technical part of the paper, including notations and equations, are dense. It will be helpful if the paper can be made more accessible, so that researchers working on algorithmic work rather than theoretical work of SWDs can more easily extract conclusions from the theoretical results derived in the paper.

---

> ### Author Response · Authors · 2022-07-29
> **Response to Reviewer diVV**
>
> We appreciate the reviewer's positive feedback, and respond to the specific comments below:
> - **Density:** Statistical rates, robustness to outliers, and computational efficiency are three important aspects in which classic OT suffers from inscalability to high dimensions. Sliced $\mathsf{W}_p$ mitigates this curse of dimensionality effect, but a comprehensive theoretical account was missing. Our work provides the theory to formally support scalability of both average- and max-sliced $\mathsf{W}_p$ on all three fronts. We acknowledge that covering for all three aspects resulted in the text being dense, but we view all three as critically important---both theoretically and to support principled application of ${\underline{\mathsf{W}}_p}$ and ${\overline{\mathsf{W}}_p}$ in practice.
> - **Extracting main message/contributions:** We agree that crystallizing the main takeaways from the papers in an accessible manner is well in-place given the technical nature of the paper. In the camera-ready version we will expand Section 7 (Summary) to give brief, semi-formal statements of the main statistical, robustness, and computational results, so that they are easier to extract and utilize, without necessarily going into the full detail.
> - **Log-concave distributions:** Log-concavity is a primitive and easy-to-verify assumption that captures a broad class of important distributions. This encompasses distributions such as Gaussian, Laplace, exponential, chi, Dirichlet, Wishart, uniform over convex domains, and many more. Our results also allow for singular covariance matrices, which further broadens the class of distributions. Log-concavity is a much lower-level assumption that those typically imposed to derive empirical convergence rates for sliced $\mathsf{W}_p$, e.g., that the population measure satisfies certain functional inequalities ($T_p$ or Poincaré), finiteness of complicated functionals (e.g., the $\mathsf{SJ}_r(P)$ mentioned in the response to Reviewer 2), etc. For ML practitioners, this means that log-concavity is a more workable in practice (to verify or negate), which makes our results easier to apply.
> - **One- and two-sample.** The one-sample case refers to when only one of the two population distributions in approximated from samples, i.e., when, say, ${\underline{\mathsf{W}}_p}(\mu,\nu)$ is approximated by ${\underline{\mathsf{W}}_p}(\hat{\mu}_n,\nu)$. Analogously, two-sample refers to when both distributions are estimated, i.e., when the proxy is  ${\underline{\mathsf{W}}_p}(\hat{\mu}_n,\hat{\nu}_n)$. This is a standard terminology in statistics, but we will include a brief explanation in the text to avoid confusion. Then case of when $\mu=\nu$ or $\mu\neq\nu$ are referred to, respectively, as the "null" or the "alternative"; we will add a comment to clarify this as well.
> - **Improved robustness in ML model:** We agree with the reviewer that the paper would benefit from empirically demonstrating the robustness of sliced distances in realistic data models. To that end, we will add an experiment to the final version of generative modeling with contaminated datasets (e.g. 90% CIFAR-10 images, 10% MNIST images), comparing the performance of a standard Wasserstein GAN to a sliced Wasserstein GAN (see, e.g., "Generative Modeling using the Sliced Wasserstein Distance," Deshpande et al., CVPR 2018), along with a discussion of potential observed gains.
> - **Limitations:** Throughout the paper we tried to thoroughly discuss the limitations of our theory by clearly stating assumptions and comparing to related works. We also pointed out aspects in which our results could be improved. We did not comment on negative societal impact due the theoretical nature of our work, but are happy to address this in the revision if the reviewers thinks this is warranted.

---

> > ### Comment · Reviewer_diVV · 2022-08-06
> > **On the rebuttal**
> >
> > I thank the authors for the detailed response. It will be helpful if the limitations of the work can also be included in Section 7 (Summary) along with the semi-formal statements of main theoretical results.
> >
> > I noted that my question on "Is it possible to show the improved robustness of max sliced Wasserstein distances upon classic Wasserstein distances in a ML model?" is not fully answered. To clairfy, the max sliced Wasserstein distances I referred to is [deshpande2019].
> >
> > [deshpande2019] Ishan Deshpande et al., Max-Sliced Wasserstein Distance and its use for GANs, CVPR, 2019.

---

> > > ### Author Response · Authors · 2022-08-07
> > > **Thank you**
> > >
> > > We again thank the author for the helpful feedback. We are happy to incorporate limitations of this work into the conclusion, and to perform the GAN robustness comparison with a max-sliced Wasserstein GAN as in the reference.

---

### Official Review · Reviewer_ULQb · 2022-07-13

**Rating:** 7
**Confidence:** 4
**Soundness:** 3 good
**Presentation:** 4 excellent
**Contribution:** 4 excellent

**Summary:**

The Sliced-Wasserstein distance (SW) has been deployed in various machine learning applications over the past few years as a computationally efficient alternative metric to the Wasserstein distance. Due to its reported empirical success, recent work have studied the theoretical properties of SW to further motivate its use. For instance, SW has been shown to retain some important topological properties of the Wasserstein distance while yielding a dimension-free sample complexity, which justifies its efficiency on high-dimensional settings. On the other hand, a different line of work identified the limitations of SW and proposed solutions, including alternative metrics inspired by SW, such as the maximum Sliced-Wasserstein distance (max-SW).

This paper is a theoretical analysis of SW and max-SW, which refines existing guarantees for these metrics and explores unsolved questions. The contributions are described more precisely below.

1) Convergence rates of empirical distributions to the "true" distributions under SW or max-SW are derived (Section 3), assuming the true distributions are log-concave (Theorem 1) or supported on compact domains (Theorem 2). Empirical convergence rates under SW or max-SW have already been established in the literature under different assumptions. The authors argue that, as compared to prior work, their rates are faster or reflect more explicitly the influence of the data dimension (Remarks 1 and 2).

2) The robustness of SW/max-SW to outliers is also studied (Section 4): the minimax risk for robust estimation under SW/max-SW for a corrupted distribution (according to "total variation contamination") is characterized (Theorem 2, Corollary 1), under assumptions on the corruption level and on the moments of the projected distributions. The authors state their result is another confirmation that SW does not suffer from the curse of dimensionality, since the minimax risk does not contain a factor of $\sqrt{d}$ as opposed to the Wasserstein distance (Remark 4). Additionally, the "resilience" condition (which bounds the difference between the means of $\mu, \nu$ for $\nu \leq \frac{\mu}{1-\varepsilon}$) is shown to be equivalent to controlling max-SW$(\mu,\nu)$ (max-SW is of order 1) for $\nu \leq \frac{\mu}{1-\varepsilon}$ (Proposition 2). This allows the authors to extend algorithmic guarantees for robsut mean estimation (under the resilience condition) to robust estimation with max-SW.

3) Then, guarantees on the errors induced by the approximation of SW or max-SW are established (Section 5). First, the error associated to the Monte Carlo estimation of SW is bounded, assuming the distributions are log-concave -- which, similarly to Section 2, helps derive rates that are explicit in the parameters of the problems ($n, m, d, p$), as opposed to existing error bounds in related work (Remark 5). Then, convergence guarantees on a subgradient method used to approximate max-SW of order 2 (Algorithm 1) are derived (Proposition 4), provided $log(n) \leq \sqrt{d}$ and $\mu, \nu$ are log-concave. As discussed in Remark 7, some algorithms have been proposed in the literature and shown to be faster, but they are less used than the subgradient method and slower in practice (Figure 2). Note that these results rely on the fact that the Wasserstein distance between projections of distributions (as a function of the projection direction) is Lipschitz (Lemma 1).

4) Finally, the authors conduct a set of experiments on synthetic data in order to illustrate their theoretical findings (Section 6).

**Questions:**

Besides adding a discussion to [*] "Minimax Confidence Intervals for the Sliced Wasserstein Distance" (Manole et al., 2019), I encourage the authors to clarify the following points.

Theorem 1:
- Equation (2a): according to the proof, if I am not mistaken, $log(n)$ should be $log(n)^{1/2}$.
- Proof of Lemma 3(ii): For $p \geq 2$, $n^{1/2}$ is missing when bounding the RHS, since $|| u - u' ||_1 \leq n^{1/2} || u - u' ||_2$ (to verify).

Remark 2: the case $\Sigma = I_d$, whose norm indeed does not depend on $d$, is not realistic since components of a data point are more likely to depend on each other. Could the authors provide a more realistic example to illustrate the dependence on $d$ in their rates?

Proposition 1:
- Such rates were already established in the literature (e.g., [Section 4, 34] or [*]). Could the authors precise what is new in Proposition 1 as compared to prior work?
- Proof (end of Appendix D.2): it is unclear to me why the rates for SW$^p$ (or max-SW$^p$) can be applied to SW (or max-SW): how to get rid of $b^{1-p}$?

Remark 4: "slicing eliminates a $\sqrt{d}$ factor" This is true for max-SW, but not for SW whose minimax risk can contain such factor depending on the value of $d/q$ and $p/d$ (see Theorem 2). Could the authors clarify this aspect?

Proof of Lemma 1 (Appendix D.8): "the last step uses the optimal transportation cost formulation of $W_p$", this step is confusing to me. Could the authors explain why they can bound the sum of one-dimensional Wasserstein distances by the product of $||\theta_1 - \theta_2||$ and a sup over $\theta$?

Proposition 3: Could the authors confirm if the second term in the RHS is in $n^{p/2}$ when $p > 2$? It seems like it should be in $n^{-1}$ according to the proof of Proposition 3 and Theorem 1.

Section 6, "Projection and sample complexity": The increase of the errors with $d$ for Model (2) is not obvious on Figure 1(b). I think the authors should give more detailed explanations on why this behavior is expected (for both models) given Proposition 3: the provided explanations are quite vague (due to a lack of space, I guess).

Minor comments:
- Typos: l.114 "For for", l.122 "nongenerate", l.211 $\mathcal{R}^d$, Algorithm 1: in the title, $\tilde{\omega_2}^2$ but in the loop, $\tilde{\omega}_p^p$, l.290 "in not necessary", l.383: "providing rigorous justification the perceived scalability"
- l.91: the definition $\omega_p^p$ should be precised here. Same for $\omega_2(\theta)$ (l.98)
- The following operators should be defined: $\lesssim_p$, $poly(d)$, $\asymp$, $Conv$
- l.187: $\sigma$ is already used to denote the uniform distribution on the sphere.
- l.219: Unclear definition for $\delta_{\mathbb{B}^d}$
- The uncertainty of the empirical results is not reported and the authors say in Appendix E that error bars will be added in the final version of this paper. This aspect is important and should have been included in this version.
- Appendix D.9: Add a reference for the dual form of $W_p^p$



**Limitations:**

The limitations of the theoretical results are adequately discussed throughout the paper. The potential negative societal impact is not addressed.

**Strengths And Weaknesses:**

This work adequately addresses some important theoretical questions raised by the increasing use of Sliced-Wasserstein distance and its variants in machine learning applications. The contributions are mostly theoretical and have clear practical implications: they help quantify the efficiency and robustness of SW and max-SW according to the problem setting (i.e. in terms of data dimension and nature of distributions), or evaluate the quality of the traditional estimates of SW or max-SW. In that sense, I believe this work is significant and useful.

I find the paper very well written and clear. I strongly appreciate that the authors put effort into providing interpretations on their theoretical findings as well as discussions on their advantages and limitations over prior related work. Overall, the contributions of this paper, namely the theoretical results, their proofs and their empirical illustration, look sound to me. The proof techniques do not seem particularly original (e.g., they are direct consequences of existing results on one-dimensional Wasserstein distances derived in [5] or consist in assuming a specific structure for the distributions, i.e. log-concavity, a common practice in statistics), but they provide answers to relevant questions on SW in a simple and elegant manner.

The main weakness of this paper is that it is too dense. I am quite familiar with the questions addressed in this work and related work, which strongly helped me understand the main conclusions of the contributions, but I could not thoroughly check all the proofs in the supplement document. Besides, some aspects of the paper are unclear to me: more details are given in the "Questions" section of my review.

On the other hand, one important related work is missing and should be discussed as compared to the contributions: "Minimax Confidence Intervals for the Sliced Wasserstein Distance", Manole et al., 2019.

---

> ### Author Response · Authors · 2022-07-29
> **Response to Reviewer ULQb, Part 1**
>
> We thank the reviewer for the positive feedback and the detailed comments, which we address below:
> - **Density:** Statistical rates, robustness to outliers, and computational efficiency are three important aspects in which classic OT suffers from inscalability to high dimensions. Sliced $\mathsf{W}_p$ mitigates this curse of dimensionality effect, but a comprehensive theoretical account was missing. Our work provides the theory to formally support scalability of both average- and max-sliced $\mathsf{W}_p$ on all three fronts. We acknowledge that covering for all three aspects resulted in the text being dense, but we view all three as critically important---both theoretically and to support principled application of ${\underline{\mathsf{W}}_p}$ and ${\overline{\mathsf{W}}_p}$ in practice.
> - **Comparison to [\*]:** We thank the reviewer for pointing out [\*], which is indeed relevant. We will gladly cite and discuss it in the revision. First, we note that [\*] only considers average slicing (and mostly deals with a truncated version of $\underline{\mathsf{W}}_p$), so our results for the max-sliced distance are not covered by their work. The two relevant results therein for the untrucated average-sliced $\mathsf{W}_p$ are addressed below:
>      - **One-sample bound:** Proposition 4 of [\*] establishes an upper bound with parametric rate $n^{-1/2}$ for $\underline{\mathsf{W}}_p$ under a high-level condition that the functional $\mathsf{SJ}_r(P)$ is finite (here $P$ is the underlying distribution). However, for $r \ge 2$,  $\mathsf{SJ}_r(P)$ is not necessarily finite for log-concave measures. This can be seen via the lower bound for normal distributions in our paper, mentioned in Remark 1: for $\mu = \mathcal{N}(0, \Sigma)$ in $\mathcal{P}(\mathbb{R}^d)$, $\mathbb{E}[\underline{\mathsf{W}}\_2(\hat{\mu}\_n, \mu)] \gtrsim  \sqrt{\tau(\Sigma)^2 \log \log n / n}$, where $\tau(\Sigma)$ is the average of the square roots of eigenvalues of $\Sigma$ (the original submission featured $\|\Sigma\|\_{\mathrm{op}}$ in error). This is indeed not parametric, and thus the setup of our Theorem 1 is not covered by the results of [\*]. In addition, our bounds rely on the low-level assumption of log-concavity, which is significantly easier to verify than finiteness of $\mathsf{SJ}\_r(P)$.
>      - **Two-sample bound:** Theorem 2 of [\*] provides a two-sample bound that is looser by a logarithmic factor than the one in our Proposition 1. We do note, however, that while we assume a compact support, the two-sample bound from [\*] only requires finite projected moments.
> - **Theorem 1:** We thank the reviewer for pointing out a typo in the $\log$ term; it was corrected. As for the proof of Lemma 3(ii), we apply the inequality $\sum_{i=1}^n a_i^{p/2} \le (\sum_{i=1}^n a_i)^{p/2}$ with $a_i = |u_i - u'_i|^2$ when $p \ge 2$. This yields a factor of $n^{-1/p}$ as stated.
> - **Remark 2:** The operator norm of a positive semidefinite matrix agrees with its maximum eigenvalue, which is typically $O\_d(1)$ for appropriately scaled matrices considered in dependent data models (unlike the trace). As an example beyond $\Sigma=\mathrm{I}\_d$, where the covariance matrix corresponds to random vectors with dependent coordinates, consider a symmetric circulant matrix of the form $A = (m(|i-j|))\_{1 \le i,j \le d}$ for some function $m: \mathbb{N} \cup \{ 0 \} \to \mathbb{R}$ (circulant matrices often appear as autocovariance matrices in time series analysis). Then the modulus of every eigenvalue is bounded above by $\sum_{i=0}^d |m(j)|$, so the operator norm (when $A$ is positive semidefinite) is $O_d(1)$ if $\sum\_{j=0}^\infty |m(j)| < \infty$ (e.g. consider $m(j) = a^j$ for some $a \in (0,1)$; note that the corresponding circulant matrix is positive semidefinite). See Proposition 4.5.1 in "Time Series Analysis: Theory and Methods" by P.J. Brockwell and R.A. Davis. We will expand Remark 2 to clarify this point and add the above example.

---

> ### Author Response · Authors · 2022-07-29
> **Response to Reviewer ULQb, Part 2**
>
> - **Proposition 1:** The two questions brought up by the reviewer are addressed below:
>      - The convergence rate bounds from Corollary 2 of [34] are slower than those in our Proposition 1. That said, we note that their result only assumes bounded moments while our requires compact support. We will add a remark after our Proposition 1 to compare our result with  [Corollary 2, 34] and [Theorem 2, \*] and thank the reviewer for pointing these works out.
>      - To clarify, the factor of $b^{1-p}$ involved in extending the result from the $p$th power of the distance to the distance itself is given by $\underline{\mathsf{W}}_p(\mu, \nu)^{1-p}$ and $\overline{\mathsf{W}}_p(\mu, \nu)^{1-p}$ for the average- and max-sliced distances, respectively. We absorbed this quantity into the constant in the corresponding bound as it is non-random, strictly positive (under the assumption that $\mu\neq \nu$), and does not contribute to the dependence on $n$. We will clarify this point in the revision and mention that the resulting constants for the $\underline{\mathsf{W}}_p$ and $\overline{\mathsf{W}}_p$ two-sample bounds under the alternative depend on the population distance. We thank the reviewer for bringing this up.
> - **Remark 4:** While the factor $\sqrt{(1 \lor d/q)(1 \land p/d)}$ nominally involves $d$, separately considering the cases $d \leq p$, $p < d \leq q$, and $d > q$ yields that the above term is bounded above by 1. We will briefly comment on that in the remark for clarification.
> - **Proof of Lemma 1:** The final step, $$\mathsf{W}\_p\big(\mathfrak{p}^{\theta\_1}\_\sharp \mu, \mathfrak{p}^{\theta\_2}\_\sharp \mu\big) \leq \|\theta\_1 - \theta\_2\| \sup\_{\theta \in \mathbb{S}^{d-1}} \big(\mu |\theta^\intercal x|^p\big)^{1/p},$$
>     follows by upper-bounding the optimal cost $\mathsf{W}_p^p\big(\mathfrak{p}^{\theta_1}_\sharp \mu, \mathfrak{p}^{\theta_2}_\sharp \mu\big)$ by the cost associated with the (possibly sub-optimal) coupling given by the joint distribution of $(\theta_1^\intercal X, \theta_2^\intercal X)$, where $X \sim \mu$. We then use the relation $$ |(\theta\_1 - \theta\_2)^\intercal X| = \|\theta\_1 - \theta\_2\| \left | \left ( \frac{\theta\_1 - \theta\_2 }{\|\theta\_1 - \theta\_2\|} \right )^\intercal X \right |.$$ Further steps to clarify the above will be added to the camera-ready version.
> - **Proposition 3:** This is indeed a typo. The $p\vee 2$ term in the superscript should be $p\wedge 2$. We thank the reviewer for pointing this out---the statement has been corrected accordingly.
> - **Figure 1(b):** Proposition 3 bounds the error of the Monte Carlo estimate by $$\frac{ \|\mu x-\nu x\|^p + \|\Sigma\_\mu\|\_{\mathrm{op}}^{p/2}+ \|\Sigma\_\nu\|\_{\mathrm{op}}^{p/2}}{\sqrt{md}} +\frac{\big( \|\Sigma\_\nu\|\_{\mathrm{op}}^{p/2} + \|\Sigma\_\mu\|\_{\mathrm{op}}^{p/2} \big ) (\log n)^{\mathbb{1}\_{\\{p=2\\}}}}{n^{(p\wedge 2)/2}},$$ up to a $p$-dependent constant. Under the simulation model of Figure 1(b), the term $\|\mu x\mspace{-2mu}-\mspace{-2mu}\nu x\|^p$ grows at a rate $d^{p/2}$, while the operator norms of the covariance matrices remain constant. For $p=2$, this implies that the first term of the bound grows as $\sqrt{d}$, which is indeed reflected in the first plot on projection complexity (Figure 1(b) left).
> - **Minor comments:** We thank the reviewer for the careful reading of our paper and for pointing out these issues. In particular, as promised, we will make sure to add error bars to our plot in the camera-ready version.

---

> ### Comment · Reviewer_ULQb · 2022-08-08
> **Post-rebuttal**
>
> I thank the authors for their detailed feedback. I am convinced by their answers and I am keeping my positive score.
>
> Minor comment: Regarding the authors' answer to "Remark 2", I am glad that another example was given to better reflect their statement; however, it is unclear to me why the modulus of every eigenvalue (hence, the operator norm of the circulant matrix) does not depend on $d$, as it is given by $\sum_{j=0}^d |m(j)|$.

---

> > ### Author Response · Authors · 2022-08-08
> > **Thank you**
> >
> > We appreciate the reviewer reading our rebuttal and for their positive feedback. Regarding Remark 2, the $O_d(1)$ eigenvalue bound only holds for specific choices of $m$, e.g. $m(j) = a^j$, so that the sum is bounded independently of $d$. Of course, not every circular matrix has bounded eigenvalues; we just meant that a large subclass of circulant matrices does.

---

### Official Review · Reviewer_tcaq · 2022-07-14

**Rating:** 7
**Confidence:** 4
**Soundness:** 4 excellent
**Presentation:** 4 excellent
**Contribution:** 4 excellent

**Summary:**

The paper is focused on Sliced-Wasserstein (SW) and Max-Sliced-Wasserstein (Max-SW) distances and it provides important theoretical results on three fronts: 1)  empirical convergence rates, 2) robustness to data contamination, and 3) the error rates of Monte Carlo estimator for SW and the computational complexity bound for the subgradient-based local optimization that is often devised to calculate Max-SW distances. The results are well positioned with respect to existing work in the literature, and whether provide tighter bounds or looser assumptions. In particular, for log-concave distributions the authors show: 1) the sample complexity of SW and Max-SW are both $n^{-\frac{1}{max(2,p)}}$, the rate is sharp, and that the constants depend on the dimensionality of the problem, 2) under $\epsilon$ TV-contamination, the p-SW and p-Max-SW distances for the class of distributions with bounded q'th moment projections, and maximum covariance eigenvalue of $\sigma$,  have minimax rates of $\sigma\epsilon^{-\frac{1}{p}-\frac{1}{q}}$, and finally 3) the convergence of the subgradient-based local optimization commonly used for Max-SW distance is $O(\epsilon^{-4})$ and the results are obtained without the entropic regularization used previously in Lin et al. [NeurIPS 2020]. Lastly, all three contributions were numerically confirmed on simple problems.

**Questions:**

Minor editorial:

Line 114 - "For for a measurable map" must be "For a measurable map."



**Limitations:**

Given the more theoretical nature of the work, I do not foresee any potential negative societal impact of this work.

In addition, the authors have done a good job describing the limitations and future work (the room for improvement), and have been transparent about all assumptions used in their derivations.

**Strengths And Weaknesses:**

Strengths:

* The paper is well written, and well organized, and the authors have done a good job placing their work among the recent literature on this topic.

* The contributions improve the existing bounds/rates with fewer/less-restricted assumptions.

* The paper provides theoretical contributions on multiple fronts to a topic that is of interest to a broad audience.

Weaknesses:

* Not necessarily a weakness but the experimental section of the paper is on the lighter side, which makes sense given the heavy theoretical contributions.

---

> ### Author Response · Authors · 2022-07-29
> **Response to Reviewer tcaq**
>
>  We appreciate the reviewer's positive feedback. For the specific comments brought up:
> - **Experimental section:** Indeed, our paper is primarily theoretical, with experiments provided mainly as proof-of-concepts to illustrate/validate the theory. Still, we agree with the reviewer that additional experiments with slightly more complex data models will strengthen the paper and our plan is to add those to the revision. Specifically, we will (i) repeat the experiment in Figure 2 with more complex Gaussian mixture models (e.g. $\mu = \frac{1}{10}\sum_{i=1}^{10} \mathcal{N}(x_i^\mu,\Sigma^\mu_i), \nu = \frac{1}{10}\sum_{i=1}^{10} \mathcal{N}(x_i^\nu,\Sigma^\nu_i)$ with random well-separated means, and random covariance matrices with an even spread of ranks); and (ii) add an experiment of generative modeling with contaminated datasets (e.g. 90% CIFAR-10 images, 10% MNIST images), comparing the performance of a standard Wasserstein GAN to a sliced Wasserstein GAN (see, e.g., "Generative Modeling using the Sliced Wasserstein Distance," Deshpande et al., CVPR 2018), along with a discussion of potential observed gains.
> - **Typo in line 114:** Thank you for pointing this out; the typo was corrected.

---

> > ### Comment · Reviewer_tcaq · 2022-08-07
> > **Post Rebuttal**
> >
> > I thank the authors for their response. The planned experimental extensions sound great and could significantly increase the authors' impact.

---

### Meta-Review · Area_Chair_yMan · 2022-08-26

**Recommendation:** Accept
**Confidence:** Certain

**Metareview:**

All the reviewers are positive about the paper, they found that it is well written and provides very interesting theoretical as well as practical contributions which are relevant for the machine learning practice.

**Award:**

No

---

### Decision · Program_Chairs · 2022-09-14

Accept